# EFFICIENT STAGEWISE PRETRAINING VIA PROGRESSIVE SUBNETWORKS

**Abhishek Panigrahi**[†,*]   **Nikunj Saunshi** [α,*]   **Kaifeng Lyu**[†]   **Sobhan Miryoosefi**[α]
**Sashank Reddi**[α]   **Satyen Kale**[β]   **Sanjiv Kumar**[α]
[†] Princeton University       [α] Google, New York       [β] Apple MLR, New York

## ABSTRACT

Recent developments in large language models have sparked interest in efficient pretraining methods. Stagewise training approaches to improve efficiency, like gradual stacking and layer dropping (Reddi et al., 2023; Zhang & He, 2020), have recently garnered attention. The prevailing view suggests that stagewise *dropping* strategies, such as layer dropping, are ineffective, especially when compared to stacking-based approaches. This paper challenges this notion by demonstrating that, with proper design, dropping strategies can be competitive, if not better, than stacking methods. Specifically, we develop a principled stagewise training framework, *progressive subnetwork training*, which only trains subnetworks within the model and progressively increases the size of subnetworks during training, until it trains the full network. We propose an instantiation of this framework — **Ra**ndom **P**art **Tr**aining (RAPTR) — that selects and trains only a random subnetwork (e.g. depth-wise, width-wise) of the network at each step, progressively *increasing* the size in stages. We show that this approach not only generalizes prior works like layer dropping but also fixes their key issues. Furthermore, we establish a theoretical basis for such approaches and provide justification for (a) *increasing* complexity of subnetworks in stages, conceptually diverging from prior works on layer dropping, and (b) *stability* in loss across stage transitions in presence of key modern architecture components like residual connections and layer norms. Through comprehensive experiments, we demonstrate that RAPTR can significantly speed up training of standard benchmarks like BERT and UL2, up to 33% compared to standard training and, surprisingly, also shows better downstream performance on UL2, improving QA tasks and SuperGLUE by 1.5%; thereby, providing evidence of better inductive bias.

## 1 INTRODUCTION

Large network based language models (e.g. Transformers) have revolutionized the field of NLP. Intriguingly, these language models have demonstrated remarkable *emergent abilities* that only begin to manifest at large scales (Wei et al., 2022; Schaeffer et al., 2023). However, training such large models is usually very slow and resource intensive (Brown et al., 2020; Touvron et al., 2023; Chowdhery et al., 2022). This has sparked interest in efficient training of large models, necessitating the development of new algorithms and paradigms for efficient pretraining. Traditionally, this was accomplished by designing better optimization algorithms (e.g., (Chen et al., 2023; Gupta et al., 2018; Shazeer & Stern, 2018; Liu et al., 2023a)) that require fewer training steps to reduce the loss. Recently, other paradigms based on *stagewise training*, especially on depth, have garnered interest. Two such contrasting paradigms are: (a) layer stacking and (b) layer dropping.

Stacking based approaches have been studied extensively since Chen et al. (2021) applied it for BERT. Progressive (Gong et al., 2019) and gradual stacking (Reddi et al., 2023) are layer stacking methods that train large models in stages, starting with a small network and gradually growing the

---

[*] Equal Contribution, [†] Work done during internships at Google, [β] Work done when author was at Google, Correspondence to: `ap34@cs.princeton.edu`, `nsaunshi@google.com`

network size by stacking a subset of layers onto itself from previous stages. Although effective in reducing FLOPs and training time, their performance is sensitive to stacking schedules and require careful tuning. Furthermore, since the model grows gradually, it is not possible to assess the full model performance during earlier stages (i.e., it is not an anytime algorithm). Additionally, using small model with way fewer parameters for part of the training can possibly hamper model quality (e.g. long term memory (Geva et al., 2021)), especially for single epoch training where each data sample is seen once during training. In our experiments(§4.2), we observe that the pretraining loss and downstream performance may even be worse than baseline (full-model) training.

Layer dropping, on the other hand, maintains the model's identity but drops layers at random during training. This area of study remains largely unexplored. The closest work for training efficiency is progressive layer dropping (PLD) (Zhang & He, 2020), a *heuristic approach* where FLOPs are saved by increasingly *dropping more layers* as training proceeds, thus, decreasing its overall capacity over time. Such a strategy to improve training efficiency typically comes at the expense of quality, and is generally worse than stacking approaches (Kaddour et al., 2023). Through theoretical analysis and experiments, we first show that there is a fundamental problem with existing dropping strategies like PLD since dropping more layers later during training can be detrimental to *learning complex features*. Other instantiations (Fan et al., 2019; Zhang et al., 2019; Liu et al., 2023c) do not achieve training efficiency, as they were proposed as a regularization strategy or for improving inference efficiency. This naturally prompts the central question of the paper:

> *Is it possible to design principled and robust stagewise dropping based techniques that are competitive or better than stacking techniques?*

In this paper, we propose a novel generalization of stagewise dropping approaches called *progressive subnetwork training* that *fundamentally addresses the above concerns*, and provides an affirmative answer to this question. The key components of this framework are:

(P1) Maintain a common base model of interest and train *subnetworks of varying complexities* within the model.

(P2) Progressively *increase* the complexity of subnetworks in each stage to explicitly impose this simple-to-complex behavior.

For (P1), one can use very general subnetworks in this framework (e.g. subset of layers or layers with smaller width). This strictly generalizes layer dropping techniques where just layers are dropped to derive efficiency. Furthermore, (P2) is motivated by the theoretically and empirically studied phenomenon that gradient-based training learns functions of increasing complexity over time (Kalimeris et al., 2019; Abbe et al., 2022) and starkly diverges from earlier layer dropping ideas where more layers are dropped as training proceeds. As we shall see soon, even a simple instantiation of this idea can already be competitive or better than stacking approaches that have been extensively studied.

Given the above framework, a natural question arises: how do we select the subnetworks? As a concrete instantiation, we choose subnetworks to be *random* parts of the network, and in each stage we perform forward and backward passes over them. The size of the subnetworks is gradually increased over time with a fixed schedule. This approach, which we call **Ra**ndom **P**art **Tr**aining (RAPTR), reduces the total number of FLOPs and wall-clock time in the initial stages, but unlike stacking, every parameter of the full model contributes to the training loss at all stages, and it allows us to track the full model throughout training. In this paper, we demonstrate that this simple, yet powerful, strategy not only has a solid theoretical foundation, but also shows strong experimental results for BERT and UL2 pretraining, both in terms of training efficiency and improving model quality. We summarize the main contributions of the paper below:

- We introduce a novel stagewise training called progressive subnetworks that generalizes prior dropping strategies in §2. Specifically we explore Random Part training (RAPTR) that trains a part comprised of a random subnetwork, with the size of the subnetwork progressively increasing in stages. We specifically study variants of RAPTR where the random subnetwork selection is restricted to single model axis e.g. depth (section 2.1) or intermediate MLP width (section 4.3). We leave the combination of multiple axes for future investigation.

- Through analysis on polynomial data in §3, we demonstrate a fundamental problem with prior dropping techniques like progressive layer dropping – dropping more layers towards the end can

hurt the models' ability to capture complex correlations in the data. Furthermore in this polynomial setting, we show, both theoretically and empirically, that RAPTR can effectively learn higher order components of the underlying ground-truth function much better than progressive layer dropping. This provides a strong justification for progressively increasing complexity of subnetworks in stages. The polynomial setting can potentially aid future research in efficient training.

- We conduct extensive experiments on BERT (Devlin et al., 2018) and UL2 language models (Tay et al., 2022) to compare a depth instantiation of RAPTR with dropping techniques (PLD), and gradual stacking and its improved variants. On BERT-Base (§4.1), RAPTR demonstrates competitive performance to state-of-the-art stacking strategies at similar FLOPs, while being better than baseline[1] training with $1.33\times$ fewer FLOPs. For UL2-1.6$B$ language model (§4.2), RAPTR matches the pretraining perplexity of baseline training by requiring $1.2\times$ fewer FLOPs. Furthermore, despite having the same perplexity, RAPTR has much better performance on a suite of 12 downstream benchmarks, evaluated in 1-shot and 5-shot in-context settings. This suggests a desirable inductive bias of RAPTR that improves its generalization abilities beyond perplexity.

- We discuss a novel implementation strategy for subnetwork training translates FLOPs improvements to wall-clock speedups. This is particularly valuable for distributed training where naive conditional dropping implementations either fail or do not speedup training (appendix G).

- We establish the *first* theoretical basis for stagewise training based on dropping of layers that studies the behavior at stage transitions. In particular, we characterize stability conditions under which RAPTR yields smooth transition in loss across stage transitions (§5). We connect this to the idea of loss preservation (Chen et al., 2021) and show that RAPTR has an even *stronger property of loss improvement* in each stage. Through an illustrative setting with linear networks, we theoretically demonstrate the importance of modern architecture components, like layernorms and residual connections, for such stability.

## 2 PROGRESSIVE SUBNETWORK TRAINING

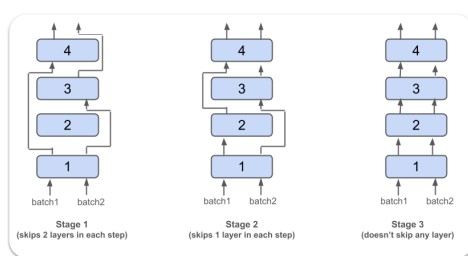

Figure 1: Pictorial description of stagewise RAPTR where the number of layers being skipped progressively decreases over stages.

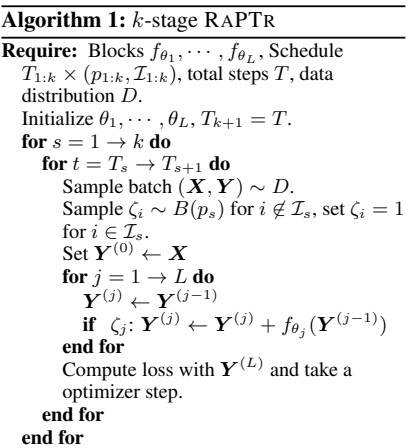

**Algorithm 1:** $k$-stage RAPTR

**Require:** Blocks $f_{\theta_1}, \cdots, f_{\theta_L}$, Schedule $T_{1:k} \times (p_{1:k}, \mathcal{I}_{1:k})$, total steps $T$, data distribution $D$.
  Initialize $\theta_1, \cdots, \theta_L, T_{k+1} = T$.
  **for** $s = 1 \rightarrow k$ **do**
    **for** $t = T_s \rightarrow T_{s+1}$ **do**
      Sample batch $(\boldsymbol{X}, \boldsymbol{Y}) \sim D$.
      Sample $\zeta_i \sim B(p_s)$ for $i \notin \mathcal{I}_s$, set $\zeta_i = 1$
      for $i \in \mathcal{I}_s$.
      Set $\boldsymbol{Y}^{(0)} \leftarrow \boldsymbol{X}$
      **for** $j = 1 \rightarrow L$ **do**
        $\boldsymbol{Y}^{(j)} \leftarrow \boldsymbol{Y}^{(j-1)}$
        **if** $\zeta_j$: $\boldsymbol{Y}^{(j)} \leftarrow \boldsymbol{Y}^{(j)} + f_{\theta_j}(\boldsymbol{Y}^{(j-1)})$
      **end for**
      Compute loss with $\boldsymbol{Y}^{(L)}$ and take a
      optimizer step.
    **end for**
  **end for**

We setup notation for the rest of the paper and describe the proposed framework of progressive subnetwork training and the specific method RAPTR that we study.

**Notation.** $[n]$ denotes the set $\{1, 2, \ldots, n\}$. $a_{1:k}$ is used to denote a sequence of $k$ scalars $a_1, \ldots, a_k \in \mathbb{R}$. $\boldsymbol{x}$ denotes a vector in $\mathbb{R}^d$. $\boldsymbol{x}_{1:k}$ denotes a sequence of $k$ vectors $\boldsymbol{x}_1, \ldots, \boldsymbol{x}_k \in \mathbb{R}^d$. We use $\boldsymbol{X} \in \mathbb{R}^{n \times d}$ to denote a matrix containing a sequence $\boldsymbol{x}_{1:n}$ as rows. We consider the setting where the goal is to learn an $L$-layer sequence-to-sequence neural network. (e.g., residual networks (with $N = 1$) and standard Transformer network).

**Definition 2.1** ($L$-layer sequence-to-sequence model). Let $f_1, \cdots, f_L : \mathbb{R}^{N \times d} \rightarrow \mathbb{R}^{N \times d}$ denote a set of $L$ sequence-to-sequence functions. Then, $F : \mathbb{R}^{N \times d} \times \mathbb{R}^L \rightarrow \mathbb{R}^{N \times d}$ takes an input sequence $\boldsymbol{X} \in \mathbb{R}^{N \times d}$, scalars $\alpha_{1:L}$, and outputs a sequence $\boldsymbol{Y}$ defined recursively with intermediate outputs $\boldsymbol{Y}^{(1)}, \cdots, \boldsymbol{Y}^{(L-1)}, \boldsymbol{Y}^{(L)}$ as

$$\boldsymbol{Y}^{(\ell)} = \boldsymbol{Y}^{(\ell-1)} + \alpha_\ell f_\ell(\boldsymbol{Y}^{(\ell-1)}) \text{ for all } 1 \leq \ell \leq L, \qquad F(\boldsymbol{X}; \alpha) = \boldsymbol{Y}^{(L)}, \qquad (1)$$

---

[1]We refer to full-model training as baseline training in our paper

where $\boldsymbol{Y}^{(0)}$ denotes $\boldsymbol{X}$, $\boldsymbol{Y}^{(\ell)}$ is the output of $\ell$-th layer. [2]

Standard model training uses scalars $\alpha_i = 1$ for $i \in [L]$. For simplicity, we use $F(\boldsymbol{X})$ to denote the output and $\boldsymbol{Y}^{(1)}, \boldsymbol{Y}^{(2)}, \cdots, \boldsymbol{Y}^{(L)}$ as the intermediate layer outputs of $F$ under standard model training, unless specified otherwise.

**Progressive subnetwork training** is a stagewise training method with two crucial components:

- **Subnetwork selection:** At each iteration, we select a (possibly random) subnetwork of the $L$-layer neural network. The forward and backward passes are computed based on this subnetwork.
- **Progressive subnetwork sizes:** The size of the subnetworks is progressively increased in a stagewise manner, starting from small subnetworks and ending with full network in final stage.

This framework is quite general since subnetworks can be selected along any model dimension, e.g., attention heads, hidden neurons in MLP layers. In this work, we primarily focus on depth.

### 2.1 RANDOM PART TRAINING: RAPTR

One strategy to select a subnetwork is choosing a *random part* within the model. To better isolate the selection mechanism, we consider variants where selection happens only along one model axis (e.g. depth or width or heads in Transformers). The systematic exploration of combination of multiple axes, which is a combinatorially large space, is left as future work. For the purpose of our discussion, we focus on the depth axis i.e., random layers of layers are selected at each step. Concretely, given a network $F$ to be trained, we pick a *random* subset of layers for the forward pass and bypass the rest of the layers by residual connections. With slight terminology abuse, we call this approach **Ra**ndom **P**art **Tr**aining (RAPTR). We will revisit RAPTR variants on other axes in §4.3.

More concretely, let $p$ denote the probability of randomly selecting a layer and $\mathcal{I}$ denote a subset of layers that are always included, i.e. they are never bypassed. We define a random subnewtork before describing RAPTR in more detail.

**Definition 2.2** (($p, \mathcal{I}$)-subnetwork). Let $\zeta_{1:L}$ be Bernoulli samples, with $\zeta_i = 1$ for all $i \in \mathcal{I}$, while $\zeta_i \sim B(p)$ for all $i \notin \mathcal{I}$. The set of all layers for which $\zeta_i = 1$ represents a random subnetwork while layers with $\zeta_i = 0$ are bypassed. The output of the selected subnetwork on a sequence input $\boldsymbol{X}$ is equivalent to the output of the model given by $F(\boldsymbol{X}, \zeta_{1:L})$ (1).

**Stagewise progressive RAPTR.** The pseudo-code is provided in Alg. 1. At a high level, the total training is split into $k$ stages. Each stage $s$ uses random $(p_s, \mathcal{I}_s)$-subnetworks between steps $T_s$ and $T_{s+1}$. We denote this schedule by $T_{1:k} \times (p_{1:k}, \mathcal{I}_{1:k})$. We progressively increase the random subnetwork selection pattern across stages, moving towards full model training by either increasing the probability of including each layer or fixing more layers in $\mathcal{I}$ or both. Such schedules satisfy: (a) $p_s \leq p_{\tilde{s}}$, and (b) $\mathcal{I}_s \subseteq \mathcal{I}_{\tilde{s}}$ for all $1 \leq s \leq \tilde{s} \leq k$."

**Training efficiency.** Training step in RAPTR involves computing forward and backward passes only on random subnetwork. Training a $(p, \mathcal{I})$ random subnetwork will require $(|\mathcal{I}| + (L - |\mathcal{I}|)p)/L$ FLOPs relative to standard training that uses all $L$ layers. By using $p \ll 1$ for majority of training, RAPTR can improve training efficiency.

**General behavior of layer probabilities.** RAPTR provides a general framework, where we can make discrete shifts in the average subnetwork lengths across stages. One potential concern is training instability due to discrete transitions. In §5, we show the stability of training loss of standard Transformers under RAPTR, and showcase the importance of different architectural components.

## 3 ILLUSTRATION: PROBLEMS WITH EARLIER SUBNETWORK TRAINING INSTANTIATIONS

In this section, we discuss problems with existing subnetwork training instantiations like progressive layer dropping (PLD). Recall that PLD drops *more* layers as training proceeds to enable FLOPs efficiency (see §D.1 for details). This is in contrast to the depth variant of RAPTR, that drops

---

[2] If $N = 1$, we represent the input and the intermediate outputs in their vector representation, $\boldsymbol{x}, \{\boldsymbol{y}^{(i)}\}_{i=1}^{L}$.

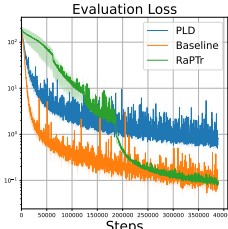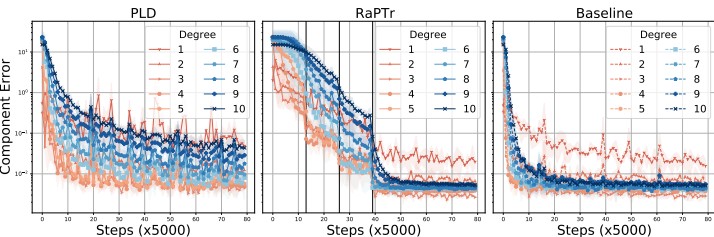

Figure 2: Evaluation loss (left) and component error (3 on the right) on basis polynomials for a 20 layer residual network. Labels are generated from polynomials of degrees 1 to 10 (Eq.2). Both RAPTR and PLD schedules use 20% fewer FLOPs than the baseline (dark vertical lines indicate RAPTR's phase transitions; details in §D.3). (a) RAPTR matches the baseline's evaluation loss, while PLD is worse. (b) RAPTR learns lower order terms faster and picks up higher order terms in the later stages. PLD is worse at capturing higher degree terms for its reduced expressivity in the end.

fewer layers as training proceeds. While this difference may seem superficial at first, it has deep implications from both conceptual and performance standpoints. Here, we argue through theoretical and empirical analysis on polynomial data, that this can have very significant impact on learning complex correlations in data. In the BERT and UL2 experiments in §4, we similarly observe that progressive layer dropping performs significantly worse than RAPTR.

We demonstrate the importance of progressively *increasing* the complexity of subnetworks in RAPTR (as opposed to progressively decreasing it in PLD) through a simple, yet instructive, setting: learning neural networks on polynomial data. For simplicity, we restrict the discussion to sequences of length $N = 1$.

**Polynomial learning setting.** Suppose $x$ are sampled from a uniform distribution over $\{\pm 1\}^d$. The ground-truth label is a polynomial $f^\star$ of degree $k$ defined as

$$F^\star(\boldsymbol{x}) = \sum_{l=1}^{k} \sum_{j=1}^{m} c_{l,j} \prod_{i \in S_{l,j}} x_i, \tag{2}$$

where $c_{l,j} \sim \mathcal{N}(0,1)$ and $S_{l,j}$ are random subsets of $[d]$ of cardinality $\ell$. Such polynomials have been studied in great detail (Abbe et al., 2022; 2023) where the higher degree terms capture more complex correlations and are harder to learn. We consider training of 20-layer residual network He et al. (2016) $F$ on such data using square loss: $\mathbb{E}(F(\boldsymbol{x}) - F^\star(\boldsymbol{x}))^2$. Here, each residual block consists of a single $4d$-hidden dimension MLP layer with ReLU activation.

**Empirical Observations.** We are interested in measuring how well each method captures the higher degree terms. To this end, we estimate the component of a learned function $F$ onto each basis polynomials. This can be done simply using $\hat{c}_{l,j} = \mathbb{E}_{\boldsymbol{x} \sim \{\pm 1\}^d} \left[ F(\boldsymbol{x}) \prod_{i \in S_{l,j}} x_i \right]$, since basis polynomials are orthogonal under the uniform distribution of the boolean data (O'Donnell, 2014). For each degree $l \le k$, we define the error as $\frac{\sum_{j=1}^{m}(c_{l,j} - \hat{c}_{l,j})^2}{\sum_{j=1}^{m} c_{l,j}^2}$. In Figure 2, we observe the following:

- PLD, that drops *more* layers as training proceeds, fails to effectively learn higher order complex correlations and, ultimately, hurts performance.
- RAPTR is competitive to baseline in terms of evaluation loss; in fact, RAPTR learns *every component* effectively. Furthermore, RAPTR quickly learns lower order terms and then picks up higher order terms during later stages.

The learning pattern of RAPTR is reminiscent of the implicit bias of standard SGD training to learn functions of increasing complexity (Kalimeris et al., 2019; Xu et al., 2019; Rahaman et al., 2019; Cao et al., 2019; Abbe et al., 2022). RAPTR naturally imposes this simple-to-complex behavior through size of subnetworks and also provides training efficiency in the process.

**Theoretical Analysis.** To further illustrate our point, we characterize the behavior of RAPTR for a simple 2-layer residual network, where each block consists of single neuron with a non-linear sine activation. We consider simple label function $F^\star(\boldsymbol{x}) = \frac{\sqrt{3}}{2} + \frac{\sqrt{3}}{2}x_1 - x_1 x_2$ for this analysis.

**Lemma 3.1** (Informal, cf theorem I.1). *For a small enough learning rate, 2-phase RAPTR first learns lower degree component and then the higher degree component.*

Table 1: Loss and fine-tuning results for BERT-Base after 675K steps and using equal peak learning rate ($10^{-4}$). Key observations: (a) RAPTR achieves similar loss as baseline at $1.33\times$ reduced FLOPs. (b) RAPTR achieves $0.02$ better evaluation loss than gradual stacking for same schedules. (c) RAPTR has a slight edge on downstream fine-tuning task.

| Method | Schedule | Rel. FLOPs | Eval Loss | MNLI | QNLI | SST-2 | Avg. |
|---|---|---|---|---|---|---|---|
| Baseline | - | 1.33 | 1.76 | 81.5 | **90.5** | 91.4 | 87.8 |
| PLD | - | 1. | 1.8 | 81.7 | 89.0 | 90.8 | 87.2 |
| Stacking | 6-8-10-12 | 1 | 1.78 | - | - | - | - |
| | 6-9-12 | 1 | 1.77 | 80.9 | 89.8 | 91.1 | 87.3 |
| RAPTR | 6-8-10-12 | 1 | **1.75** | **82.1** | 89.8 | **92.4** | **88.1** |
| | 6-9-12 | 1 | 1.75 | 82.3 | 89.2 | 91.0 | 87.5 |

In contrast, one layer is not expressive enough to represent the true label function (I.2). Thus, progressively dropping more layers (e.g., PLD) will reduce its expressivity. **Remark:** Stacking (Reddi et al., 2023) approaches, that grow the model size, can be shown to have similar simple-to-complex inductive bias in the above setting. We leave a detailed study of this connection to future work.

## 4 EXPERIMENTS

We present comprehensive experiments that compare RAPTR with baseline (full-model) training, gradual stacking and progressive layer dropping (PLD) for BERT and UL2. Since our primary goal was to make subnetwork based strategies competitive to stacking, we primarily focus on comparisons to gradual stacking. However, we also show comparisons on BERT with improved versions of gradual stacking, used in MSG (Yao et al., 2023), LEMON (Wang et al., 2023b), and bert2BERT (Chen et al., 2021). The details of relation between gradual stacking and these methods are in §D.2.

**RAPTR setup:** We describe the experimental setting of RAPTR from §2.1. We keep $\mathcal{I}$ equal to the first and last layer ($\mathcal{I} = \{1, L\}$) across all stages[3]. Additionally, scaling layer outputs during initial stages helps track validation loss better, though its final effect is minimal (details in §J). Finally, we use a constant learning rate in all phases, except in the final phase where we decay the learning rate, following gradual stacking (Reddi et al., 2023).

**Notations for RAPTR and stacking schedules:** We succinctly denote the stage schedules by a set of subnetwork lengths separated by hyphens. For stacking (RAPTR) training for a 24-layer model, 3-6-9-12 refers to 4 stages with $3, 6, 9,$ and $12$ layer (average subnetwork) training respectively. The length of each stage is chosen based on the reduction in FLOPs to achieve during training (see §E). Unless specified otherwise, we split the training steps equally across the stages.

**Wall clock speedups:** In In §F, we report wall-clock times for UL2 and BERT experiments. For UL2, a $1.2\times$ FLOPs reduction leads to a $1.19\times$ speedup, closely matching FLOPs improvements. For BERT, a $1.33\times$ FLOPs reduction gives a $1.26\times$ speedup. We also introduce a new implementation strategy for RAPTR in §G, essential for achieving speedups in distributed training where naive conditional dropping fails or doesn't give any speedup. As runtime depends on architecture and hardware, we focus on FLOPs reductions in the following discussion.

### 4.1 EXPERIMENTS WITH BERT

**Experiment setup.** We pretrain BERT models (Devlin et al., 2018) on Wikipedia + Books dataset with AdamW optimizer (Loshchilov & Hutter, 2019). We report the evaluation loss and fine-tuning performance of the trained models on 3 GLUE tasks (Wang et al., 2018) (refer to §D for details).

**Comparisons to gradual stacking and baseline:** We observe that RAPTR achieves similar or better evaluation loss to baseline training, despite baseline training using $1.33\times$ more FLOPs than RAPTR for BERT-BASE. Compared to gradual stacking at similar FLOPs, RAPTR again has better validation loss. Additionally, all three methods exhibit similar performance in downstream fine-tuning tasks (see Table 1). However, PLD performs significantly worse compared to others.

**Comparisons with variants of stacking:** We further compare RAPTR to variants of gradual stacking used in MSG (Yao et al., 2023), LEMON (Wang et al., 2023b), and bert2BERT (Chen et al.,

---

[3]Please see Appendix Table 10 and the discussion later for the relevant ablation study

Table 2: We compare the performance of RAPTR against depth-wise stacking operations used in bert2BERT, LEMON, and MSG, which implement slight variations of gradual stacking (see appendix D.2) after $300K$ steps of training. RAPTR demonstrates competitive or superior performance across all variants, achieving comparable loss to the best-performing version of gradual stacking.

| | Schedule | Loss | SST-2 | QNLI | MNLI | SNLI | RTE | MRPC | CoLA | QQP | Avg |
|---|---|---|---|---|---|---|---|---|---|---|---|
| bert2bert | 3-6-12 | 1.85 | 89.3 (0.1) | 84.5 (0.1) | 76.9 (0.0) | **87.0** (0.0) | 56.1 (0.7) | 75.2 (1.3) | 77.3 (0.4) | 89.3 (0.0) | 79.5 |
| MSG | 3-6-12 | 1.87 | 88.4 (0.2) | 83.9 (0.2) | 76.4 (0.3) | **87.0** (0.2) | 56.0 (1.1) | 74.8 (1.0) | 76.6 (1.1) | 89.4 (0.1) | 79.1 |
| MSG | 6-9-12 | 1.84 | **89.7** (0.3) | 85.6 (0.2) | **77.7** (0.1) | 86.6 (0.2) | **59.5** (1.1) | 77.2 (1.1) | 78.9 (0.2) | **89.6** (0.1) | 80.6 |
| LEMON | 6-9-12 | **1.83** | **89.8** (1.1) | **87.2** (0.0) | 77.3 (0.1) | 86.4 (0.1) | 58.4 (1.5) | 79.8 (0.6) | 78.0 (0.3) | **89.7** (0.1) | **81.1** |
| MSG (+ layer interpolation) | 6-9-12 | 1.84 | 89.0 (0.7) | 85.1 (0.0) | 77.5 (0.3) | 86.9 (0.1) | 57.7 (1.9) | **79.9** (0.8) | 77.5 (0.2) | 89.5 (0.0) | 80.4 |
| Stacking | 6-9-12 | 1.84 | 89.0 (0.4) | 85.3 (0.1) | 77.0 (0.1) | 86.2 (0.1) | 56.6 (1.3) | 76.4 (1.1) | 78.0 (0.4) | 89.3 (0.0) | 79.7 |
| RAPTR | 6-9-12 | **1.83** | 89.3 (0.6) | 85.8 (0.3) | 78.6 (0.2) | 87.0 (0.1) | 57.7 (0.5) | **80.8** (0.4) | 79.1 (0.7) | **89.6** (0.1) | **81.0** |

Table 3: Validation loss and 1-shot downstream evals on UL2-1.6B. Key observations: (a) RAPTR with 30k initial training improves performance by atleast $2 - 4\%$ on Trivia QA and SQuADv2. RAPTR is atleast $5\%$ better than gradual stacking on various downstream tasks, (b) Compared to baseline, RAPTR is $1 - 2\%$ better on all downstream tasks on average. (c) Stagewise training achieves $(1 - 2\%)$ lower variance on Tydi QA and SQuADv2, implying implicit benefits of stagewise training. **On an extensive suite of** $12$ **tasks and multi-shot settings, we observe a** $1.5\%$ **improvement in performance on average.** See table 5 in the appendix for extensive evaluations.

| Method | Rel. FLOPs | Eval Loss | Trivia QA | Tydi QA | SQuADv2 | SuperGLUE | Avg. |
|---|---|---|---|---|---|---|---|
| Baseline | 1.2 | 2.06 (0.01) | 25.0 (0.2) | 34.4 (3.1) | 42.1 (2.9) | 60.0 (0.4) | 40.4 |
| PLD | 1 | 2.09 (0.00) | 21.3 (0.3) | 32.4 (2.1) | 40.2 (0.9) | 59.9 (0.2) | 38.5 |
| 12-16-20-24 Stacking | 1 | 2.08 (0.00) | 20.1 (1.3) | 28.6 (2.4) | 36.0 (1.9) | 60.4 (0.9) | 36.3 |
| 12-16-20-24 RAPTR | 1 | **2.06** (0.00) | **25.8** (0.2) | **36.7** (1.0) | **44.1** (0.5) | **60.9** (0.2) | **41.9** |

2021) (details are in appendix D.2). We focus on depth-only stacking operations in these frameworks in table 2. We show that RAPTR performs competitive or even better than all variants of stacking, achieving same evaluation loss as the best tuned variant of stacking.

**Loss behavior during phase transitions:** Perhaps surprisingly, we observe that **the training loss decreases quite smoothly** when transitioning between two stages in RAPTR (see Fig. 3 (a) for BERT training under RAPTR). This is in contrast to stacking, where new variants have been developed to explicitly impose loss preservation across stage transitions (Shen et al., 2022; Wang et al., 2023b). We delve deeper into this favorable behavior of RAPTR in §5.

**Ablations with RAPTR parameters:** We observe robustness of the final trained model to different RAPTR schedules (Appendix Table 9). Furthermore, we observe that fixing the first and last layers at all steps during training helps RAPTR (Appendix Table 10). Since RAPTR is robust to schedule selections (Table 9), we recommend the schedule in the box below for all practical purposes.

---

**Recommended RAPTR schedule selection**

Schedule contains 4 phases. Initially, random subnetworks of size $L/2$ are selected, and we increase by $L/6$ layers in each transition until the full model is trained in the final phase.

---

## 4.2 EXPERIMENTS WITH UL2-1.6B

Experiments on BERT (§4.1) show that RAPTR can be competitive or better than gradual stacking and can outperform baseline training. We ask whether the same observations hold when training billion-parameter language models on large text corpora. We pretrain a 1.6B decoder-only UL2 model (Tay et al., 2022) with 24 layers for $400k$ steps with batch size 512. Table 3 reports the validation loss and downstream 1-shot performance without fine-tuning. Please see §D.5 for experimental details and table 5 for more extensive evaluations.

**Schedules for RAPTR and gradual stacking.** In table 3, we report for a schedule with 4 stages, denoted by 12-16-20-24. The length of each stage has been adjusted appropriately to use an average subnetwork length of 20 out of 24, giving $1.2\times$ FLOP reductions compared to the baseline training

Table 4: Comparisons at $300K$ steps of training on BERT-BASE. **(Top)**: Comparison between Attention-RaPTr and MSG's attention head growth operator (Yao et al., 2023). Both use schedules that use an average of 9 heads per layer, reducing FLOPs in self-attention by $1.33\times$. **(Bottom)**: Comparison between MLP-RaPTr and MSG's MLP growth operator. Both use schedules that use an average intermediate MLP dimension of 2048, reducing MLP FLOPs by $1.7\times$.

| | Eval Loss | SST-2 | QNLI | MNLI | SNLI | RTE | MRPC | CoLA | QQP | Avg |
|---|---|---|---|---|---|---|---|---|---|---|
| | | | | Head Operators (6-9-12) | | | | | | |
| MSG | 2.07 | 87.5 (0.1) | 83.4 (0.1) | 73.3 (0.3) | 85.5 (0.2) | 52.3 (0.4) | 72.5 (1.1) | 73.5 (0.2) | 88.6 (0.0) | 77.1 |
| Attention-RaPTr | **1.84** | 89.8 (0.6) | 86.4 (0.2) | 77.5 (0.1) | 89.5 (0.4) | 60.5 (1.3) | 80.4 (1.7) | 78.7 (0.1) | 89.7 (0.0) | **81.6** |
| | | | | Intermediate MLP size Operators (768-1536-3072) | | | | | | |
| MSG | 1.88 | **89.0** (0.3) | **85.5** (0.2) | 77.2 (0.2) | 87.2 (0.0) | **58.3** (0.6) | **80.0** (1.9) | 77.1 (1.0) | 89.6 (0.1) | **80.5** |
| MLP-RaPTr | **1.84** | 88.7 (0.3) | 85.3 (0.1) | **77.3** (0.3) | **88.6** (0.1) | 55.8 (1.9) | 79.8 (1.1) | **78.3** (0.2) | 89.6 (0.1) | 80.4 |

(see §E). RAPTR uses an initial full-model training phase for 30K steps before transitioning to its schedule. For parity in FLOPs, we reduce 30K steps from the final phase. This is useful for RAPTR– we attribute this to the fast drop in loss that large models can achieve initially. The ability to include full model training is another benefit of RAPTR over stacking due to its flexibility.

We summarize the key findings from table 3 below.

- **Perplexity results.** At $1.2\times$ FLOP reduction, RAPTR achieves similar evaluation perplexities to baseline training and also performs better than stacking (table 3).
- **Downstream inductive bias.** Intriguingly, we find that despite having similar perplexity, RAPTR has much better downstream metrics compared to both baseline and stacking. The improvements are particularly large for TydiQA (3.8%) and SQuADv2 (2.0%). This perhaps suggests RAPTR has a desirable inductive bias of RAPTR, a phenomenon that has been recently studied for other pretraining settings (Saunshi et al., 2022; Liu et al., 2023b).
- **Lower variance.** Another notable observation is that RAPTR has lower variance on all metrics and is more stable to training runs. These inductive bias benefits deserve further exploration.

## 4.3 SUBNETWORK TRAINING BEYOND DEPTH

Stacking based approaches have been studied for other dimensions beyond depth (e.g. width) by (Gu et al., 2020; Shen et al., 2022; Wang et al., 2023a; Gesmundo & Maile, 2023). In this section we demonstrate the flexibility of the subnetwork training framework to handle other growth operators. We extend RAPTR to intermediate MLP dimensions and attention heads. For attention heads, we apply stagewise training by randomly selecting a subset of heads in each layer, increasing the number of heads across stages. For MLP dimensions, we divide each MLP into 4 sub-MLPs, pick random subset of sub-MLPs in each layer to define a random subnetwork, and gradually increase the number selected at each stage. We compare these variants, named Attention-RAPTR and MLP-RAPTR, to their MSG counterparts. Attention-RAPTR and MLP-RAPTR perform better than their corresponding operators in MSG, providing further evidence of better performance than stacking-based approaches (table 4). More comprehensive study to other axes is left as future work.

## 5 LOSS PRESERVING BEHAVIOR OF RAPTR

RAPTR provides a general framework, as we can adjust the average size of random subnetworks in discrete steps between stages (§2.1). Training time instabilities are a possibility, as one might expect an arbitrarily large spike in the loss at stage transitions, due to the discrete shift in the training of the model. Remarkably, we observe that **the training loss decreases quite smoothly** when transitioning between two stages (see Fig. 3 (a) for BERT training under RAPTR). Hence, we ask the following:

> *What are the general conditions under which training loss under* RAPTR *stays stable at stage transitions?*

This question has been been extensively studied for stacking based approaches (Wang et al., 2023b; Chen et al., 2021; 2015; Shen et al., 2022). Here, we introduce, to our knowledge, the *first* framework to analyze models under discrete dropping based training strategies, and understand the necessity of different architecture components for robustness to discrete transitions. Related works

look into the generalization theory for dropout (Srivastava et al., 2014) as regularization (Gal & Ghahramani, 2016; Arora et al., 2021; Wager et al., 2013; Hu et al., 2019; Helmbold & Long, 2015). Building on these frameworks, our analysis relies on a crucial notion of stability (Arora et al., 2018). The stability analysis, when instantiated for linear networks highlights the importance of two popular architectural choices in Transformer models – residual connections and layer normalization.

## 5.1 LAYER STABILITY

We focus on sequences of length $N = 1$; generalization to $N > 1$ is fairly easy. Let $\mathcal{L} : \mathbb{R}^d \to \mathbb{R}$ represent the training loss of the backbone $F$. For transformers, $F$'s output passes through a Layer normalization layer before projecting to logits. This makes $\mathcal{L}$ scale-invariant (Ioffe & Szegedy, 2015; Ba et al., 2016), which gives a favourable stability condition as any perturbation in $F$'s output results in a proportional loss perturbation relative to the norm of $F$'s output. Thus, we make the following assumption.

**Assumption 5.1** (Relative-Lipschitzness). There exists a constant $\mu_{\mathcal{L}} > 0$ such that for any input $\boldsymbol{x} \in \mathbb{R}^d$, label $\boldsymbol{y}$ and perturbation $\eta \in \mathbb{R}^d$, $\mathcal{L}$ satisfies $\mathcal{L}(\boldsymbol{x} + \eta, \boldsymbol{y}) - \mathcal{L}(\boldsymbol{x}, \boldsymbol{y}) \leq \mu_{\mathcal{L}} \left( \|\eta\|_2 / \|\boldsymbol{x}\|_2 \right)$.

To understand loss changes across stage boundaries of RAPTR, we consider two stage RAPTR: the first stage trains with subnetwork of length $L - 1$ by dropping a single layer at random, and the second stage trains the entire model. Suppose $\mathcal{L}_1$ and $\mathcal{L}_2$ denote the effective loss functions being minimized in the two stages. The aim is to show that $F$ learned in the first stage by minimizing $\mathcal{L}_1$ should also result in small value for $\mathcal{L}_2$.

**Definition 5.2.** Let $F_{-\ell}$ denote the subnetwork after skipping layer $\ell$. The stability of network output with respect to dropping layer $\ell$ is defined as $\Psi_\ell(\boldsymbol{x}) = \|F_{-\ell}(\boldsymbol{x}) - F(\boldsymbol{x})\|_2$.

Next theorem shows that $\mathcal{L}_2(F) - \mathcal{L}_1(F)$ is small if stability scales slower relative to output norm.

**Theorem 5.3** (Informal, cf theorem H.1). *Under assumption 5.1 of loss $\mathcal{L}$, $|\mathcal{L}_2(F) - \mathcal{L}_1(F)|$ is upper bounded by $(\mu_{\mathcal{L}}/L)\mathbb{E}_{\boldsymbol{x} \in D}(\sum_{\ell=1}^{L} \Psi_\ell(\boldsymbol{x}))/ \|F(\boldsymbol{x})\|_2$.*

The proof follows by observing that $\mathcal{L}_1(F) = \frac{1}{L} \sum_{\ell \in [L]} \mathcal{L}(F_{-\ell})$ and $\mathcal{L}_2(F) = \mathcal{L}(F)$. Thus, the losses are close if $F_{-\ell}$ is close enough to $F$ for a random $\ell$. So, the relative stability of the network determines the success of the approach.

**Empirical verification with BERT:** In fig. 3 (b, c, d), we report the stability of BERT-LARGE during training. We observe that the norm of the layers grow linearly with depth, while the stability $\Psi_\ell$ grows sub-linearly with depth. This implies that our theoretical upper bound on the loss gap $\mathcal{L}_2(F) - \mathcal{L}_1(F)$ will show a rough linear decrease with the depth of the model. [4]

> We find that $\mathcal{L}_2(F)$ is lower than $\mathcal{L}_1(F)$, even when training with subnetworks of length $L-1$. This shows that RAPTR can reduce training losses at stage transitions without requiring explicit patterns, unlike stacking approaches that need carefully designed growth functions to avoid loss spikes (Wang et al., 2023b; Shen et al., 2022). Moreover, this loss reduction surpasses the concept of loss preservation (Chen et al., 2021), which opens up interesting future directions.

Instead, we focus on the following question, "when is a network stable (by def. 5.2) across the stages of RAPTR?" For simplicity, we consider the special case of linear residual networks. We show that layer normalization layers and the residual connection helps maintain the stability of RAPTR.

**Illustrative example: linear networks** We present a more concrete instantiation of Theorem 5.3 for residual network where the layers $f_{1:L}$ are linear with parameters $\boldsymbol{W}_{1:L}$. The layer output is $\boldsymbol{y}^\ell = \boldsymbol{y}^{(\ell-1)} + \boldsymbol{W}_\ell \boldsymbol{y}^{(\ell-1)}$ or $\boldsymbol{y}^\ell = \boldsymbol{y}^{(\ell-1)} + \boldsymbol{W}_\ell \boldsymbol{y}^{(\ell-1)}/ \left\| \boldsymbol{y}^{(\ell-1)} \right\|_2$ depending on whether layernorm is enabled or not respectively . We also study another setting with layernorm but no residual connection; so $\boldsymbol{y}^\ell = \boldsymbol{W}_\ell \boldsymbol{y}^{(\ell-1)}/ \left\| \boldsymbol{y}^{(\ell-1)} \right\|_2$.

**Lemma 5.4.** *For $\delta > 0$, a finite input set $S$ and dimension $d \geq \Omega(|S|L \log(1/\delta))$, w.p. atleast $1 - \delta$ w.r.t. random initialization $\boldsymbol{W}_\ell \sim \mathcal{N}(0, d^{-1/2}\boldsymbol{I})$, the following holds true for all input $\boldsymbol{x} \in S$.*

---

[4]Note here that we are talking about our theoretical upper bound, from which we can only infer $\mathcal{L}_2(F) \leq \mathcal{L}_1(F) + \mathcal{O}(1/L)$. However, empirically, we observe $\mathcal{L}_2(F) < \mathcal{L}_1(F)$, explaining this is kept for future work.

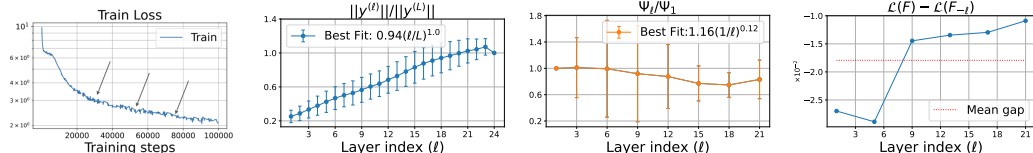

Figure 3: (left to right) (a): Training trajectory of BERT under 4-stage 6-8-10-12 RAPTR, with the stage transitions (denoted by arrows); (b), (c), (d): Stability study on BERT-LARGE trained for $50k$ steps by RAPTR with subnetworks of length $L - 1$. Behavior of (b) norms of intermediate activations $\boldsymbol{y}^{(\ell)}$, (c) $\Psi_\ell / \Psi_1$ (definition 5.2), and (d) Loss gap between different random subnetwork $F_{-\ell}$ and model $F$, given by $\mathcal{L}(F) - \frac{1}{L} \sum_{\ell=1}^{L} \mathcal{L}(F_{-\ell})$. Key observations: (b) Norms of the intermediate activations grow linearly with $\ell$, (c) $\Psi_\ell$ changes slowly with $\ell$ as $(L/\ell)^{0.12}$, suggesting a worse-case bound of $\mathcal{O}(L^{-0.88})$ on loss gap based on Theorem 5.3 (d) Interestingly, $\mathcal{L}(F) \leq \frac{1}{L} \sum_{\ell=1}^{L} \mathcal{L}(F_{-\ell})$, even when model is trained with $L - 1$ random subnetworks.

(a) *With residual connection & layernorm, $\Psi_\ell(\boldsymbol{x}) = \mathcal{O}(\sqrt{L/\ell})$ & $\|F(\boldsymbol{x})\|_2 = \Omega(\sqrt{L})$. Then the gap in losses between stages scales as $\mathcal{O}(1/\sqrt{L})$.*

(b) *Without residual connection, $\Psi_\ell(\boldsymbol{x}) = \Omega(1)$ & $\|F(\boldsymbol{x})\|_2 = \mathcal{O}(1)$. Thus the gap in losses between stages can be $\Omega(1)$.*

(c) *Without layernorm, we have $\Psi_\ell(\boldsymbol{x}) = \Omega(2^{(L-1)/2})$ and $\|F(\boldsymbol{x})\|_2 = \mathcal{O}(2^{L/2})$. Thus the gap in losses between stages can be $\Omega(1)$.*

In the Appendix, we show similar results for perfectly aligned layers (Lemma H.5). We can consider even more general scenarios, where the layers parameters are expressed as $\tau$-combinations of a Gaussian and a shared matrix (Appendix Fig. 4). We run simulations and observe that for each $\tau$, the loss gap between a $L - 1$ random subnetwork and the full model scales as $\mathcal{O}(L^{-0.4})$.

## 6 RELATED WORKS

Here, we focus on the most relevant works. Refer to appendix C for more related works.

**Stochastic depth.** RAPTR is similar to stochastic depth, which drops layers with a *fixed probability* during training to reduce the cost of training deep networks (Huang et al., 2016). A key distinction is that the probability of dropping layers is fixed during training and can thus be viewed as a regularization (Pham et al., 2019; Steiner et al., 2021; Tolstikhin et al., 2021; Liu et al., 2023c). Fixed stochastic depth has also been used for inference efficiency (Fan et al., 2019). (Devvrit et al., 2023; Valipour et al., 2023) explore nested training for dynamic inference with multiple subnetworks.

**Subnetwork training.** Training random paths and subnetworks has been used in other contexts, such as parameter-efficient fine-tuning (Houlsby et al., 2019; Pfeiffer et al., 2021; Hu et al., 2022; Liu et al., 2022) to reduce memory footprint, distributed and federated training (Dun et al., 2022), and incremental learning (Jathushan et al., 2019) to avoid forgetting in continual learning. However, these approaches are not aimed at reducing FLOPs during pretraining.

## 7 LIMITATIONS, CONCLUSION AND FUTURE WORK

We propose a stagewise training framework of *progressive subnetworks* for efficient pretraining, and evaluate a natural instantiation of this framework (RAPTR) based on training random paths/subsets of layers. Overall RAPTR yields better quality language models than baseline training, while showing a reduction of $1.2 - 1.33\times$ the total FLOPs. At the same speedup, it is also better than prior layer dropping and competitive to stacking based approaches.

The current analysis provides insights into loss stability at stage boundaries but does not explain the observed decrease in loss during transitions. Furthermore, schedule selection is not well understood due to expensive compute necessity and deserves more exploration. Our analysis does not explain the role of initial full-model warmup for UL2 and the desirable inductive biases of RAPTR style training, such as better downstream performance and smaller variance. A deeper understanding of these phenomena could lead to the development of more efficient training algorithms.

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

# Appendix

## Table of Contents

## A    BROADER IMPACT

Training LLMs demands extensive computational resources and infrastructure. Our paper proposes an algorithm that aims to accelerate their pre-training. Accelerating the training of large language models (LLMs) can significantly reduce their environmental impact by lowering energy consumption and minimizing carbon footprints. Optimizations in algorithms and infrastructure will lead to more efficient use of computational resources and will promote sustainability of the field of AI.

Furthermore, we observe benefits of structured pre-training (simple-to-complex) of these models on downstream tasks. Hopefully, these ideas can lead to better pre-training strategies for LLMs.

## B    MORE EXTENSIVE DOWNSTREAM TESTS ON UL2 MODELS

We conduct extensive downstream evaluation of the trained models from table 3 on additional downstream tasks and under higher shot in-context setting. Additional tasks include QA tasks like natural QA (Kwiatkowski et al., 2019), web QA (Berant et al., 2013), DROP (Dua et al., 2019), CoQA (Reddy et al., 2019), and QuAC (Choi et al., 2018); and completion tasks like Storyclose (Mostafazadeh et al., 2017), Hellaswag (Zellers et al., 2019), and LAMBADA (Paperno et al., 2016).

Table 5: We extensively compare all trained models from table 3 on multiple downstream tasks and few-shot in-context settings. We follow prompt design from Chowdhery et al. (2022) in each setting. For tasks marked with † and ⋆, we report Exact match and F1 scores respectively. For the rest, we use accuracy. The tasks have been sub-divided into 4 major groups, memorization QA, QA with context, completion, and SuperGLUE. On average, RAPTR is 1-2% better than baseline and stacking.

| | 1-shot | | | 5-shot | | |
|---|---|---|---|---|---|---|
| | Baseline | Stacking | RAPTR | Baseline | Stacking | RAPTR |
| Trivia QA† | 25.0 (0.2) | 20.1 (1.3) | 25.8 (0.2) | 26.5 (1.1) | 21.1 (1.3) | 25.1 (0.5) |
| Web QA† | 6.4 (0.4) | 5.8 (0.6) | 7.6 (0.5) | 10.6 (0.4) | 9.2 (0.5) | 11.2 (0.2) |
| Natural QA† | 4.2 (0.5) | 3.4 (0.4) | 4.4 (0.1) | 5.7 (0.1) | 4.6 (0.3) | 6.0 (0.3) |
| Tydi QA† | 34.4 (3.1) | 28.6 (2.7) | 36.7 (1.0) | - | - | - |
| SQuaDv2† | 42.1 (2.0) | 36.0 (0.9) | 44.1 (0.5) | 43.2 (3.0) | 36.2 (2.6) | 44.5 (1.3) |
| DROP† | 21.4 (0.8) | 19.5 (0.6) | 23.0 (0.4) | - | - | - |
| CoQA⋆ | 49.2 (0.7) | 43.9 (0.8) | 52.4 (0.7) | - | - | - |
| QuAC⋆ | 18.1 (0.5) | 16.8 (0.6) | 18.1 (0.4) | - | - | - |
| LAMBADA | 13.7 (2.9) | 12.0 (1.1) | 18.7 (3.1) | 30.5 (3.4) | 29.5 (2.1) | 38.7 (2.2) |
| Storycloze | 72.9 (0.4) | 71.0 (0.4) | 73.3 (0.4) | 75.1 (0.1) | 72.6 (0.8) | 75.3 (0.6) |
| Hellaswag | 58.3 (0.2) | 56.1 (0.1) | 58.5 (0.3) | 58.3 (0.2) | 56.1 (0.1) | 58.4 (0.3) |
| SuperGLUE | 60.0 (0.4) | 60.4 (0.9) | 60.9 (0.2) | 60.7 (0.3) | 58.8 (0.5) | 62.1 (0.2) |
| Average | 33.8 | 31.1 | 35.3 | 38.8 | 36.0 | 40.2 |

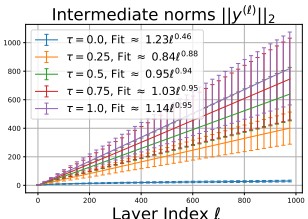 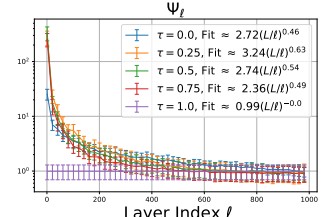 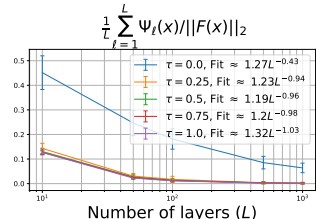

Figure 4: Behavior on a linear residual network with normalization layers with 100 random samples from $\mathbb{S}^{d-1}$, and dimension $d = 100$. The parameters of each layer $\ell$ is represented as $\sqrt{\tau}\boldsymbol{A} + \sqrt{1-\tau}\boldsymbol{G}^{(\ell)}$ for a shared matrix $\boldsymbol{A} \in \mathbb{R}^{d \times d}$ with $\|\boldsymbol{A}\|_2 \leq 1$ and $\boldsymbol{G}^{(\ell)} \sim \mathcal{N}\left(0, d^{-1/2}\boldsymbol{I}\right)$. Left to right: Behavior of (a) the norms of intermediate activation $\boldsymbol{y}^{(\ell)}$ with index $\ell$, (b) $\Psi_\ell$ (definition 5.2) for each stack of layers $F_{\ell:L}$, and (c) $\frac{1}{L}\sum_{\ell=1}^{L}\Psi_\ell(\boldsymbol{x})/\|F(\boldsymbol{x})\|_2$ that appears in our bounds in theorem 5.3.

## C   ADDITIONAL RELATED WORKS

**Residual networks as ensembles.** Training with random paths can be weakly viewed as ensembles over paths, and the connection between ResNets (He et al., 2016) and ensembles of shallow networks was first point out in (Veit et al., 2016) for vision models. (Dong et al., 2021) showed the same phenomenon for self-attention models, where longer paths lead to rank-collapse in the self-attention module. Chang et al. (2023) study vision transformers (Dosovitskiy et al., 2020) as a cascade of multiple paths and propose pruning and self-distillation to remove long paths and improve performance. All these works mainly provide a novel perspective or inference efficiency but do not focus on training efficiency.

**Learnable scaling of residual blocks.** Bachlechner et al. (2021), Zhang et al. (2019), Touvron et al. (2021) consider learnable scales on the output of the residual blocks. These works aim to understand favorable initialization of the learnable scales under various constraints for faster training, which is very different from the role of scaling in our algorithm.

Table 6: Additional rows from table 3: we compare baseline runs at two learning rate schedules, and run compare RAPTR and Gradual stacking with a new 6-12-18-24 schedule. **Key Observations**: (a) Cosine decay learning works best for the baseline, compared to Square-root schedule proposed by Tay et al. (2022), (b) The performance of gradual stacking improves with the new schedule, implying a performance dependence with appropriate scheduling, and (c) RAPTR's performance doesn't change much with the new schedule.

|  | Rel. FLOPs | Eval Loss | Trivia QA | Tydi QA | SQuADv2 | SGLUE | Avg. |
|---|---|---|---|---|---|---|---|
| Baseline (Square-root LR decay) | 1.2 | 2.07 | 23.4 | 31.9 | 44.3 | 60.0 | 39.9 |
| Baseline (Cosine LR decay) | 1.2 | 2.06 | 25.0 | 34.4 | 42.1 | 60.0 | 40.4 |
| 6-12-18-24 Stacking | 1 | 2.06 | 22.1 | 34.6 | 38.0 | 60.5 | 38.8 |
| 12-16-20-24 Stacking | 1 | 2.08 | 20.1 | 28.6 | 36.0 | 60.4 | 36.3 |
| 12-16-20-24 RAPTR | 1 | 2.08 | 22.2 | 38.2 | 40.6 | 60.1 | 40.3 |
| (+30k initial full-model train) | 1 | 2.06 | 25.8 | 36.7 | 44.1 | 60.9 | 41.9 |
| 6-12-18-24 RAPTR |  |  |  |  |  |  |  |
| (+30k initial full-model train) | 1 | 2.06 | 24.2 | 37.3 | 42.3 | 61.1 | 41.2 |

**Early exit for efficient inference** A lot of recent works have focused on improving inference efficiency for large language models. (Lei et al., 2023; Tay et al., 2022; Del Corro et al., 2023; Xin et al., 2020; Zhou et al., 2020; Hou et al., 2020). However, none of these works focus on efficient fine-tuning. Lei et al. (2023) modified pre-training by substituting resource-intensive MLP computations with straightforward classifiers for a predetermined fraction of tokens within the sequence. It's worth noting that their primary focus was on accelerating inference time rather than enhancing the efficiency of the pre-training process.

|  | Rel. FLOPs | Eval Loss | Trivia QA | Tydi QA | SQuADv2 | SGLUE | Avg. |
|---|---|---|---|---|---|---|---|
| Baseline | 1.2 | 2.06 | 25.0 | 34.4 | 42.1 | 60.0 | 40.4 |
| Equiflop Baseline | 1 | 2.07 | 24.3 | 35.7 | 42.4 | 60.6 | 40.7 |
| RaPTr | 1 | 2.06 | 25.8 | 36.7 | 44.1 | 60.9 | 41.9 |

Table 7: Performance comparison of models with relative FLOPs and various evaluation metrics.

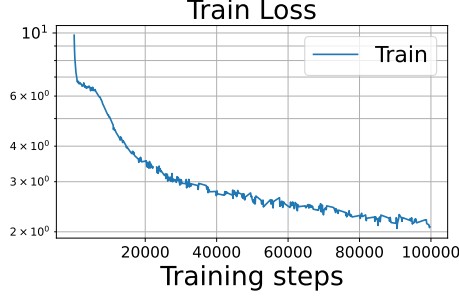
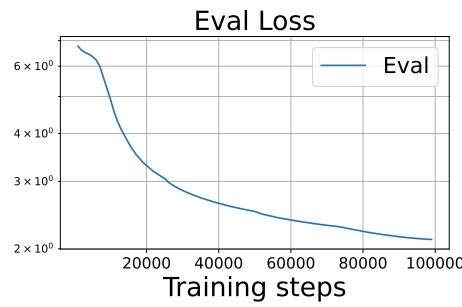

Figure 5: Train and Evaluation Loss behavior for a BERT-BASE model trained with RAPTR for 100k steps. We have 4 stages with 6-8-10-12 schedule (see section 4 for details). The boundaries are at 25k, 50k, and 75k. Key observation: the model's train and evaluation loss change smoothly across stage transitions, indicating stability of the model to subnetwork training.

# D ADDITIONAL EXPERIMENT DETAILS

## D.1 DETAILS ON PLD

The progressive layer dropping algorithm Zhang & He (2020) is given in algorithm 2. PLD employs a time and depth schedule that determines the probability of dropping each block. The time schedule

begins with a zero probability of dropping each block. Then it increases this probability throughout the training process until it reaches a maximum of $(1 - \bar{\alpha})$, where the hyperparameter $\bar{\alpha} = 0.5$. We vary $\bar{\alpha}$ depending on the average FLOP perplexity we want to achieve in the end. With $\bar{\alpha} = p$, the FLOPs reduce by $1 - (p/2)$.

The depth schedule ensures that blocks located earlier in the model are dropped with a lower probability than those located later/deeper. An important hyperparameter of the depth schedule in PLD is $\gamma_f$, which controls the rate at which the probability of dropping layers increases. A higher value of $\gamma_f$ leads to a quicker increase in the probability. (Zhang & He, 2020) set $\gamma_f$ to 100 in their experiments.

## D.2 DETAILS ON VARIANTS OF GRADUAL STACKING

Here, we give the details of the variants of gradual stacking, which we use to compare RAPTR against in table 2. We simply initiate the depth-wise growth operator of each method to compare to RAPTR.

**bert2BERT** (Chen et al., 2021) bert2BERT is a stacking based strategy that aims to progressively grow BERT models along width and depth at different stages of training. bert2BERT used progressive stacking (Gong et al., 2019) to grow the depth of bert models during the course of training. Here, the model's depth is doubled at each transition phase (e.g. 3 layer BERT $\rightarrow$ 6 layer BERT $\rightarrow$ 12 layer BERT).

**MSG** (Yao et al., 2023) MSG is a stacking based strategy that aims to progressively grow BERT models along 4 dimensions: width, depth, number of attention heads, and MLP intermediate dimension at different stages of training. The main recipe is a small warm-up during phase transitions, where the model is slowly allowed to grow from its previous configuration to its new one. This is done to preserve the loss during phase transitions. MSG uses gradual stacking to increase the depth of the model, where top few layers are repeated to initialize new models at phase transitions.

**LEMON** (Wang et al., 2023b) LEMON modified gradual stacking to instead use consecutive stacking. Here, the new layers are put consecutive to the source layers, whose parameters are copied to initialize the new ones. For example, for a schedule that grows a model from 6 layers to 9 layers, we choose the top 3 layers as source layers and insert new layers between layers $(4,5), (5,6), (6,)$ repeating the parameters of layers $4, 5, 6$ respectively.

**MSG with layer interpolation** We modify MSG's depth growth operator to LEMON's growth operttor. Here, instead of using the top few layers to stack on top of the existing model, we use consecutive stacking to put the newly initialized layers close to their source layers.

*Remark:* In order to keep the comparisons uniform across the methods, we use the same learning rate schedule as RAPTR across all methods.

## D.3 DETAILS FOR POLYNOMIAL TRAINING

We present in fig. 2 the results from a 20-layer residual network trained on 100-dimensional data. Our $f^\star$ has max degree $k = 10$ restricted to the first $t = 20$ variables ($\boldsymbol{x}_{1:20}$) in the input. We pick $m = 20$ random basis polynomials for each degree $\ell \leq 10$. We set the average drop rate of PLD to $\bar{\alpha} = 0.6$, such that the average FLOPs is $80\%$ lower to a baseline run. For RAPTR, we use 4-stage training, with the stages represented by its $(p, \mathcal{I})$ as $(8/20, \{\}), (12/20, \{\}), (16/20, \{\})$, and$(20/20, \{\})$ respectively. The lengths of each stage have been set such that the average FLOPs is $80\%$ lower to a baseline run (please see appendix E).

## D.4 DETAILS FOR BERT

**Equal step Pretraining** The batch-size is set to 256 and 512 for BERT-BASE and BERT-LARGE respectively. Following (Reddi et al., 2023), we use a learning rate warmup of 10k steps and keep learning rate constant in all phases, followed by a linear learning rate decay to 0 in the final full model training phase. We set the peak learning rate as $10^{-4}$ in all our experiments.

Table 8: Equal FLOPs comparisons for BERT-BASE and BERT-LARGE with extensive peak LR tuning. For BERT-BASE we use the best performing schedules from table 1. For BERT-LARGE, we use 6-12-18-24 for both RAPTR and stacking. FLOPs denotes the number of steps involved in baseline training. Key observations: for BERT-BASE (a) RAPTR achieves better loss compared to baseline at all FLOP measures, with larger differences for fewer FLOPs. (b) RAPTR has 0.02 better loss than gradual stacking at all FLOPs. (c) For BERT-LARGE, stacking and RAPTR are competitive to each other at all FLOPs. Both methods have better loss than baseline at lower FLOPs.

| Model | FLOPs | Baseline | Stacking | RAPTR |
|---|---|---|---|---|
| BERT-BASE | $75k$ | 2.09 | 2.02 | **2.01** |
| | $170k$ | 1.90 | 1.88 | **1.86** |
| | $510k$ | 1.74 | 1.75 | **1.73** |
| BERT-LARGE | $62.5k$ | 1.84 | **1.78** | 1.80 |
| | $140k$ | 1.63 | **1.60** | 1.61 |
| | $625k$ | **1.40** | 1.41 | 1.41 |

**Equivalent flop Pretraining** To make a fair comparison across all methods, we tune the peak the learning rate in the grid $\{1, 1.6, 2.5, 4\} \times 10^{-4}$ for BERT-BASE and grid $\{1, 1.6\} \times 10^{-4}$ for BERT-LARGE.

**Finetuning** We fine-tune BERT models for 3 epochs with batch-size 32, using a learning rate grid search over $\{1, 2, 3, 5\} \times 10^{-5}$. We use a linear warmup schedule over initial 6% training steps, followed by constant learning rate schedule for the rest of the training.

---

**Algorithm 2:** Progressive layer dropping (PLD) Zhang & He (2020)

---

1: **Input:** iterations $T$, layer keep probability $\bar{\alpha}$, temperature budget $\gamma_f > 0$, layers $L$, functions (self-attention, layer-norm, feed-forward) $f_{ATTN}, f_{LN}, f_{FFN}$, loss function $\mathcal{L}$, data $(\mathbf{x}_0, \mathbf{y})$, output layer $f_O$
2: $\gamma \leftarrow \frac{\gamma_f}{T}$
3: **for** $t \leftarrow 1$ to $T$ **do**
4: $\quad p \leftarrow 1$ {Keep probability.}
5: $\quad \alpha_t \leftarrow (1 - \bar{\alpha}) \exp(-\gamma \cdot t) + \bar{\alpha}$
6: $\quad p_d \leftarrow \frac{1 - \alpha_t}{L}$ {Layer decay.}
7: $\quad$ **for** $i \leftarrow 0$ to $L - 1$ **do**
8: $\quad\quad s \sim \text{Bernoulli}(p)$ {Keep or drop.}
9: $\quad\quad$ **if** $s == 0$ **then**
10: $\quad\quad\quad \mathbf{x}_{i+1} \leftarrow \mathbf{x}_i$ {Drop.}
11: $\quad\quad$ **else**
12: $\quad\quad\quad \mathbf{x}'_i \leftarrow \mathbf{x}_i + \frac{f_{ATTN}(f_{LN}(\mathbf{x}_i))}{p}$
13: $\quad\quad\quad \mathbf{x}_{i+1} \leftarrow \mathbf{x}'_i + \frac{f_{FFN}(f_{LN}(\mathbf{x}'_i))}{p}$
14: $\quad\quad$ **end if**
15: $\quad\quad p \leftarrow p - p_d$ {Decay prob.}
16: $\quad$ **end for**
17: $\quad \ell \leftarrow \mathcal{L}(f_O(\mathbf{x}_L), \mathbf{y})$
18: $\quad f_{ATTN}, f_{LN}, f_{FFN}, f_O \leftarrow \text{Update}(\ell)$
19: **end for**

---

### D.4.1 ADDITIONAL PRETRAINING EXPERIMENTS

**Results at equal FLOPs.** Inspired by (Kaddour et al., 2023), we further compare RAPTR and gradual stacking to baseline training by adjusting the number of training steps of baseline training to match its FLOPs to RAPTR (Table 8). For BERT-BASE, we observe that at shorter FLOP experiments, RAPTR achieves better validation loss compared to baseline training and gradual stacking. This difference gets smaller as we move to larger horizon settings. Similar findings for BERT-LARGE.

### D.5 DETAILS FOR UL2

**Pretraining**    Similar to our experiments on BERT, we use a linear warmup of 10K steps, followed by constant learning rate in all stages, and a cosine decay to $0.1\times$ the peak learning rate in the final full model training stage. The peak learning rate was searched over $\{10^{-2}, 5 \times 10^{-3}, 2 \times 10^{-2}, 3 \times 10^{-2}\}$ and was fixed to $10^{-2}$.

We use Adafactor (Shazeer & Stern, 2018) optimizer and train with a batch size 512 for $400k$ steps on a mixture of Arxiv, C4 (Raffel et al., 2020), Github, and Wikipedia (Foundation) datasets, with mixing ratios $9\%, 57\%, 17\%, 17\%$ respectively (for a total of 100B tokens). This roughly corresponds to 0.8 epochs of C4.

Reported datasets in table 3 include Trivia QA (Joshi et al., 2017), Tydi QA (Clark et al., 2020), SQuADV2 (Rajpurkar et al., 2018), and SuperGLUE (Wang et al., 2019). For QA tasks, we report Exact match scores, and report average accuracy for SupreGLUE tasks. To reduce variance, we report average performance across 3 runs for the most representative setting for each method.

Boundaries for 12-16-20-24 RAPTR and Gradual Stacking are 40k, 120k, and 240k (decided using appendix E). The average subnetwork length used during training turns out to be 20 with this schedule. When we introduce an initial 30k full-model training for RAPTR, we shift the boundaries by 30k so that the average subnetwork length stays 20.

### D.5.1 ADDITIONAL PRETRAINING EXPERIMENTS

In table 6, we have the following additional runs compared to table 3 in the main paper.

We first compare Baseline model with cosine decay learning rate to a Baseline model with Square-root learning rate decay that was used in the original paper (Tay et al., 2022). We observe that the model with square-root learning rate performs significantly worse compared to the model with cosine decay learning rate. Hence, we use model with cosine decay learning rate as our baseline.

Secondly, we try another schedule for Gradual Stacking, which contains 4 stages and is represented by 6-12-18-24. The stage lengths have been adjusted such that the average number of layers used over training is 20. We set the stage boundaries as 17k, 74k, and 172k respectively. We observe that 6-12-18-24 Gradual Stacking performs much better than 12-16-20-24 Gradual Stacking on various downstream tasks. This signals towards the dependency of Gradual Stacking on proper schedule selection.

On the other hand RAPTR doesn't show a big difference with 6-12-18-24 schedule, when compared to 12-16-20-24 schedule. A proper analysis of both the methods on schedule dependencies is kept for future work.

## E    SCHEDULE SELECTION

We follow two schedules from Reddi et al. (2023), Equal and Proportional, to build the lengths of each stage in RAPTR and Stacking. For Equal scheduling, used in Tables 1 and 8, we split training into $k$ equal stages. For Proportional scheduling in Table 3, we increase the length of a stage in proportion to index of the stage. For example, for 12-16-20-24 RAPTR and Stacking with proportional scheduling for $400k$ training steps, the stage lengths are set as $40k$, $80k$, and $120k$ respectively.

The flop counts are decided on the basis of average subnetwork length during training. For example, if the average subnetwork length is 18 for a model with 24 layers, we consider relative flops of RAPTR compared to baseline as 0.75.

For 12-16-20-24 RAPTR with Equal schedule, the average subnetwork length during training is given by 18, while for Proportional schedule, the average subnetwork length during training is given by 20. However, when either Equal and Proportional schedules don't return the target average subnetwork length, we re-consider the stage lengths by taking the closest candidate among Equal and Proportional schedules and remove equal steps from each of the first $k-1$ stages and add to the final full model training stage. For example, in table 6, with 6-12-18-24 RAPTR and target average subnetwork length 20, Proportional schedule returns an average subnetwork length of 18 and stage

lengths $40k$, $80k$, $120k$ and $160k$ respectively. We then aim to find $x$ such that stage lengths of $40k - x$, $80k - x$, $120k - x$ and $160k + 3x$ returns the desired average subnetwork length. On solving for $x$, we get $x = 22k$.

## E.1 ABLATION ON DIFFERENT SCHEDULES FOR BERT-BASE

We conduct experiments on RAPTR with different stage schedules that have same relative flop counts in table 1. We observe on BERT-BASE that different schedules differ by atmost 0.01 in pre-training loss and $0.1 - 0.5\%$ in downstream performance on average. Thus, we conclude that RAPTR is robust to the stage schedule selection.

For UL2, we use the following stage schedules to minimize computational overhead. We restrict the number of phases in RAPTR to 4. The first stage selects a sub-network with half of the layers at random at each step. We increase the sub-network size by an equal quantity in each stage transition, such that we train the full model in the final phase.

Table 9: Performance of RAPTR on BERT-BASE with different schedules. The average subnetwork size for each schedule is 9, which makes each schedule 1.33x faster than the baseline model (provided for reference). We use the same hyperparameters as used in table 1. We observe minor differences between different schedules, indicating robustness of RAPTR.

| RAPTR schedule | Rel. flops | Eval loss | MNLI | QNLI | SST-2 | Avg. |
|---|---|---|---|---|---|---|
| 3-6-9-12 | 1 | 1.76 | 82.0 | 89.6 | 92.0 | 87.9 |
| 4-8-12 | 1 | 1.76 | 81.9 | 89.3 | 91.5 | 87.6 |
| 6-9-12 | 1 | 1.75 | 82.3 | 89.2 | 91.0 | 88.0 |
| 6-8-10-12 | 1 | 1.75 | 82.1 | 89.8 | 92.4 | 88.1 |
| Baseline | 1.33 | 1.76 | 81.5 | 90.5 | 91.4 | 87.8 |

## E.2 ABLATION ON FIXED LAYER SET $\mathcal{I}$ FOR BERT-BASE

We conduct experiments on 6-8-10-12 RAPTR on BERT-BASE with different fixed layer set $\mathcal{I}$ (table 10). We observe that fixed set selection of $\{1, 12\}$ performs the best among other candidates. This selection however can be expensive to verify for other settings like UL2. Hence, in all subsequent experiments, we continue with the first and last layer fixed in all stages of RAPTR.

Table 10: Performance of RAPTR on BERT-BASE with different $\mathcal{I}$ set for 6-8-10-12 RAPTR run for $100k$ steps. We use the same hyperparameters as used in table 1, except the training steps reduced to $100k$ steps. We observe that fixing the first and the last layer at all times lead to slightly better performance, compared to other candidates.

| $\mathcal{I}$ set | Eval loss | MNLI | QNLI | SST-2 | Avg. |
|---|---|---|---|---|---|
| $\{\}$ | 2.12 | - | - | - | - |
| $\{1\}$ | 2.13 | - | - | - | - |
| $\{1, 12\}$ | **2.11** | 78.6 | 86.77 | 88.69 | 84.7 |
| $\{1, 2, 12\}$ | 2.13 | - | - | - | - |
| $\{1, 11, 12\}$ | 2.12 | 77.4 | 87.0 | 89.4 | 84.6 |
| $\{1, 2, 11, 12\}$ | 2.14 | 78.0 | 86.1 | 88.8 | 84.3 |
| $\{1, 2, 3, 11, 12\}$ | 2.18 | 77.4 | 86.5 | 88.4 | 84.1 |
| $\{1, 2, 10, 11, 12\}$ | 2.16 | - | - | - | - |

## F SYSTEM SPEED OF DIFFERENT SUBNETWORKS

We report the speed of training in steps/sec for UL2 models with different subnetwork sizes on a TPU 16x16 chips with percore batch size of 2 in table 11. Suppose $t$ is the total time for $400k$ steps with a 24-layer model in UL2 training. In table 3, we use 12-16-20-24 RAPTR that has

20% "theoretical" flop count reduction, with the stage lengths given by $40k$, $80k$, $120k$, and $240k$ respectively (appendix E). Theoretically, the necessary time required be $0.83t$. Using the training speed numbers of table 11, it takes approximately $0.84t$ to complete $400k$ steps with RAPTR on our machine, close to the theoretical training time.

Table 11: Training Steps/sec at different stages on 24-layer UL2

| subnetwork size | steps/sec |
|---|---|
| 12 | 2.0 |
| 16 | 1.6 |
| 20 | 1.4 |
| 24 | 1.1 |

We trained our BERT-base models on $4 \times 4$ TPU cores with percore batch size of $8$. We report the speed of training in steps/sec with different subnetwork sizes.

Table 12: Training Steps/sec at different stages on BERT-base

| subnetwork size | steps/sec |
|---|---|
| 6 | 17.9 |
| 8 | 14.3 |
| 10 | 11.8 |
| 12 | 10.2 |

So, looking at table 8, let $t$ be the total time for completing $675k$ steps in BERT-base training. In the case of a 6-8-10-12 RAPTR with Equal stage lengths, it takes approximately $0.79t$ to complete $675k$ steps on our machine. For an ideal machine that perfectly realizes the theoretical speedups based on the desired flop count reduction, the total time required will be $0.75t$.

## G  EFFICIENT IMPLEMENTATION OF RAPTR IN DISTRIBUTED TRAINING

**Issues with distributed training:**  RAPTR can be naively implemented with if-else statements that bypass a layer if a layer isn't part of a random sub-network at a specific time step. This will work with pytorch if we are training with a single GPU. However, if we train a model on multiple GPUs with pytorch, we get the following error.

> RuntimeError: Expected to have finished reduction in the prior iteration before starting a new one. This error indicates that your module has parameters that were not used in producing loss. You can enable unused parameter detection by passing the keyword argument 'find unused parameters=True' to 'torch.nn.parallel.DistributedDataParallel', and by making sure all 'forward' function outputs participate in calculating loss. Parameter indices which did not receive grad for rank 1: 33 34 35 36 37 38 39 40 41 42 43 44 45 46 47 48 49 50 51 52 53 54 55 56 57 58 59 60 61 62 63 64 65 66 67 68 69 70 71 72 73 74 75 76 77 78 79 80 81 82 83 84 85 86 87 88 89 90 91 92 93 94 95 96 97 98 99 100 101 102 103 104 105 106 107 108 109 110 111 112 113 114 115 116 117 118 119 120 121 122 123 124 125 126 127 128 129 130 131 132 ...

We get this error, because the optimizer expects gradients from each trainable model parameter from each GPU at each step of training. We get similar error, when we use jax framework to implement RAPTR.

**Solution – Layer Selection as Linear Algebraic formulation on Parameters:**  Instead of viewing RAPTR as selecting a random sub-network composed of a random selection of layers, we should instead view sub-networks as a set of selected parameters in the Transformer model. All Transformer layers are identical in structure but with different parameters.

Let $f^{(1)}, f^{(2)}, f^{(3)}, \cdots, f^{(L)}$ represent the $L$ Transformer layers with corresponding parameters $\theta^{(1)}, \theta^{(2)}, \cdots, \theta^{(L)}$. Each of these layers can be expressed as a common Transformer layer $f$, where

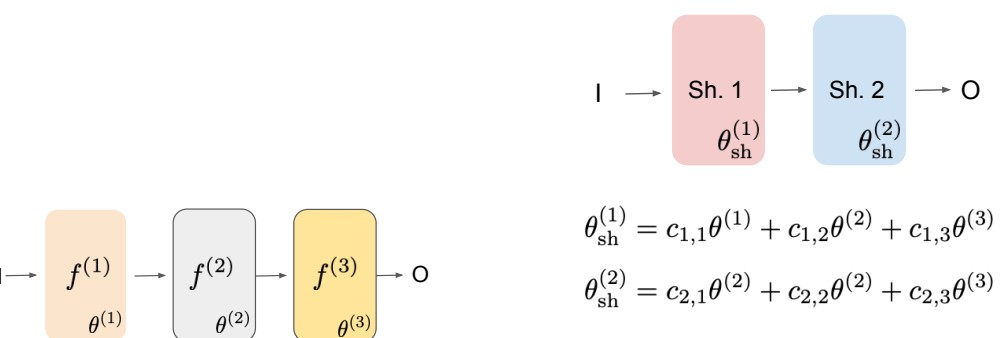

(a) A representative 3-layer network

(b) Shared base sub-network across the 3 possible 2-layer sub-networks

$$(c_{1,1}, c_{1,2}, c_{1,3}), (c_{2,1}, c_{2,2}, c_{2,3})$$

$(1, 0, 0), (0, 1, 0)$

$(1, 0, 0), (0, 0, 1)$

$(0, 1, 0), (0, 0, 1)$

(c) Representing each 2-layer sub-network in the shared base sub-network

Figure 6: Representative overview of efficient implementation of RAPTR in distributed training setting. A simple 'if-else' implementation of dropping layers doesn't work in distributed setting, as the optimizer expects gradients from each parameter from each GPU at each step of training. We instead represent all possible random sub-networks as a shared sub-network, whose parameters are represented as linear combination of all parameters. Then, we can back-propagate through a random sub-network by setting the linear coefficients appropriately.

the parameters $\theta^{(1)}, \theta^{(2)}, \cdots, \theta^{(L)}$ are used to represent $f^{(1)}, f^{(2)}, f^{(3)}, \cdots, f^{(L)}$, respectively. Furthermore, each layer $f^i$ can be represented with parameters $\sum_{j=1}^{L} c_{ij}\theta^{(j)}$, where $c_{ii} = 1$ and $c_{ij} = 0$ for all $j \neq i$.

We can extend this idea to a sub-network that selects $k$ layers at random from the $L$ Transformer layers. We can maintain a $k$-layer shared sub-network, where shared layer $1 \leq \ell \leq k$ uses parameter $\theta_{\text{sh}}^{(\ell)} = \sum_{j=1}^{L} c_{\ell,j}\theta^{(j)}$. $c_{\ell,j}$ are fixed depending on the random sub-network of layers that we pick at each step of training. For example, for a sub-network that picks layers $1, 2, 3, \cdots, k$, $c_{\ell,i}$ is set as 1 for $i = \ell$ for all $1 \leq \ell \leq k$ and 0 otherwise. In doing so, all the parameters get gradients in the loss computation at each step of training.

**Additional FLOPs involved in sub-network selection:** Because we have additional parameter additions to represent sub-networks at each step of training, one might wonder whether the additional FLOPs incurred will lead to drastic slowdown in model training. To do so, we will compare the number of FLOPs necessary for backpropagation for a $k$-random sub-network to the number of FLOPs necessary to compute gradients for a $k$-layer static sub-network used in stacking. The backpropagation step includes a forward pass, a backward pass, and a gradient descent step.

To further simplify our explanation, we will focus on an $L$-layer network that uses linear layers as its Transformer layers with parameters $\boldsymbol{W}^{(1)}, \cdots, \boldsymbol{W}^{(L)} \in \mathbb{R}^{d \times d}$. We will focus on Gradient Descent algorithm for simplicity. Batch size is given by $B$.

**Back-propagation FLOP Computation for a $k$-layer static sub-network:** For simplicity, we will use the first $k$ layer parameters to define the static sub-network. We will refer to the output of each layer $\ell$ as $\boldsymbol{Y}^{(\ell)} \in \mathbb{R}^{d \times d}$.

- **Forward pass:** For an input $\boldsymbol{Y}^{(\ell)} \in \mathbb{R}^{B \times d}$, the forward pass involves computing $\boldsymbol{W}^{(\ell)}\boldsymbol{y}_i^{(\ell)}$ for each $\boldsymbol{y}_i^{(\ell)} \in \boldsymbol{Y}^{(\ell)}$, for each layer $\ell \in [k]$. Thus, the number of FLOPs involved is $kBd^2$.

- **Backward pass:** On gradient $\boldsymbol{G}^{(\ell)} \in \mathbb{R}^{B \times d}$ back-propagating to a layer $\ell$, the backward pass computes the gradient w.r.t. the parameter $\boldsymbol{W}^{(\ell)}$ and the gradient that flows to the next layer.

  The gradient w.r.t. the weight is given as $\frac{1}{B}\sum_{i=1}^{B} \boldsymbol{g}_i^{(\ell)}(\boldsymbol{x}_i^{(\ell)})^\top$. This involves $Bd^2$ computation, and overall a $kBd^2$ computation for all weights.

  Furthermore, the gradient to be back-propagated beyond layer $\ell$ is given as $\left[\boldsymbol{g}_1^{(\ell)}(\boldsymbol{W}^{(\ell)})^\top, \cdots \boldsymbol{g}_i^{(\ell)}(\boldsymbol{W}^{(\ell)})^\top, \cdots, \boldsymbol{g}_B^{(\ell)}(\boldsymbol{W}^{(\ell)})^\top\right]$, which incurs an additional $Bd^2$ computation. Overall, a $kBd^2$ computation across all layers.

- **Gradient Descent step:** Finally, the descent step is a simple addition of negative gradients to each weight, which requires $kd^2$ for all weights.

**Back-propagation FLOP Computation for a $k$-layer random sub-network:** For a randomly picked sub-network, we will use $\boldsymbol{W}_{\text{sh}}^{(\ell)} = \sum_{j=1}^{L} c_{\ell,j}\boldsymbol{W}^{(j)}$ to denote the parameters of the $k-$layer shared sub-network that we use to perform back-propagation at particular step.

- **Forward pass:** Our proposed method requires an additional sum of weights in each layer, with the coefficients $c_{\ell,j}$ for $1 \leq \ell \leq k, 1 \leq j \leq L$ in the forward pass. Hence, the total FLOP computation involved is $kBd^2 + kLd^2$.

- **Backward pass:** Here, there are no changes for backpropagation through the $k$-layer shared network, as the gradients are computed and backpropagated w.r.t. $\boldsymbol{W}_{\text{sh}}^{(\ell)}$.

- **Gradient Descent step:** Here, the gradients need to be distributed to the weight parameters $\boldsymbol{W}^{(\ell)}$ before taking the descent step.

$$\nabla \boldsymbol{W}^{(\ell)} = \sum_{i=1}^{k} c_{i,\ell} \nabla \boldsymbol{W}_{\text{sh}}^{(\ell)}.$$

This requires $kLd^2$ computation, in addition to the necessary gradient descent steps.

Thus, the relative FLOPs of backpropagating through a $k$-layer random sub-network to a $k$-layer static sub-network (used in stacking) is

$$\frac{2kBd^2 + kd^2 + 2kLd^2}{2kBd^2 + kd^2} = 1 + \frac{2L}{2B+1}.$$

With $B$ set high, this relative FLOPs gap is close to 1.

**Discussion for language models:** For language models, the batch size represents the number of tokens that pass through the layer during backpropagation. In our UL2 experiments, sequences were of length 512 and the number of sequences in each minibatch was set as 2. Hence, the effective batch size is $B = 512 \times 2 = 1024$. The number of layers $L$ in our UL2 setting is 24. Hence, the relative FLOPs of backpropagating through a $k$-layer random network compared to a $k$-layer static network (used in stacking) would be $\approx (48/(2048 + 1) + 1) = 1.02$ if we had used the same minibatch for our linear network setting.

*Remark:* When training large models, we can structure the selection of random sub-networks to minimize the performance overhead associated with random sampling. For instance, if we are training a 100-layer network and wish to train a random sub-network of 50 layers at a given step, rather than selecting 50 layers arbitrarily from the entire model, we can organize the layers into 10 groups, each containing 10 layers. Then, we create a 50-layer shared sub-network, where each shared layer is a linear combination of 10 layer parameters from a specific group. This approach reduces the computational overhead associated with combining parameters from $50 \times 10 \times d^2$ to $5 \times 10 \times d^2$.

# H    NOISE STABILITY FOR TRANSFORMERS

**Theorem H.1.** *Suppose $\mathcal{L}$ satisfies the following condition $\mathcal{L}$ satisfies the following condition for any input $\tilde{\boldsymbol{x}}, \eta \in \mathbb{R}^d$ and label $\boldsymbol{v} \in \{0, 1\}^V$ for some constant $\mu_{loss}$ :*

$$\mathcal{L}(\tilde{\boldsymbol{x}} + \eta \left\| \tilde{\boldsymbol{x}} \right\|, \boldsymbol{v}) - \mathcal{L}(\tilde{\boldsymbol{x}}, \boldsymbol{v}) \leq \mu_{\mathcal{L}} \left\| \eta \right\|_2. \tag{3}$$

*The difference between expected loss of $(L-1)$-RAPTR and $L$-RAPTR, i.e. $|\mathcal{L}_2(F) - \mathcal{L}_1(F)|$, is upper bounded by $\frac{C}{L} \mathbb{E}_{\boldsymbol{x} \in D}(\sum_{\ell=1}^{L} \Psi_\ell(\boldsymbol{x})) / \left\| F(\boldsymbol{x}) \right\|_2$ for some constant $C$ that depends on regularity properties of the head $H$ on top of the backbone $F$.*

*Proof of theorem 5.3.* Denote by $F_{-\ell}$ the $L$ candidates that we can randomly choose from during $L-1$ RAPTR training. The output of $F_{-\ell}$ on any input $\boldsymbol{x}$ is equivalent to the output of the model $F$ on $\boldsymbol{x}$ and $\alpha_{1:L}$, with $\alpha_\ell = 0$ and the rest set to 1.

Then, the average loss of $L-1$ random training is given by

$$\mathcal{L}_2(F) = \frac{1}{L} \sum_{i=1}^{L} \mathbb{E}_{\boldsymbol{x}, \boldsymbol{y} \sim D} \mathcal{L}\left(F_{-\ell}(\boldsymbol{x}), \boldsymbol{v}\right).$$

On the other hand, the loss of the full model is given by

$$\mathcal{L}_1(F) = \mathbb{E}_{\boldsymbol{x}, \boldsymbol{v} \sim D} \mathcal{L}\left(F(\boldsymbol{x}), \boldsymbol{v}\right).$$

Hence, the expected difference between loss of $L-1$ random training and full model is given by

$$\mathcal{L}_2(F) - \mathcal{L}_1(F) = \frac{1}{L} \sum_{i=1}^{L} \mathbb{E}_{\boldsymbol{x}, \boldsymbol{v} \sim D} \left(\mathcal{L}\left(F_{-\ell}(\boldsymbol{x}), \boldsymbol{v}\right) - \mathcal{L}\left(F(\boldsymbol{x}), \boldsymbol{v}\right)\right)$$

$$\leq \frac{1}{L} \sum_{i=1}^{L} \mathbb{E}_{\boldsymbol{x}, \boldsymbol{v} \sim D} \quad \mu_{\mathcal{L}} \frac{\left\| F_{-\ell}(\boldsymbol{x}) - F(\boldsymbol{x}) \right\|_2}{\left\| F(\boldsymbol{x}) \right\|_2}$$

$$\leq \frac{1}{L} \sum_{i=1}^{L} \mathbb{E}_{\boldsymbol{x}, \boldsymbol{v} \sim D} \quad \mu_{\mathcal{L}} \frac{\psi_\ell}{\left\| F(\boldsymbol{x}) \right\|_2}.$$

Here the pre-final step uses the definition of $\mu_{\mathcal{L}}$ from eq. (3), and the final step uses the definition of $\Psi_\ell$ from definition 5.2. $\qquad \square$

**Discussion** An example of a loss function with bounded $\mu_{\mathcal{L}}$ is a transformer model, that uses a layer normalization layer before using an embedding matrix $\Phi$ to compute the logits. The logits are then compared with the true label with a cross-entropy loss with a softmax function on logits. $\psi(\mathcal{L})$ hence depends on the $\ell_2$ norm of $\Phi$ and the weight parameters of the layer normalization layer.

### H.1 Noise stability of linear residual networks

#### H.1.1 Useful lemmas

To prove, we will require the following theorem (and its variaions). For random initialization, we use $\boldsymbol{W}_1, \cdots, \boldsymbol{W}_L \sim \mathcal{N}(0, d^{-1})$.

**Lemma H.2** (Norm of the output of linear layers at initialization). *Consider the case of a linear network with no residual connections and layernorm, i.e.*

$$\boldsymbol{y}^{(\ell)} = \boldsymbol{W}_\ell \boldsymbol{y}^{(\ell-1)}, \text{ for all } \ell \geq 1,$$

*with $\boldsymbol{y}^{(0)} = \boldsymbol{x}$.*

*For an input $\boldsymbol{x}$, any $\varepsilon \geq 0$, with probability at least $1 - Le^{-\Omega(d\varepsilon^2/L^2)}$ over the randomness of $\boldsymbol{W}_1, \cdots, \boldsymbol{W}_L$, we have*

$$\left\| \boldsymbol{y}^{(\ell)} \right\|_2 \in [1 - \varepsilon, 1 + \varepsilon] \quad \text{for all} \ell \geq 1.$$

*Proof.* Because $\boldsymbol{W}_\ell$ is gaussian, $\boldsymbol{y}^{(\ell)} = \boldsymbol{W}_\ell \boldsymbol{y}^{(\ell-1)}$ follows $\mathcal{N}(0, \left\| \boldsymbol{y}^{(\ell-1)} \right\|_2^2 / d)$. Thus, $\{(\boldsymbol{y}_i^{(\ell)})^2\}_{i=1}^d$ follow a Chi-squared distribution, with $\mathbb{E} \left\| \boldsymbol{y}^{(\ell)} \right\|_2^2 = \mathbb{E} \sum_{i=1}^d (\boldsymbol{y}_i^{(\ell)})^2$ given by $\left\| \boldsymbol{y}^{(\ell-1)} \right\|_2^2$. The proof then follows from applying concentration bounds on the norm of gaussian vectors (Formula 3.7, Vershynin (2020)), i.e. for any $\varepsilon \geq 0$,

$$\Pr\left( \left| \left\| \boldsymbol{y}^{(\ell)} \right\|_2 - \left\| \boldsymbol{y}^{(\ell-1)} \right\|_2 \right| \geq \varepsilon \left\| \boldsymbol{y}^{(\ell-1)} \right\|_2 \right) \leq e^{-\Omega(d\varepsilon^2)}.$$

We apply a union bound over all layers, to get the final bound. $\qquad \square$

**Corollary H.3** (Norm of the output of linear layers with residual connections at initialization). *Consider the case of a linear network with residual connections but no layernorm, i.e.*

$$\boldsymbol{y}^{(\ell)} = \boldsymbol{y}^{(\ell-1)} + \boldsymbol{W}_\ell \boldsymbol{y}^{(\ell-1)}, \text{ for all } \ell \geq 1,$$

*with $\boldsymbol{y}^{(0)} = \boldsymbol{x}$.*

*For an input $\boldsymbol{x}$, any $\varepsilon \geq 0$, with probability at least $1 - Le^{-\Omega(d\varepsilon^2/L^2)}$ over the randomness of $\boldsymbol{W}_1, \cdots, \boldsymbol{W}_L$, we have*

$$\left\| \boldsymbol{y}^{(\ell)} \right\|_2 \in [2^{\ell/2}(1 - \varepsilon), 2^{\ell/2}(1 + \varepsilon)] \quad \text{for all } \ell \geq 1.$$

*Proof.* The proof follows similar to H.2. W.h.p., we can show that for any $\varepsilon \geq 0$,

$$\Pr\left( \left\| \boldsymbol{W}_\ell \boldsymbol{y}^{(\ell-1)} \right\|_2 - \left\| \boldsymbol{y}^{(\ell-1)} \right\|_2 \geq \varepsilon \right) \leq e^{-\Omega(d\varepsilon^2/L^2)}.$$

The only remaining piece is to show that $\boldsymbol{y}^{(\ell-1)}$ and $\boldsymbol{W}_\ell \boldsymbol{y}^{(\ell-1)}$ are roughly orthogonal. First of all, $\mathbb{E}(\boldsymbol{y}^{(\ell-1)})^\top \boldsymbol{W}_\ell \boldsymbol{y}^{(\ell-1)} = 0$ because $\boldsymbol{W}_\ell$ is concentrated around 0. Furthermore, $\mathbb{E}((\boldsymbol{y}^{(\ell-1)})^\top \boldsymbol{W}_\ell \boldsymbol{y}^{(\ell-1)})^2 = \sum_{i,j} w_{\ell,i,j}^2 (\boldsymbol{y}_i^{(\ell-1)})^2 (\boldsymbol{y}_j^{(\ell-1)})^2 = \frac{1}{d} \left\| \boldsymbol{y}^{(\ell-1)} \right\|^4$. Thus, using Hanson wright inequality (theorem 6.2.1, Vershynin (2020)), we have w.p. atleast

$$\Pr\left[ \left| (\boldsymbol{y}^{(\ell-1)})^\top \boldsymbol{W}_\ell \boldsymbol{y}^{(\ell-1)} \right| \geq \varepsilon \left\| \boldsymbol{y}^{(\ell-1)} \right\|^2 \right] \leq e^{-\Omega(d\varepsilon^2)}.$$

We apply a union bound over all layers, to get the final bound.

$\qquad \square$

**Corollary H.4** (Norm of the output of linear layers with residual connections at initialization). *Consider the case of a linear network with residual connections and layer-normalization, i.e.*

$$\boldsymbol{y}^{(\ell)} = \boldsymbol{y}^{(\ell-1)} + \boldsymbol{W}_\ell \boldsymbol{y}^{(\ell-1)} / \left\|\boldsymbol{y}^{(\ell-1)}\right\|_2, \text{ for all} \ell \geq 1,$$

*with $\boldsymbol{y}^{(0)} = \boldsymbol{x}$.*

*For an input $\boldsymbol{x}$, any $\varepsilon \geq 0$, with probability at least $1 - Le^{-\Omega\left(d\varepsilon^2/L^2\right)}$ over the randomness of $\boldsymbol{W}_1, \cdots, \boldsymbol{W}_L$, we have*

$$\left\|\boldsymbol{y}^{(\ell)}\right\|_2 \in [\sqrt{\ell+1}(1-\epsilon), \sqrt{\ell+1}(1+\epsilon)] \quad \text{for all } \ell \geq 1.$$

*Proof.* The proof stays extremely similar to H.3. For simplicity, we simply refer to H.3 for any such variations. □

### H.1.2 PROOFS OF LEMMA 5.4

*Proof of lemma 5.4.* We outline the proof for each case.

1. With residual connection and layer normalization, the function $F$ computes the intermediate activations $\boldsymbol{y}^{(1)}, \cdots, \boldsymbol{y}^{(L)}$ on an input $\boldsymbol{x}$ as follows.

$$\boldsymbol{y}^{(\ell)} = \left(\boldsymbol{I} + \frac{\boldsymbol{W}_\ell}{\left\|\boldsymbol{y}^{(\ell-1)}\right\|}\right) \boldsymbol{y}^{(\ell-1)}.$$

By corollary H.4, we can show w.h.p. with respect to the randomness of $\boldsymbol{W}_1, \cdots, \boldsymbol{W}_L$, for all $\ell \leq L$, $\sqrt{\ell}(1 - \mathcal{O}(1)) \leq \left\|\boldsymbol{y}^{(\ell)}\right\|_2 \leq \sqrt{\ell}(1 + \mathcal{O}(1))$.

Ignoring the error terms, we can simply write

$$\boldsymbol{y}^{(\ell)} = \left(\boldsymbol{I} + \frac{\boldsymbol{W}_\ell}{\sqrt{\ell}}\right) \boldsymbol{y}^{(\ell-1)}.$$

Pick a general layer $\ell$ to be dropped. Then,

$$F_{-\ell}(\boldsymbol{x}) - F(\boldsymbol{x}) = \prod_{\ell'=\ell+1}^{L} \left(1 + \frac{\boldsymbol{W}_{\ell'}}{\sqrt{\ell'}}\right) \frac{\boldsymbol{W}_\ell}{\sqrt{\ell}} \boldsymbol{y}^{(\ell-1)} + err,$$

where the $err$ term appears because of the change in scales of activations $\boldsymbol{y}^{(\ell+1)}, \cdots, \boldsymbol{y}^{(L)}$ with the dropping of layer $\ell$. This error can be bounded as $\mathcal{O}(\sqrt{L}/\ell)$ using the same procedure followed below to bound the first term on R.H.S..

Similar to bounding the norms of $\boldsymbol{y}^{(\ell)}$, w.h.p. we can show that

$$\left\|\prod_{\ell'=\ell+1}^{L} \left(1 + \frac{\boldsymbol{W}_{\ell'}}{\sqrt{\ell'}}\right) \frac{\boldsymbol{W}_\ell}{\sqrt{\ell}} \boldsymbol{y}^{(\ell-1)}\right\|_2 \leq \sqrt{\frac{L}{\ell}}.$$

Hence,

$$\psi_\ell := \|F_{-\ell}(\boldsymbol{x}) - F_\ell(\boldsymbol{x})\|_2 \leq \mathcal{O}(\sqrt{L/\ell}).$$

This implies

$$\frac{1}{L} \sum_\ell \psi_\ell(\boldsymbol{x}) = \frac{1}{L} \sum_\ell \mathcal{O}(\sqrt{L/\ell}) = \mathcal{O}(1).$$

We then apply a union over all $\boldsymbol{x} \in S$ such that the above condition holds true for all inputs in set $S$. Since the gap in $\mathcal{L}_2$ and $\mathcal{L}_1$ is bounded by $\mathcal{O}(\mathbb{E}_{\boldsymbol{x}} \frac{1}{L} \sum_\ell \psi_\ell(\boldsymbol{x})/\|F(\boldsymbol{x})\|)$ from theorem 5.3, we have the gap as $\mathbf{O}\left(\frac{1}{\sqrt{L}}\right)$.

2. With no normalization, the function $F$ looks as follows.

$$\boldsymbol{y}^{(\ell)} = (\boldsymbol{I} + \boldsymbol{W}_\ell)\,\boldsymbol{y}^{(\ell-1)}.$$

By corollary H.3, we can show w.h.p. with respect to the randomness of $\boldsymbol{W}_1, \cdots, \boldsymbol{W}_L$, for all $\ell \le L$, $2^{\ell/2}(1 - \mathcal{O}(1)) \le \left\|\boldsymbol{y}^{(\ell)}\right\|_2 \le 2^{\ell/2}(1 + \mathcal{O}(1))$.

With a drop in layer, we get

$$F_{-\ell}(\boldsymbol{x}) - F(\boldsymbol{x}) = \prod_{\ell'=\ell+1}^{L} \left(\boldsymbol{I} + \boldsymbol{W}^{\ell'}\right) \boldsymbol{W}_\ell \prod_{\ell'=1}^{\ell-1} \left(\boldsymbol{I} + \boldsymbol{W}^{\ell'}\right) \boldsymbol{x}.$$

Similar to corollary H.3, we can show w.h.p. with respect to the randomness of $\boldsymbol{W}_1, \cdots, \boldsymbol{W}_L$, $\left\|\prod_{\ell'=\ell+1}^{L} \left(\boldsymbol{I} + \boldsymbol{W}^{\ell'}\right) \boldsymbol{W}_\ell \prod_{\ell'=1}^{\ell-1} \left(\boldsymbol{I} + \boldsymbol{W}^{\ell'}\right) \boldsymbol{x}\right\|_2 \ge \mathcal{O}(2^{(\ell-1)/2})$.

This implies $\|F_{-\ell}(\boldsymbol{x}) - F(\boldsymbol{x})\|_2 = \Omega(2^{(\ell-1)/2})$.

3. Without residual connection, the function $F$ looks as follows.

$$\boldsymbol{y}^{(\ell)} = \boldsymbol{W}_\ell \frac{\boldsymbol{y}^{(\ell-1)}}{\left\|\boldsymbol{y}^{(\ell-1)}\right\|_2}.$$

Similar to corollary H.4, we can show w.h.p. with respect to the randomness of $\boldsymbol{W}_1, \cdots, \boldsymbol{W}_L$, for all $\ell \le L$, $(1 - \mathcal{O}(1)) \le \left\|\boldsymbol{y}^{(\ell)}\right\|_2 \le (1 + \mathcal{O}(1))$. Thus, ignoring error terms, the network $F$ roughly looks like

$$\boldsymbol{y}^{(\ell)} = \boldsymbol{W}_\ell \boldsymbol{y}^{(\ell-1)}.$$

With a drop in layer, we get

$$F_{-\ell}(\boldsymbol{x}) - F(\boldsymbol{x}) = \prod_{\ell'=\ell+1}^{L} \boldsymbol{W}_{\ell'}(\boldsymbol{W}^{(\ell)} - \boldsymbol{I}) \prod_{\ell'=1}^{\ell-1} \boldsymbol{W}_{\ell'} \boldsymbol{x}.$$

Using randomness of $\boldsymbol{W}^{(\ell)}$ this can be shown to be of norm $\Omega(1)$.

$\square$

**Lemma H.5.** *When the layers are perfectly aligned, i.e. all weights $\boldsymbol{W}_i = \boldsymbol{A}$ for some matrix $\boldsymbol{A}$ with $\|\boldsymbol{A}\|_2 = 1$ and for simplicity, assume second eigenvalue $\lambda_2(\boldsymbol{A}) = 1 - \delta$ for some $\delta > 0$, we have*

(a) *With residual connection & layernorm, we have $\Psi_\ell(\boldsymbol{x}) = \mathcal{O}(1)$ and $\|F(\boldsymbol{x})\| = \Omega(L)$. Thus, the gap in losses between stages is $\mathcal{O}(1/L)$.*

(b) *Without residual connection, we have both $\Psi_\ell(\boldsymbol{x}) = \mathcal{O}(\delta^{-1}e^{-\ell})$ and $\|F(\boldsymbol{x})\| = \mathcal{O}(1)$. Thus the gap in losses between stages is $\mathcal{O}(\frac{1}{L})$.*

(c) *Without layernorm, we have $\Psi_\ell(\boldsymbol{x}) = \Omega(2^{L-1})$ and $\|F(\boldsymbol{x})\| = \mathcal{O}(2^L)$. Thus the gap in losses between stages can be $\Omega(1)$.*

*Proof of lemma H.5.* We outline the proof for each case.

1. With residual connection & layernorm, the function $F$ computes the intermediate activations $\boldsymbol{y}^{(1)}, \cdots, \boldsymbol{y}^{(L)}$ on an input $\boldsymbol{x}$ as follows.

$$\boldsymbol{y}^{(\ell)} = \left(\boldsymbol{I} + \frac{\boldsymbol{A}}{\left\|\boldsymbol{y}^{(\ell-1)}\right\|}\right) \boldsymbol{y}^{(\ell-1)}.$$

We show that the above method will behave like power method, i.e. $\boldsymbol{y}^{(\ell)}$ will be of magnitude $\Omega(\ell)$ and will be $\varepsilon$-close in angle to the top eigenvector of $\boldsymbol{A}$, denoted as $\boldsymbol{v}_1(\boldsymbol{A})$, provided $\ell \geq \Omega((1/\delta)\log(1/\varepsilon))$.

We give an inductive argument as follows. Consider any layer $\ell$. Say $\theta_\ell$ denotes the angle between $\boldsymbol{y}^{(\ell)}$ and $\boldsymbol{v}_1(\boldsymbol{A})$. Also, say $\Pi^{\perp}_{\boldsymbol{v}1}$ denotes an orthogonal projection to subspace spanned by the rest of the eigenvectors of $\boldsymbol{A}$. Then,

$$
\begin{aligned}
|\tan\theta_{\ell+1}| &= \frac{\left|\langle \boldsymbol{v}_1(\boldsymbol{A}), \boldsymbol{y}^{(\ell+1)}\rangle\right|}{\left\|\Pi^{\perp}_{\boldsymbol{v}1}(\boldsymbol{y}^{(\ell+1)})\right\|_2} \\
&= \frac{\left|\langle \boldsymbol{v}_1(\boldsymbol{A}), \boldsymbol{y}^{(\ell)}\rangle + \langle \boldsymbol{v}_1(\boldsymbol{A}), \boldsymbol{A}\boldsymbol{y}^{(\ell)}\rangle\right|}{\left\|\Pi^{\perp}_{\boldsymbol{v}1}(\boldsymbol{y}^{(\ell)}) + \Pi^{\perp}_{\boldsymbol{v}1}\boldsymbol{A}\boldsymbol{y}^{(\ell)}\right\|_2} \\
&= \frac{2\left|\langle \boldsymbol{v}_1(\boldsymbol{A}), \boldsymbol{y}^{(\ell)}\rangle\right|}{\left\|\Pi^{\perp}_{\boldsymbol{v}1}(\boldsymbol{y}^{(\ell)}) + \Pi^{\perp}_{\boldsymbol{v}1}\boldsymbol{A}\boldsymbol{y}^{(\ell)}\right\|_2} \\
&\leq \frac{2\langle \boldsymbol{v}_1(\boldsymbol{A}), \boldsymbol{y}^{(\ell)}\rangle}{\left\|\Pi^{\perp}_{\boldsymbol{v}1}(\boldsymbol{y}^{(\ell)}) + \lambda_2(\boldsymbol{A})\Pi^{\perp}_{\boldsymbol{v}1}\boldsymbol{y}^{(\ell)}\right\|_2} \\
&= \frac{2}{1 + \lambda_2(\boldsymbol{A})}\left|\tan\theta_\ell\right|.
\end{aligned}
$$

This implies, under $\lambda_2(\boldsymbol{A}) < 1$, $|\tan\theta_\ell|$ decreases with $\ell$. Under the assumption that $\boldsymbol{x}$ isn't orthogonal to $\boldsymbol{v}_1(\boldsymbol{A})$, the above condition simply shows that if $\ell \geq \mathcal{O}((1/\delta)\log(1/\varepsilon))$, than $|\tan\theta_\ell| < \varepsilon$.

Furthermore, once aligned (or closely aligned) to $\boldsymbol{v}_1(\boldsymbol{A})$, the norm of $\boldsymbol{y}^{(\ell)}$ grows linearly. Hence, $\left\|\boldsymbol{y}^{(\ell)}\right\| = \Omega(L)$. Furthermore, for any $\ell$, the gap between $F_{-\ell}$ and $F_\ell$ simply becomes equal to the gap between $L$ and $L-1$ steps of the modified power method, which can be bounded as $\mathcal{O}(1)$.

2. With no residual connection, the function $F$ computes the intermediate activations $\boldsymbol{y}^{(1)}, \cdots, \boldsymbol{y}^{(L)}$ on an input $\boldsymbol{x}$ as follows.

$$
\boldsymbol{y}^{(\ell)} = \frac{\boldsymbol{A}}{\left\|\boldsymbol{y}^{(\ell-1)}\right\|}\boldsymbol{y}^{(\ell-1)}.
$$

From the update, it's trivial to come to the conclusion that $\boldsymbol{y}^{(\ell)}$ will stay unit norm for any layer $\ell$.

This update is exactly power-method, where $\boldsymbol{y}^{(\ell)}$ gets $\varepsilon$-close to the top eigenvector direction of $\boldsymbol{A}$ in $\mathcal{O}((1/\delta)\log(1/\varepsilon))$ steps. The result then follows from bounding the gap between $L$ and $L-1$ steps of power-method.

3. The proof is very similar to the proof of lemma 5.4 (3), where the major difference comes from the blowup in the norms of the intermediate activations.

$\square$

## I    RAPTR MOTIVATION WITH BOOLEAN DATA: THEORY

**Data and labels:**    We use uniformly sampled boolean data $\boldsymbol{x} \sim \mathcal{U}(\{\pm 1\}^d)$. The true label function $f^* : \mathbb{R}^d \to \mathbb{R}$ for each data is given as

$$
f^*(\boldsymbol{x}) = \frac{\sqrt{3}}{2} + \frac{\sqrt{3}}{2}x_1 - x_1 x_2.
$$

**Network to train:**    We train a 2-layer residual network $f_{p_0, \boldsymbol{w}_1, \boldsymbol{w}_2, b_1, b_2} : \mathbb{R}^d \to \mathbb{R}$ comprised of two single neuron residual blocks, where $p_0$ behaves as a position bias that we add to the output

of the first block. On an input $\boldsymbol{x}$, the network computes output $y \in \mathbb{R}$ with intermediate output $y^{(1)}, y^{(2)} \in \mathbb{R}$ as follows.

$$
\begin{aligned}
y^{(1)} &= p_0 + \sin(\langle \boldsymbol{w}_1, \boldsymbol{x} \rangle + b_1), \\
y^{(2)} &= y^{(1)} + \sin(\langle \boldsymbol{w}_2, \boldsymbol{x} \rangle + y^{(1)} + b_2).
\end{aligned}
\tag{4}
$$

Here $y := y^{(2)}$ is returned as the final output. The network is trained with mean squared error, given by

$$
\mathcal{L} = \mathbb{E}_{\boldsymbol{x} \sim \mathcal{U}(\{\pm 1\}^d)} (y - f^*(\boldsymbol{x}))^2.
\tag{5}
$$

We consider population gradients for the sake of clean exposition of the important ideas. Interested readers can modify the proof to work for finite sample regimes.

**Initialization of weights and biases:** The elements of the weights and biases $\boldsymbol{w}_1, \boldsymbol{w}_2, p_0, b_1, b_2$ have been initialized from all 0s.

**RAPTR Initial position bias training:** In the initial bias phase training, we simply train the bias $p_0$, i.e. at each step, we train with the following mean squared loss,

$$
\mathcal{L} = \mathbb{E}_{\boldsymbol{x} \sim \mathcal{U}(\{\pm 1\}^d)} (p_0 - f^*(\boldsymbol{x}))^2.
\tag{6}
$$

**RAPTR First phase training:** In the first phase, both the layers are trained independently. That is, at each step, we pick a random layer $\ell \in \{1, 2\}$ and train with the mean squared loss, given by

$$
\mathcal{L} = \mathbb{E}_{\boldsymbol{x} \sim \mathcal{U}(\{\pm 1\}^d)} (p_0 + \sin(\langle \boldsymbol{w}_\ell, \boldsymbol{x} \rangle + b_\ell) - f^*(\boldsymbol{x}))^2.
\tag{7}
$$

**RAPTR Second phase training:** In the second phase of training, we train the full model (eq. (4)) with the mean squared loss (eq. (5)). For simplicity, we fix the parameters of the first layer and simply train the parameters $\boldsymbol{w}_2, b_2$.

We first recall the theorem corresponding to the stage-wise training of RAPTR.

**Theorem I.1.** *For a small enough learning rate $\eta < \frac{1}{poly(d)}$, under parameter initialization from all 0s, an initial position bias only training, followed by a 2-stage RAPTR on the network $f_{p_0, \boldsymbol{w}_1, \boldsymbol{w}_2, b_1, b_2}$ (eq. (4)) with mean squared error shows the following behavior.*

- *After $\Theta(1/\eta)$ steps of position bias only training and $\Theta(1/\eta)$ steps of first phase, for both layers $\ell \in \{1, 2\}$,*

$$
\left| p_0 + \sin(\langle \boldsymbol{w}_\ell, \boldsymbol{x} \rangle + b_\ell) - \frac{\sqrt{3}}{2} - \frac{\sqrt{3}}{2} x_1 \right| \leq \mathcal{O}(\eta).
$$

- *After $\Theta(1/\eta)$ steps of second phase, the ouput of the second layer is given as*

$$
\left| y^{(2)} - y^{(1)} - (-x_1 x_2) \right| := \left| \sin(\langle \boldsymbol{w}_2, \boldsymbol{x} \rangle + y^{(1)} + p_0 + b_2) - (-x_1 x_2) \right| \leq \mathcal{O}(\eta).
$$

*Hence, after $\Theta(1/\eta)$ steps of initial position bias and 2-phase RAPTR training, loss $\mathcal{L} < \mathcal{O}(\eta)$ (5).*

*Proof.* The proof proceeds by analyzing the behavior of the weights and the biases in the different phases.

- In the initial position bias only training, we show that in $\Theta(1/\eta)$ steps,

$$
\left| p_0 - \sqrt{3}/2 \right| \leq \mathcal{O}(\eta).
$$

- In the first phase, after $\Theta(\eta \log(d/\eta))$ steps, we show in lemma I.4 that for any layer $\ell \in \{1, 2\}$, the weights and the biases satisfy the following conditions.

$$
\begin{aligned}
\left| w_{\ell,1} - \frac{\pi}{3} \right| &\leq \mathcal{O}(\eta), \\
|w_{\ell,j}| &\leq \mathcal{O}(\eta), \text{ for all } 2 \leq j \leq d, \\
|b_\ell| &\leq \mathcal{O}(\eta).
\end{aligned}
$$

Hence, after first phase of training, the network's intermediate outputs $y^{(1)}$ and $y^{(2)}$ (from eq. (4)) are given as

$$y^{(1)} = p_0 + \sin(\langle \boldsymbol{w}_1, \boldsymbol{x} \rangle + b_1) = \frac{\sqrt{3}}{2} + \frac{\sqrt{3}}{2} x_1 + \mathcal{O}(\eta),$$

$$y^{(2)} = y^{(1)} + \sin(\langle \boldsymbol{w}_2, \boldsymbol{x} \rangle + y^{(1)} + p_0 + b_2)$$

$$= \frac{\sqrt{3}}{2} + \frac{\sqrt{3}}{2} x_1 + \sin(\langle \boldsymbol{w}_2, \boldsymbol{x} \rangle + \sqrt{3}/2 x_1 + \sqrt{3}/2 + b_2) + \mathcal{O}(\eta).$$

- In the second phase, after $\Theta(1/\eta)$ steps, we show in lemma I.5 that the weights and the biases $\boldsymbol{w}_2, b_2$ satisfy the following conditions.

$$\left| w_{2,1} + \sqrt{3}/2 - \frac{\pi}{2} \right| \leq \mathcal{O}(\eta),$$

$$\left| w_{2,2} - \frac{\pi}{2} \right| \leq \mathcal{O}(\eta),$$

$$\left| b_2 + p_0 - \frac{\pi}{2} \right| \leq \mathcal{O}(\eta),$$

$$|w_{2,j}| \leq \mathcal{O}(\eta), \text{ for } 3 \leq j \leq d.$$

Thus, at the end of the second phase, the network's output $y := y^{(2)}$ (from eq. (4)) is given as

$$y^{(2)} = y^{(1)} + \sin(\langle \boldsymbol{w}_2, \boldsymbol{x} \rangle + y^{(1)} + p_0 + b_2)$$

$$= \frac{\sqrt{3}}{2} + \frac{\sqrt{3}}{2} x_1 + \sin(\langle \boldsymbol{w}_2, \boldsymbol{x} \rangle + \sqrt{3}/2 x_1 + \sqrt{3}/2 + b_2) + \mathcal{O}(\eta)$$

$$= \frac{\sqrt{3}}{2} + \frac{\sqrt{3}}{2} x_1 + \sin(\frac{\pi}{2} x_1 + \frac{\pi}{2} x_2 + \frac{\pi}{2}) + \mathcal{O}(\eta) = \frac{\sqrt{3}}{2} + \frac{\sqrt{3}}{2} x_1 - x_1 x_2 + \mathcal{O}(\eta).$$

$$\square$$

Furthermore, we can show that each layer in itself isn't expressive enough to represent the true label function.

**Theorem I.2.** *For each layer when trained independently, the best $\boldsymbol{w}_\ell, \boldsymbol{b}_\ell$ parameters will have an $\Omega(1)$ error.*

*Proof.* The proof of the theorem is simple. We argue for the first layer (same argument holds for the second layer). We simply calculate the projection of the function learned by $\boldsymbol{w}_1, \boldsymbol{b}_1$ into the components of the true label function. Suppose the function define dby them is given by $\alpha x_1 + \beta x_1 x_2$

Then, using orthogonality of the basis polynomials $x_1$ and $x_1 x_2$, we have

$$\alpha = \mathbb{E}_{\boldsymbol{x} \in \{\pm 1\}^d} \sin(\langle \boldsymbol{w}_1, \boldsymbol{x} \rangle + b) x_1$$

$$= \mathbb{E}_{\boldsymbol{x} \in \{\pm 1\}^d} \sin(\pi/2 - \langle \boldsymbol{w}_1, \boldsymbol{x} \rangle - b) x_1$$

$$= \sin(b) (\prod_{i \neq 1} \cos w_i) \sin(w_1). \qquad \text{Using corollary I.9}$$

$$\beta = \mathbb{E}_{\boldsymbol{x} \in \{\pm 1\}^d} \sin(\langle \boldsymbol{w}_1, \boldsymbol{x} \rangle + b) x_1 x_2$$

$$= \mathbb{E}_{\boldsymbol{x} \in \{\pm 1\}^d} \sin(\pi/2 - \langle \boldsymbol{w}_1, \boldsymbol{x} \rangle - b) x_1 x_2$$

$$= -\cos(b) (\prod_{i \neq 1,2} \cos w_i) \sin(w_1) \sin(w_2). \qquad \text{Using corollary I.9}$$

Adding/Subtracting these terms, we get

$$\alpha - \beta = \sin(b + w_2)(\prod_{i \neq 1,2} \cos w_i),$$

$$\alpha + \beta = \sin(b - w_2)(\prod_{i \neq 1,2} \cos w_i),$$

$$-\alpha + \beta = \sin(-b + w_2)(\prod_{i \neq 1,2} \cos w_i),$$

$$-\alpha - \beta = \sin(-b - w_2)(\prod_{i \neq 1,2} \cos w_i)$$

Thus, we must be restricted by the set $|\alpha|, |\beta| < 1, |\alpha| + |\beta| < 1$. As the coefficients of the true label function don't lie in this set, the proof follows. $\square$

## I.1 ANALYSIS OF THE DIFFERENT PHASES

Initial position bias only training linearly increases $p_0$ to $\sqrt{3}/2$.

**Lemma I.3.** *[Position bias only training] For any learning rate $\eta > 0$, after $\Theta(1/\eta)$ steps, $p_0$ satisfies*

$$\left| p_0 - \sqrt{3}/2 \right| \leq \mathcal{O}(\eta).$$

The first lemma analyses the behavior of the weights during phase 1.

**Lemma I.4.** *[Phase 1 of RAPTR] For a small enough learning rate $\eta < \frac{1}{poly(d)}$, with position bias $p_0$ at $\sqrt{3}/2 + \mathcal{O}(\eta)$, and under parameter from all 0s, after $\Theta(1/\eta)$ steps, for any layer $\ell \in \{1, 2\}$, the weights and the biases satisfy the following conditions.*

$$\left| w_{\ell,1} - \frac{\pi}{3} \right| \leq \mathcal{O}(\eta),$$
$$|w_{\ell,j}| \leq \mathcal{O}(\eta), \text{ for all } 2 \leq j \leq d,$$
$$|b_\ell| \leq \mathcal{O}(\eta).$$

The second lemma analyses the behavior of the second layer weights in phase 2.

**Lemma I.5.** *[Phase 2 of RAPTR] For a small enough learning rate $\eta < \frac{1}{poly(d)}$, starting from the weights and biases reached by $\Theta(1/\eta)$ steps of position bias only and first phase of RAPTR training (lemmas I.3 and I.4), after $\Theta(1/\eta)$ steps of second phase of RAPTR training, the weights and biases $w_2$ and $b_2$ satisfy the following conditions.*

$$\left| w_{2,1} + \sqrt{3}/2 - \frac{\pi}{2} \right| \leq \mathcal{O}(\eta),$$
$$\left| w_{2,2} - \frac{\pi}{2} \right| \leq \mathcal{O}(\eta),$$
$$\left| b_2 + \sqrt{3}/2 - \frac{\pi}{2} \right| \leq \mathcal{O}(\eta),$$
$$|w_{2,j}| \leq \mathcal{O}(\eta), \text{ for } 3 \leq j \leq d.$$

### I.1.1 PROOF OF LEMMA I.3

**Lemma I.3.** *[Position bias only training] For any learning rate $\eta > 0$, after $\Theta(1/\eta)$ steps, $p_0$ satisfies*

$$\left| p_0 - \sqrt{3}/2 \right| \leq \mathcal{O}(\eta).$$

*Proof.* Under all 0s initialization, the output of both of the layers is 0. Thus, the gradient of $p_0$ is given by

$$\mathbb{E}_{\boldsymbol{x} \sim \{-1,+1\}^d} \left( p_0 - \frac{\sqrt{3}}{2} - \frac{\sqrt{3}}{2} x_1 + x_1 x_2 \right) = p_0 - \frac{\sqrt{3}}{2}.$$

Hence, $p_0$ increases up until it reaches $\sqrt{3}/2 + \mathcal{O}(\eta)$. $\square$

### I.1.2 PROOF OF LEMMA I.4

**Lemma I.4.** *[Phase 1 of* RAPTR*] For a small enough learning rate $\eta < \frac{1}{poly(d)}$, with position bias $p_0$ at $\sqrt{3}/2 + \mathcal{O}(\eta)$, and under parameter from all 0s, after $\Theta(1/\eta)$ steps, for any layer $\ell \in \{1, 2\}$, the weights and the biases satisfy the following conditions.*

$$\left| w_{\ell,1} - \frac{\pi}{3} \right| \leq \mathcal{O}(\eta),$$
$$|w_{\ell,j}| \leq \mathcal{O}(\eta), \text{ for all } 2 \leq j \leq d,$$
$$|b_\ell| \leq \mathcal{O}(\eta).$$

*Proof.* Similar to the proof of lemma I.3, we can show that the position bias $p_0$ stays $\mathcal{O}(\eta)$ close to $\sqrt{3}/2$. Thus, for simplicity of exposition, we simply assume it without formally proving so.

With the first stage of RAPTR, we only keep one of the two layers during training. Hence, both the layer parameters train similarly during training. Without loss of generality, we discuss for a single layer weights $\ell \in \{1, 2\}$.

**General formulation for gradient w.r.t weights:**   The population gradients of $\boldsymbol{w}_\ell$ is given by

$$\mathbb{E}_{\boldsymbol{x} \sim \{-1,+1\}^d} \left( p_0 + \sin(\langle \boldsymbol{w}_\ell, \boldsymbol{x} \rangle + b_\ell) - \frac{\sqrt{3}}{2} - \frac{\sqrt{3}}{2} x_1 + x_1 x_2 \right) \cdot \cos(\langle \boldsymbol{w}_\ell, \boldsymbol{x} \rangle + b_\ell) \cdot \boldsymbol{x}$$

$$= \mathbb{E}_{\boldsymbol{x} \sim \{-1,+1\}^d} \sin(\langle \boldsymbol{w}_\ell, \boldsymbol{x} \rangle + b_\ell) \cdot \cos(\langle \boldsymbol{w}_\ell, \boldsymbol{x} \rangle + b_\ell) \cdot \boldsymbol{x} - \mathbb{E}_{\boldsymbol{x} \sim \{-1,+1\}^d} \left( \frac{\sqrt{3}}{2} x_1 - x_1 x_2 \right) \cdot \cos(\langle \boldsymbol{w}_\ell, \boldsymbol{x} \rangle + b_\ell) \cdot \boldsymbol{x}$$

$$+ \mathbb{E}_{\boldsymbol{x} \sim \{-1,+1\}^d} \left( p_0 - \frac{\sqrt{3}}{2} \right) \cdot \cos(\langle \boldsymbol{w}_\ell, \boldsymbol{x} \rangle + b_\ell) \cdot \boldsymbol{x}.$$

We will argue about the contributions of the three terms separately.

**Lemma I.6** (Contribution of Term 1). *For any coordinate $j \in [d]$,*

$$\mathbb{E}_{\boldsymbol{x} \sim \{-1,+1\}^d} \sin(\langle \boldsymbol{w}_\ell, \boldsymbol{x} \rangle + b_\ell) \cdot \cos(\langle \boldsymbol{w}_\ell, \boldsymbol{x} \rangle + b_\ell) x_j$$

$$= \frac{1}{2} \cos(2b_\ell) \left( \prod_{i=1 \to d, i \neq j} \cos(2w_{\ell,i}) \right) \sin(2w_{\ell,j}).$$

*Proof.* Term 1 in the gradients is given as

$$\mathbb{E}_{\boldsymbol{x} \sim \{-1,+1\}^d} \sin(\langle \boldsymbol{w}_\ell, \boldsymbol{x} \rangle + b_\ell) \cdot \cos(\langle \boldsymbol{w}_\ell, \boldsymbol{x} \rangle + b_\ell) \cdot \boldsymbol{x}$$

$$= \frac{1}{2} \mathbb{E}_{\boldsymbol{x} \sim \{-1,+1\}^d} \sin(2\langle \boldsymbol{w}_\ell, \boldsymbol{x} \rangle + 2b_\ell) \cdot \boldsymbol{x}.$$

Consider a coordinate $j \in [d]$. The gradient concerning the coordinate is given as

$$\frac{1}{2} \mathbb{E}_{\boldsymbol{x} \sim \{-1,+1\}^d} \sin(2\langle \boldsymbol{w}_\ell, \boldsymbol{x} \rangle + 2b_\ell) x_j$$

$$= \frac{1}{2} \mathbb{E}_{\boldsymbol{x} \sim \{-1,+1\}^d} \sin \left( 2 \sum_{i=1 \to d, i \neq j} w_{\ell,i} x_i + 2w_{\ell,j} x_j + 2b_\ell \right) x_j$$

$$= \frac{1}{2} \cos(2b_\ell) \left( \prod_{i=1 \to d, i \neq j} \cos(2w_{\ell,i}) \right) \sin(2w_{\ell,j}),$$

using lemma I.7. □

**Contribution of Term 2:**   Term 2 in the gradients is given as

$$\mathbb{E}_{\boldsymbol{x}\sim\{-1,+1\}^d}\ \left(\frac{\sqrt{3}}{2}x_1 - x_1 x_2\right)\cdot\cos(\langle\boldsymbol{w}_\ell,\boldsymbol{x}\rangle + b_\ell)\cdot\boldsymbol{x}.$$

Consider the coordinate $i = 1$. The gradient concerning the coordinate is given as

$$\mathbb{E}_{\boldsymbol{x}\sim\{-1,+1\}^d}\ \left(\frac{\sqrt{3}}{2}x_1 - x_1 x_2\right)\cdot\cos(\langle\boldsymbol{w}_\ell,\boldsymbol{x}\rangle + b_\ell)x_1$$

$$= \frac{\sqrt{3}}{2}\mathbb{E}_{\boldsymbol{x}\sim\{-1,+1\}^d}\cos(\langle\boldsymbol{w}_\ell,\boldsymbol{x}\rangle + b_\ell) - \mathbb{E}_{\boldsymbol{x}\sim\{-1,+1\}^d}x_2\cos(\langle\boldsymbol{w}_\ell,\boldsymbol{x}\rangle + b_\ell)\quad\text{Replace } x_1^2 \text{ by } 1$$

$$= \frac{\sqrt{3}}{2}\mathbb{E}_{\boldsymbol{x}\sim\{-1,+1\}^d}\cos(\langle\boldsymbol{w}_\ell,\boldsymbol{x}\rangle + b_\ell) + \sin(b_\ell)\left(\prod_{i=1\to d, i\neq 2}\cos(w_{\ell,i})\right)\sin(w_{\ell,2})$$
$$\text{Simplified second term with lemma I.8}$$

$$= \frac{\sqrt{3}}{2}\cos(b_\ell)\left(\prod_{i=1\to d}\cos(w_{\ell,i})\right) + \sin(b_\ell)\left(\prod_{i=1\to d, i\neq 2}\cos(w_{\ell,i})\right)\sin(w_{\ell,2}).$$
$$\text{Simplified first term with corollary I.9}$$

Consider the coordinate $i = 2$. The gradient concerning the coordinate is given as

$$\mathbb{E}_{\boldsymbol{x}\sim\{-1,+1\}^d}\ \left(\frac{\sqrt{3}}{2}x_1 - x_1 x_2\right)\cdot\cos(\langle\boldsymbol{w}_\ell,\boldsymbol{x}\rangle + b_\ell)x_2$$

$$= \frac{\sqrt{3}}{2}\mathbb{E}_{\boldsymbol{x}\sim\{-1,+1\}^d}\cos(\langle\boldsymbol{w}_\ell,\boldsymbol{x}\rangle + b_\ell)x_1 x_2 - \mathbb{E}_{\boldsymbol{x}\sim\{-1,+1\}^d}x_1\cos(\langle\boldsymbol{w}_\ell,\boldsymbol{x}\rangle + b_\ell)$$
$$\text{Replace } x_2^2 \text{ by } 1$$

$$= -\frac{\sqrt{3}}{2}\cos(b_\ell)\left(\prod_{i=1\to d, i\notin\{1,2\}}\cos(w_{\ell,i})\right)\left(\prod_{j\in\{1,2\}}\sin(w_{\ell,j})\right) + \sin(b_\ell)\left(\prod_{i=1\to d, i\neq 1}\cos(w_{\ell,i})\right)\sin(w_{\ell,1}).$$
$$\text{Using corollary I.9}$$

For any other coordinate $t\notin\{1,2\}$, we have

$$\mathbb{E}_{\boldsymbol{x}\sim\{-1,+1\}^d}\ \left(\frac{\sqrt{3}}{2}x_1 - x_1 x_2\right)\cdot\cos(\langle\boldsymbol{w}_\ell,\boldsymbol{x}\rangle + b_\ell)x_t$$

$$= \frac{\sqrt{3}}{2}\mathbb{E}_{\boldsymbol{x}\sim\{-1,+1\}^d}\cos(\langle\boldsymbol{w}_\ell,\boldsymbol{x}\rangle + b_\ell)x_1 x_t - \mathbb{E}_{\boldsymbol{x}\sim\{-1,+1\}^d}x_1 x_2 x_t\cos(\langle\boldsymbol{w}_\ell,\boldsymbol{x}\rangle + b_\ell)$$

$$= -\frac{\sqrt{3}}{2}\cos(b_\ell)\left(\prod_{i=1\to d, i\notin\{1,t\}}\cos(w_{\ell,i})\right)\left(\prod_{j\in\{1,t\}}\sin(w_{\ell,j})\right)$$

$$- \sin(b_\ell)\left(\prod_{i=1\to d, i\notin\{1,2,t\}}\cos(w_{\ell,i})\right)\left(\prod_{j\in\{1,2,t\}}\sin(w_{\ell,j})\right).\qquad\text{Using corollary I.9}$$

**Contribution of Term 3:**   For any coordinate $j\in[d]$,

$$\mathbb{E}_{\boldsymbol{x}\sim\{-1,+1\}^d}\ \left(p_0 - \frac{\sqrt{3}}{2}\right)\cdot\cos(\langle\boldsymbol{w}_\ell,\boldsymbol{x}\rangle + b_\ell)x_j$$

$$= -\left(p_0 - \frac{\sqrt{3}}{2}\right)\sin(b_\ell)\left(\prod_{i=1\to d, i\neq j}\cos(w_{\ell,i})\right)\sin(w_{\ell,j})$$

$$= \mathcal{O}(\eta),$$

where the pre-final step follows from using corollary I.9 and the final step follows since $\left| p_0 - \frac{\sqrt{3}}{2} \right| = \mathcal{O}(\eta)$ from the initial bias training phase.

Thus, the combination of the 3 terms gives

$$\nabla w_{\ell,1} = -\frac{\sqrt{3}}{2} \cos(b_\ell) \left( \prod_{i=1 \to d} \cos(w_{\ell,i}) \right) - \sin(b_\ell) \left( \prod_{i=1 \to d, i \neq 2} \cos(w_{\ell,i}) \right) \sin(w_{\ell,2})$$

$$+ \frac{1}{2} \cos(2b_\ell) \left( \prod_{i=1 \to d, i \neq 1} \cos(2w_{\ell,i}) \right) \sin(2w_{\ell,1}) + \mathcal{O}(\eta) \tag{8}$$

$$\nabla w_{\ell,2} = \frac{\sqrt{3}}{2} \cos(b_\ell) \left( \prod_{i=1 \to d, i \notin \{1,2\}} \cos(w_{\ell,i}) \right) \left( \prod_{j \in \{1,2\}} \sin(w_{\ell,j}) \right) - \sin(b_\ell) \left( \prod_{i=1 \to d, i \neq 1} \cos(w_{\ell,i}) \right) \sin(w_{\ell,1})$$

$$+ \frac{1}{2} \cos(2b_\ell) \left( \prod_{i=1 \to d, i \neq 2} \cos(2w_{\ell,i}) \right) \sin(2w_{\ell,2}) + \mathcal{O}(\eta) \tag{9}$$

For $t \notin \{1, 2\}$,

$$\nabla w_{\ell,t} = \frac{\sqrt{3}}{2} \cos(b_\ell) \left( \prod_{i=1 \to d, i \notin \{1,t\}} \cos(w_{\ell,i}) \right) \left( \prod_{j \in \{1,t\}} \sin(w_{\ell,j}) \right)$$

$$+ \sin(b_\ell) \left( \prod_{i=1 \to d, i \notin \{1,2,t\}} \cos(w_{\ell,i}) \right) \left( \prod_{j \in \{1,2,t\}} \sin(w_{\ell,j}) \right)$$

$$+ \frac{1}{2} \cos(2b_\ell) \left( \prod_{i=1 \to d, i \neq t} \cos(2w_{\ell,i}) \right) \sin(2w_{\ell,t}) + \mathcal{O}(\eta) \tag{10}$$

**General formulation for gradient w.r.t bias:** Following a similar approach as the weights, the population gradients of $b_\ell$ is given by

$$\mathbb{E}_{\boldsymbol{x} \sim \{-1,+1\}^d} \left( \sin(\langle \boldsymbol{w}_\ell, \boldsymbol{x} \rangle + b_\ell) - \frac{\sqrt{3}}{2} - \frac{\sqrt{3}}{2} x_1 + x_1 x_2 \right) \cdot \cos(\langle \boldsymbol{w}_\ell, \boldsymbol{x} \rangle + b_\ell)$$

$$= \mathbb{E}_{\boldsymbol{x} \sim \{-1,+1\}^d} \sin(\langle \boldsymbol{w}_\ell, \boldsymbol{x} \rangle + b_\ell) \cos(\langle \boldsymbol{w}_\ell, \boldsymbol{x} \rangle + b_\ell)$$

$$- \mathbb{E}_{\boldsymbol{x} \sim \{-1,+1\}^d} \left( \frac{\sqrt{3}}{2} x_1 - x_1 x_2 \right) \cos(\langle \boldsymbol{w}_\ell, \boldsymbol{x} \rangle + b_\ell)$$

$$+ \mathbb{E}_{\boldsymbol{x} \sim \{-1,+1\}^d} \left( p_0 - \frac{\sqrt{3}}{2} \right) \cos(\langle \boldsymbol{w}_\ell, \boldsymbol{x} \rangle + b_\ell)$$

Following similar approach as above, the terms can be simplified as

$$\frac{1}{2} \sin(2b_\ell) \left( \prod_{i=1 \to d} \cos(2w_{\ell,i}) \right) \tag{11}$$

$$+ \frac{\sqrt{3}}{2} \sin(b_\ell) \left( \prod_{i=1 \to d, i \neq 1} \cos(w_{\ell,i}) \right) \sin(w_{\ell,1})$$

$$- \cos(b_\ell) \left( \prod_{i=1 \to d, i \notin \{1,2\}} \cos(w_{\ell,i}) \right) \left( \prod_{j \in \{1,2\}} \sin(w_{\ell,j}) \right) + \mathcal{O}(\eta). \tag{12}$$

**Behavior of gradients at initialization:** Since the coordinates of $\boldsymbol{w}_\ell$ and biases $\boldsymbol{b}_\ell$ are initialized from 0s, we have for all $j$,

$$\sin(b_\ell) = 0, \quad \sin(w_{\ell,j}) = 0, \quad \cos(w_{\ell,j}) = 1, \quad \cos(b_\ell) = 1. \tag{13}$$

Furthermore, recall that $p_0 = \sqrt{3}/2 + \mathcal{O}(\eta)$ after the initial bias training phase.

Using the above, we can simplify the gradients from eqs. (8) to (10) and (12)

$$\nabla w_{\ell,1} = -\frac{\sqrt{3}}{2} + \mathcal{O}(\eta), \tag{14}$$

$$\nabla w_{\ell,t} = \mathcal{O}(\eta), \quad t \neq 1 \tag{15}$$

$$\nabla b_\ell = \mathcal{O}(\eta), \tag{16}$$

Hence, we observe three kinds of behavior at initialization.

1. First coordinate of the weight grows by atleast

$$w_{\ell,1} \leftarrow w_{\ell,1} + \frac{\eta\sqrt{3}}{2} + \mathcal{O}(\eta^2).$$

2. Other coordinates of the weight get only $\mathcal{O}(\eta^2)$ updates.

$$w_{\ell,j} \leftarrow w_{\ell,j} + \mathcal{O}(\eta^2).$$

3. The magnitude of the bias drops as well.

$$b_\ell \leftarrow b_\ell + \mathcal{O}(\eta^2).$$

**Beyond Initialization:** Due to increasing magnitude of $w_{\ell,1}$, the behavior of the gradients changes slightly from Equations (14) to (16). However, assuming that the weight coordinates $[2, d]$ and the biases are smaller than $\mathcal{O}(\eta)$, we can still give similar formulations as before using eq. (13) using the following inequalities for all $j \neq 1$.

$$\sin(b_\ell) = \mathcal{O}(\eta), \quad \sin(w_{\ell,j}) = 1 - \mathcal{O}(\eta).$$

We state them directly.

$$\nabla w_{\ell,1} = -\frac{\sqrt{3}}{2}\cos(b_\ell)\left(\prod_{i=1\to d}\cos(w_{\ell,i})\right) - \sin(b_\ell)\left(\prod_{i=1\to d, i\neq 2}\cos(w_{\ell,i})\right)\sin(w_{\ell,2})$$

$$+ \frac{1}{2}\cos(2b_\ell)\left(\prod_{i=1\to d, i\neq 1}\cos(2w_{\ell,i})\right)\sin(2w_{\ell,1}) + \mathcal{O}(\eta)$$

$$= (1 - \mathcal{O}(\eta))^d\left(\sin(w_{\ell,1}) - \frac{\sqrt{3}}{2}\right) + \mathcal{O}(\eta) \leq \frac{1}{2}\left(\sin(w_{\ell,1}) - \frac{\sqrt{3}}{2}\right) + \mathcal{O}(\eta), \tag{17}$$

for $\eta \leq 1/poly(d)$.

$$\nabla w_{\ell,2} = \frac{\sqrt{3}}{2}\cos(b_\ell)\left(\prod_{i=1\to d, i\notin\{1,2\}}\cos(w_{\ell,i})\right)\left(\prod_{j\in\{1,2\}}\sin(w_{\ell,j})\right) - \sin(b_\ell)\left(\prod_{i=1\to d, i\neq 1}\cos(w_{\ell,i})\right)\sin(w_{\ell,1})$$

$$+ \frac{1}{2}\cos(2b_\ell)\left(\prod_{i=1\to d, i\neq 2}\cos(2w_{\ell,i})\right)\sin(2w_{\ell,2}) + \mathcal{O}(\eta)$$

$$= \mathcal{O}(\eta). \tag{18}$$

For any $t \notin \{1, 2\}$,

$$
\begin{aligned}
\nabla w_{\ell,j} = &\frac{\sqrt{3}}{2} \cos(b_\ell) \left( \prod_{i=1 \to d, i \notin \{1,t\}} \cos(w_{\ell,i}) \right) \left( \prod_{j \in \{1,t\}} \sin(w_{\ell,j}) \right) \\
&+ \sin(b_\ell) \left( \prod_{i=1 \to d, i \notin \{1,2,t\}} \cos(w_{\ell,i}) \right) \left( \prod_{j \in \{1,2,t\}} \sin(w_{\ell,j}) \right) \\
&+ \frac{1}{2} \cos(2b_\ell) \left( \prod_{i=1 \to d, i \neq t} \cos(2w_{\ell,i}) \right) \sin(2w_{\ell,t}) + \mathcal{O}(\eta), \\
= &\mathcal{O}(\eta).
\end{aligned}
\tag{19}
$$

And finally for the bias,

$$
\begin{aligned}
\nabla b_\ell = &\frac{1}{2} \sin(2b_\ell) \left( \prod_{i=1 \to d} \cos(2w_{\ell,i}) \right) + \frac{\sqrt{3}}{2} \sin(b_\ell) \left( \prod_{i=1 \to d, i \neq 1} \cos(w_{\ell,i}) \right) \sin(w_{\ell,1}) \\
&- \cos(b_\ell) \left( \prod_{i=1 \to d, i \notin \{1,2\}} \cos(w_{\ell,i}) \right) \left( \prod_{j \in \{1,2\}} \sin(w_{\ell,j}) \right) + \mathcal{O}(\eta) \tag{20} \\
= &\mathcal{O}(\eta). \tag{21}
\end{aligned}
$$

Thus, we observe three properties.

- First coordinate of the weight grows until $\sin(w_{\ell,1}) = \frac{\sqrt{3}}{2}$ or $w_{\ell,1}$ reaches $\pi/3$,

$$
w_{\ell,1} \leftarrow w_{\ell,1} + \frac{\eta}{2} \left( \frac{\sqrt{3}}{2} - \sin(w_{\ell,1}) \right) + \mathcal{O}(\eta^2).
$$

  Thus, in $\Theta(1/\eta)$ steps, $w_{\ell,1}$ can reach arbitrarily close to $\pi/3$.

- Any other coordinate $t \neq 1$ still receives $\mathcal{O}(\eta^2)$ updates as

$$
w_{\ell,t} \leftarrow w_{\ell,t} + \mathcal{O}(\eta^2).
$$

  Hence, in $\Theta(1/\eta)$ steps, $w_{\ell,t}$ can only reach $\mathcal{O}(\eta)$ magnitude.

- $b_\ell$ also receives $\mathcal{O}(\eta^2)$ updates as

$$
b_\ell \leftarrow b_\ell + \mathcal{O}(\eta^2).
$$

  Hence, in $\Theta(1/\eta)$ steps, $b_\ell$ can only reach $\mathcal{O}(\eta)$ magnitude.

□

### I.1.3 PROOF OF LEMMA I.5

**Lemma I.5.** *[Phase 2 of* RAPTR*] For a small enough learning rate $\eta < \frac{1}{poly(d)}$, starting from the weights and biases reached by $\Theta(1/\eta)$ steps of position bias only and first phase of* RAPTR *training (lemmas I.3 and I.4), after $\Theta(1/\eta)$ steps of second phase of* RAPTR *training, the weights and biases $\mathbf{w}_2$ and $b_2$ satisfy the following conditions.*

$$
\begin{aligned}
\left| w_{2,1} + \sqrt{3}/2 - \frac{\pi}{2} \right| &\leq \mathcal{O}(\eta), \\
\left| w_{2,2} - \frac{\pi}{2} \right| &\leq \mathcal{O}(\eta), \\
\left| b_2 + \sqrt{3}/2 - \frac{\pi}{2} \right| &\leq \mathcal{O}(\eta), \\
|w_{2,j}| &\leq \mathcal{O}(\eta), \text{ for } 3 \leq j \leq d.
\end{aligned}
$$

*Proof.* The proof follows along similar lines as lemma I.4.

**First layer output:** At the end of the first phase training, the weights $\boldsymbol{w}_1, \boldsymbol{w}_2$ and the biases look as follows. For $\ell \in \{1, 2\}$,

$$\left| w_{\ell,1} - \frac{\pi}{3} \right| \leq \mathcal{O}(\eta), |w_{\ell,j}| \leq \mathcal{O}(\eta), \text{ for all } j \geq 2, |b_\ell| \leq \mathcal{O}(\eta),$$

and $\left| p_0 - \frac{\sqrt{3}}{2} \right| \leq \mathcal{O}(\eta)$.

Hence, using a Taylor expansion, the output of the first layer is given as (for simplicity, we denote it as $o^{(1)}$)

$$o^{(1)} := \sin(\langle \boldsymbol{w}_1, \boldsymbol{x} \rangle + b_1) = \frac{\sqrt{3}}{2} x_1 + \mathcal{O}(\eta). \tag{22}$$

**Second layer output:** The output of the second layer is now given by (for simplicity, we denote it as $o^{(2)}$)

$$\begin{aligned}
o^{(2)} &:= \sin\left(\langle \boldsymbol{w}_2, \boldsymbol{x} \rangle + o^{(1)} + p_0 + b_2\right) \\
&= \sin\left(\langle \boldsymbol{w}_2, \boldsymbol{x} \rangle + \sin(\langle \boldsymbol{w}_1, \boldsymbol{x} \rangle + b_1) + p_0 + b_2\right) \\
&= \sin\left(\langle \boldsymbol{w}_2, \boldsymbol{x} \rangle + \left(\frac{\sqrt{3}}{2} x_1 + \mathcal{O}(\eta)\right) + (\sqrt{3}/2 + \mathcal{O}(\eta)) + b_2\right) \\
&= \sin\left(\langle \boldsymbol{w}_2, \boldsymbol{x} \rangle + \frac{\sqrt{3}}{2} x_1 + \sqrt{3}/2 + b_2\right) + \mathcal{O}(\eta) \qquad \text{With Taylor expansion} \\
&= \sin\left((w_{2,1} + \sqrt{3}/2) x_1 + \sum_{j=2}^{d} w_{2,j} x_j + \sqrt{3}/2 + b_2\right) + \mathcal{O}(\eta).
\end{aligned}$$

For brevity, we will use new notations to represent the coefficients of $x_1, \cdots, x_d$ in the above formulation.

$$\begin{aligned}
\tilde{w}_1 &= w_{2,1} + \sqrt{3}/2, \\
\tilde{w}_j &= w_{2,j}, \text{ for } j \geq 2, \\
\tilde{b} &= \sqrt{3}/2 + b_2.
\end{aligned}$$

At initialization, their corresponding values are

$$\begin{aligned}
\tilde{w}_1 &= \frac{\pi}{3} + \sqrt{3}/2 + \mathcal{O}(\eta), \\
\tilde{w}_j &= \mathcal{O}(\eta), \text{ for all } j \geq 2 \\
\tilde{b} &= \frac{\sqrt{3}}{2} + \mathcal{O}(\eta). \tag{23}
\end{aligned}$$

Then, the above formulation is given as

$$o^{(2)} = \sin\left(\langle \tilde{\boldsymbol{w}}, \boldsymbol{x} \rangle + \tilde{b}\right) + \mathcal{O}(\eta). \tag{24}$$

**General formulation for population gradients:** The gradients w.r.t. the weight $\boldsymbol{w}_2$ and $b_2$ are given as

$$\nabla \boldsymbol{w}_2 = \mathbb{E}_{\boldsymbol{x} \sim \{-1,+1\}^d} \left(o^{(1)} + o^{(2)} - \frac{\sqrt{3}}{2} x_1 + x_1 x_2\right) \cos\left(\langle \tilde{\boldsymbol{w}}, \boldsymbol{x} \rangle + \tilde{b}\right) \boldsymbol{x} + \mathcal{O}(\eta).$$

$$\nabla b_2 = \mathbb{E}_{\boldsymbol{x} \sim \{-1,+1\}^d} \left(o^{(1)} + o^{(2)} - \frac{\sqrt{3}}{2} x_1 + x_1 x_2\right) \cos\left(\langle \tilde{\boldsymbol{w}}, \boldsymbol{x} \rangle + \tilde{b}\right) + \mathcal{O}(\eta).$$

We will consider the first two terms in the above gradient formulations and add the error term later. First of all, we observe that

$$o^{(1)} + o^{(2)} - \frac{\sqrt{3}}{2} x_1 + x_1 x_2$$
$$= \sin\left(\langle \tilde{\boldsymbol{w}}, \boldsymbol{x}\rangle + \tilde{b}\right) + x_1 x_2 + \mathcal{O}(\eta).$$

We have the gradients w.r.t. $\boldsymbol{w}_2$ as

$$\nabla \boldsymbol{w}_2 = \mathbb{E}_{\boldsymbol{x}\sim\{-1,+1\}^d} \sin\left(\langle \tilde{\boldsymbol{w}}, \boldsymbol{x}\rangle + \tilde{b}\right) \cos\left(\langle \tilde{\boldsymbol{w}}, \boldsymbol{x}\rangle + \tilde{b}\right) \boldsymbol{x}$$
$$+ \mathbb{E}_{\boldsymbol{x}\sim\{-1,+1\}^d} x_1 x_2 \cos\left(\langle \tilde{\boldsymbol{w}}, \boldsymbol{x}\rangle + \tilde{b}\right) \boldsymbol{x}$$
$$+ \mathcal{O}(\eta).$$

We treat the two terms separately.

CONTRIBUTION OF TERM 1 : Using a similar strategy as lemma I.6, for any coordinate $t \in [d]$,

$$\mathbb{E}_{\boldsymbol{x}\sim\{-1,+1\}^d} \sin\left(\langle \tilde{\boldsymbol{w}}, \boldsymbol{x}\rangle + \tilde{b}\right) \cos\left(\langle \tilde{\boldsymbol{w}}, \boldsymbol{x}\rangle + \tilde{b}\right) x_t$$
$$= \frac{1}{2} \cos(2\tilde{b}) \left(\prod_{i=1\to d, i\neq t} \cos(2\tilde{w}_i)\right) \sin(2\tilde{w}_t).$$

CONTRIBUTION OF TERM 2: For coordinate $t = 2$,

$$\mathbb{E}_{\boldsymbol{x}\sim\{-1,+1\}^d} x_1 x_2 \cos\left(\langle \tilde{\boldsymbol{w}}, \boldsymbol{x}\rangle + \tilde{b}\right) x_2$$
$$= \mathbb{E}_{\boldsymbol{x}\sim\{-1,+1\}^d} \cos\left(\langle \tilde{\boldsymbol{w}}, \boldsymbol{x}\rangle + \tilde{b}\right) x_1 \qquad\qquad \text{Replace } x_2^2 \text{ by } 1$$
$$= -\sin(\tilde{b}) \left(\prod_{i=1\to d, i\neq 1} \cos(\tilde{w}_i)\right) \sin(\tilde{w}_1). \qquad\qquad \text{Using corollary I.9}$$

For coordinate $t = 1$,

$$\mathbb{E}_{\boldsymbol{x}\sim\{-1,+1\}^d} x_1 x_2 \cos\left(\langle \tilde{\boldsymbol{w}}, \boldsymbol{x}\rangle + \tilde{b}\right) x_1$$
$$= \mathbb{E}_{\boldsymbol{x}\sim\{-1,+1\}^d} \cos\left(\langle \tilde{\boldsymbol{w}}, \boldsymbol{x}\rangle + \tilde{b}\right) x_2 \qquad\qquad \text{Replace } x_1^2 \text{ by } 1$$
$$= -\sin(\tilde{b}) \left(\prod_{i=1\to d, i\neq 2} \cos(\tilde{w}_i)\right) \sin(\tilde{w}_2). \qquad\qquad \text{Using corollary I.9}$$

For any other coordinate $t \notin \{1, 2\}$,

$$\mathbb{E}_{\boldsymbol{x}\sim\{-1,+1\}^d} \cos\left(\langle \tilde{\boldsymbol{w}}, \boldsymbol{x}\rangle + \tilde{b}\right) x_1 x_2 x_t$$
$$= \sin(\tilde{b}) \left(\prod_{i=1\to d, i\notin\{1,2,t\}} \cos(\tilde{w}_i)\right) \left(\prod_{i\notin\{1,2,t\}} \sin(\tilde{w}_t)\right). \qquad\qquad \text{Using corollary I.9}$$

**Combination of all terms:** For coordinate $t = 1$,

$$\nabla w_{2,1} = \frac{1}{2} \cos(2\tilde{b}) \left(\prod_{i=1\to d, i\neq 1} \cos(2\tilde{w}_i)\right) \sin(2\tilde{w}_1) - \sin(\tilde{b}) \left(\prod_{i=1\to d, i\neq 2} \cos(\tilde{w}_i)\right) \sin(\tilde{w}_2) + \mathcal{O}(\eta).$$

$$(25)$$

For coordinate $t = 2$,

$$\nabla w_{2,2} = \frac{1}{2}\cos(2\tilde{b})\left(\prod_{i=1\to d, i\neq 2}\cos(2\tilde{w}_i)\right)\sin(2\tilde{w}_2) - \sin(\tilde{b})\left(\prod_{i=1\to d, i\neq 1}\cos(\tilde{w}_i)\right)\sin(\tilde{w}_1) + \mathcal{O}(\eta).$$

(26)

For any other coordinate $t \notin \{1, 2\}$,

$$\nabla w_{2,t} = \frac{1}{2}\cos(2\tilde{b})\left(\prod_{i=1\to d, i\neq t}\cos(2\tilde{w}_i)\right)\sin(2\tilde{w}_t) - \sin(\tilde{b})\left(\prod_{i=1\to d, i\notin\{1,2,t\}}\cos(\tilde{w}_i)\right)\left(\prod_{i\in\{1,2,t\}}\sin(\tilde{w}_i)\right) + \mathcal{O}(\eta).$$

(27)

We have similar computation for the gradient of the bias $b_2$, which we report directly.

$$\nabla b_2 = \mathbb{E}_{\boldsymbol{x}\sim\{-1,+1\}^d}\left(o^{(1)} + o^{(2)} - \frac{\sqrt{3}}{2}x_1 + x_1 x_2\right)\cos\left(\langle\tilde{\boldsymbol{w}}, \boldsymbol{x}\rangle + \tilde{b}\right) + \mathcal{O}(\eta)$$

$$= \frac{1}{2}\sin(2\tilde{b})\left(\prod_{i=1\to d}\cos(2\tilde{w}_i)\right) - \cos(\tilde{b})\left(\prod_{i=1\to d, i\notin\{1,2\}}\cos(\tilde{w}_i)\right)\sin(\tilde{w}_1)\sin(\tilde{w}_2) + \mathcal{O}(\eta).$$

(28)

AT INITIALIZATION: With $\tilde{b}$ being initialzied at $\frac{\sqrt{3}}{2} + \mathcal{O}(\eta)$ (from eq. (23)),

$$\cos(\tilde{b}) = \cos(\sqrt{3}/2) + \mathcal{O}(\eta) > 0$$
$$\cos(2\tilde{b}) = \cos(\sqrt{3}) + \mathcal{O}(\eta) < 0$$
$$\sin(\tilde{b}) = \sin(\sqrt{3}/2) + \mathcal{O}(\eta) > 0.$$
$$\sin(2\tilde{b}) = \sin(\sqrt{3}) + \mathcal{O}(\eta) > 0.$$

The initial value of $\tilde{w}_1$ is $\frac{\sqrt{3}}{2} + \frac{\pi}{3} + \mathcal{O}(\eta)$ (from eq. (23)) and hence the values of

$$\cos(2\tilde{w}_1) = \cos(\sqrt{3} + 2\pi/3) + \mathcal{O}(\eta) < 0$$
$$\sin(2\tilde{w}_1) = \sin(\sqrt{3} + 2\pi/3) + \mathcal{O}(\eta) < 0$$
$$\cos(\tilde{w}_1) = \cos(\sqrt{3}/2 + \pi/3) + \mathcal{O}(\eta) < 0$$
$$\sin(\tilde{w}_1) = \sin(\sqrt{3}/2 + \pi/3) + \mathcal{O}(\eta) > 0.$$

The other coordinates $\tilde{w}_j$ are of order $\mathcal{O}(\eta)$. Thus, the behavior of the weights and biases at initialization can be summarized as follows.

- $w_{2,1}$ gets a decreasing update as both the terms involved in eq. (25) are positive.

$$-\eta\nabla w_{2,1} = -\eta\left|\frac{1}{2}\cos(2\tilde{b})\left(\prod_{i=1\to d, i\neq 1}\cos(2\tilde{w}_i)\right)\sin(2\tilde{w}_1)\right| - \eta\left|\sin(\tilde{b})\left(\prod_{i=1\to d, i\neq 2}\cos(\tilde{w}_i)\right)\sin(\tilde{w}_2)\right| + \mathcal{O}(\eta^2)$$

$$= -\frac{\eta}{2}\left|\cos(\sqrt{3})\right|(1 - \mathcal{O}(2\eta))^{d-1}\left|\sin(\sqrt{3} + 2\pi/3)\right| + \mathcal{O}(\eta^2) \leq -\frac{\eta}{4}\cos(\sqrt{3})\sin(\sqrt{3} + 2\pi/3) + \mathcal{O}(\eta^2),$$

as $\eta = \mathcal{O}(1/poly(d))$ small.

- $w_{2,2}$ gets an increasing update as both the terms involved in eq. (26) are negative.

$$-\eta\nabla w_{2,2} = \frac{\eta}{2}\left|\cos(2\tilde{b})\right|\left|\left(\prod_{i=1\to d, i\neq 2}\cos(2\tilde{w}_i)\right)\right||\sin(2\tilde{w}_2)| + \left|\sin(\tilde{b})\right|\left|\left(\prod_{i=1\to d, i\neq 1}\cos(\tilde{w}_i)\right)\right||\sin(\tilde{w}_1)| + \mathcal{O}(\eta^2)$$

$$= \eta\sin(\sqrt{3})\sin(\sqrt{3}/2 + \pi/3)(1 - \mathcal{O}(\eta))^{d-1} + \mathcal{O}(\eta^2) \geq \frac{\eta}{2}\sin(\sqrt{3})\sin(\sqrt{3}/2 + \pi/3) + \mathcal{O}(\eta^2),$$

as $\eta = \mathcal{O}(1/poly(d))$ small.

- Other coordinates $w_{2,t}$ for $t \notin \{1, 2\}$ get small gradients as both the involved terms in eq. (27) depend on how large $\tilde{w}_t := w_{2,t} = \mathcal{O}(\eta)$ is.

$$-\eta \nabla w_{2,t} = -\frac{\eta}{2} \cos(2\tilde{b}) \left( \prod_{i=1\to d, i\neq t} \cos(2\tilde{w}_i) \right) \sin(2\tilde{w}_t)$$

$$+ \eta \sin(\tilde{b}) \left( \prod_{i=1\to d, i\notin\{1,2,t\}} \cos(\tilde{w}_i) \right) \left( \prod_{i\in\{1,2,t\}} \sin(\tilde{w}_i) \right) + \mathcal{O}(\eta^2) = \mathcal{O}(\eta^2).$$

- Bias $b_2$ get an increasing update as both the involved terms in eq. (28) are negative.

$$-\eta \nabla b_2 = \frac{\eta}{2} \left| \sin(2\tilde{b}) \right| \left| \left( \prod_{i=1\to d} \cos(2\tilde{w}_i) \right) \right| + \eta \left| \cos(\tilde{b}) \right| \left| \left( \prod_{i=1\to d, i\notin\{1,2\}} \cos(\tilde{w}_i) \right) \sin(\tilde{w}_1) \right| \left| \sin(\tilde{w}_2) \right| + \mathcal{O}(\eta)$$

$$= \frac{\eta}{2} \sin(\sqrt{3}) \left| \cos(\sqrt{3} + 2\pi/3) \right| (1 - \mathcal{O}(\eta))^{d-1} + \mathcal{O}(\eta) \geq \frac{\eta}{4} \sin(\sqrt{3}) \left| \cos(\sqrt{3} + 2\pi/3) \right|.$$

BEYOND INITIALIZATION: In fact, we can extend the above argument beyond initialization, where $w_{2,1}$ decreases and $w_{2,2}, b_2$ increase by $\Theta(\eta)$ updates, up until we reach the conditions

$$\left| \tilde{w}_1 - \frac{\pi}{2} \right| \leq \mathcal{O}(\eta), \quad \text{(or)} \quad \left| w_{2,1} - \frac{\pi}{2} + \sqrt{3}/2 \right| \leq \mathcal{O}(\eta),$$

$$\left| \tilde{w}_2 - \frac{\pi}{2} \right| \leq \mathcal{O}(\eta), \quad \text{(or)} \quad \left| w_{2,2} - \frac{\pi}{2} \right| \leq \mathcal{O}(\eta)$$

$$\left| \tilde{b} - \frac{\pi}{2} \right| \leq \mathcal{O}(\eta), \quad \text{(or)} \quad \left| b_2 - \frac{\pi}{2} + \sqrt{3}/2 \right| \leq \mathcal{O}(\eta).$$

Since, we get $\Theta(\eta)$ updates in each step, this condition can be reached in $\Theta(1/\eta)$ steps.

The argument is as follows. For any $\tilde{w}_1, \tilde{w}_2, \tilde{b}$ satisfying the following ranges,

$$\tilde{w}_1 \in (\frac{\pi}{2}, \frac{\sqrt{3}}{2} + \frac{\pi}{3}),$$

$$\tilde{w}_2 \in (0, \frac{\pi}{2}),$$

$$\tilde{b} \in (\sqrt{3}/2, \frac{\pi}{2}),$$

the following conditions hold true.

$$\cos(\tilde{b}), \ \sin(\tilde{b}), \ \sin(2\tilde{b}) > 0, \quad \cos(2\tilde{b}) < 0$$
$$\cos(2\tilde{w}_1), \ \sin(2\tilde{w}_1), \cos(\tilde{w}_1) < 0, \quad \sin(\tilde{w}_1) > 0.$$
$$\sin(\tilde{w}_2), \ \cos(\tilde{w}_2), \ \sin(2\tilde{w}_2) > 0, \quad \cos(2\tilde{w}_2) < 0.$$

We can then show that the terms involved in the update rule of $w_{2,1}$ (eq. (25)) are positive, implying $-\eta \nabla w_{2,1} < -\Omega(\eta)$ up until $\tilde{w}_1$ reaches $\mathcal{O}(\eta)$ close to $\frac{\pi}{2}$. Similar argument can be given for the updates of $b_2$ and $w_{2,2}$ respectively.

Furthermore, the coordinates $w_{2,t}$ for $t \notin \{1, 2\}$ get $\mathcal{O}(\eta^2)$ updates per step and stay $\mathcal{O}(\eta)$ small for $\Theta(1/\eta)$ steps of training.

$\square$

## I.2 USEFUL LEMMAS

**Lemma I.7.**

$$\mathbb{E}_{\boldsymbol{x}\sim\{-1,+1\}^d} \ \sin\left(\langle \boldsymbol{w}, \boldsymbol{x} \rangle + b\right) x_j = \cos(b) \left( \prod_{i=1\to d, i\neq j} \cos(w_i) \right) \sin(w_j).$$

*Proof.*

$$\mathbb{E}_{\boldsymbol{x} \sim \{-1,+1\}^d} \sin\left(\langle \boldsymbol{w}, \boldsymbol{x} \rangle + b\right) x_j$$

$$= \mathbb{E}_{\boldsymbol{x} \sim \{-1,+1\}^d} \sin\left(\sum_{i=1 \to d, i \neq j} w_i x_i + w_j x_j + b\right) x_j$$

$$= \cos(b)\left(\mathbb{E}_{\boldsymbol{x} \sim \{-1,+1\}^d} \sin\left(\sum_{i=1 \to d, i \neq j} w_i x_i\right) \cdot \mathbb{E}_{x_j} \cos\left(w_j x_j\right) x_j\right)$$
$$\text{Equals 0 as an odd function}$$

$$+ \cos(b)\left(\mathbb{E}_{\boldsymbol{x} \sim \{-1,+1\}^d} \cos\left(\sum_{i=1 \to d, i \neq j} w_i x_i\right) \cdot \mathbb{E}_{x_j} \sin\left(w_j x_j\right) x_j\right)$$
$$\text{Equals 0 as an odd function}$$

$$+ \sin(b)\left(\mathbb{E}_{\boldsymbol{x} \sim \{-1,+1\}^d} \cos\left(\sum_{i=1 \to d, i \neq j} w_i x_i\right) \cdot \mathbb{E}_{x_j} \cos\left(w_j x_j\right) x_j\right)$$

$$- \sin(b)\left(\mathbb{E}_{\boldsymbol{x} \sim \{-1,+1\}^d} \sin\left(\sum_{i=1 \to d, i \neq j} w_i x_i\right) \cdot \mathbb{E}_{x_j} \sin\left(w_j x_j\right) x_j\right)$$
$$\text{Equals 0 as an odd function}$$

$$= \cos(b)\left(\mathbb{E}_{\boldsymbol{x} \sim \{-1,+1\}^d} \cos\left(\sum_{i=1 \to d, i \neq j} w_i x_i\right) \cdot \mathbb{E}_{x_j} \sin\left(w_j x_j\right) x_j\right)$$

$$= \cos(b)\left(\prod_{i=1 \to d, i \neq j} \cos(w_i)\right) \sin(w_j)$$

$$\square$$

**Lemma I.8.**

$$\mathbb{E}_{\boldsymbol{x} \sim \{-1,+1\}^d} \cos\left(\langle \boldsymbol{w}, \boldsymbol{x} \rangle + b\right) x_j = -\sin(b)\left(\prod_{i=1 \to d, i \neq j} \cos(w_i)\right) \sin(w_j).$$

*Proof.*

$$\mathbb{E}_{\boldsymbol{x} \sim \{-1,+1\}^d} \cos\left(\langle \boldsymbol{w}, \boldsymbol{x} \rangle + b\right) x_j$$

$$= \mathbb{E}_{\boldsymbol{x} \sim \{-1,+1\}^d} \cos\left(\frac{\pi}{2} - \langle \boldsymbol{w}, \boldsymbol{x} \rangle - b\right) x_j$$

$$= \cos(\pi/2 - b)\left(\prod_{i=1 \to d, i \neq j} \cos(-w_i)\right) \sin(-w_j) \qquad \text{Using lemma I.7}$$

$$= -\sin(b)\left(\prod_{i=1 \to d, i \neq j} \cos(w_i)\right) \sin(w_j).$$

$$\square$$

**Corollary I.9.** *Consider a set $S \subseteq [d]$.*

$$\mathbb{E}_{\boldsymbol{x} \sim \{-1,+1\}^d} \cos\left(\langle \boldsymbol{w}, \boldsymbol{x} \rangle + b\right) \prod_{j \in S} x_i = c_b \left(\prod_{i=1 \to d, i \notin S} \cos(w_i)\right)\left(\prod_{j \in S} \sin(w_j)\right),$$

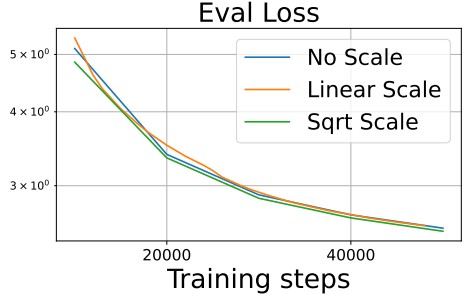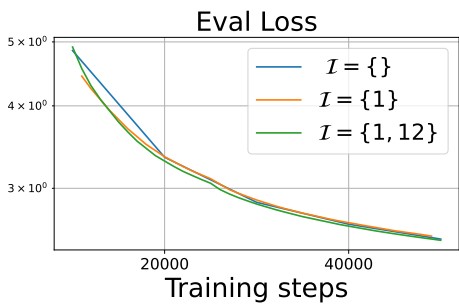

Figure 7: Ablation study on choices for RAPTR algorithm. We train BERT-base with RAPTR for 100k steps with a 6-8-10-12 schedule (see Section 4). Left to right: (a) Square-root scaling ($h_{sqrt}$) has better validation loss compared with linear scaling and no scaling at training, especially in the earlier stages. With *no scale*, we scale layer's output by $1/p_i$ during inference (following Huang et al. (2016)). (b) Different candidates for the fixed set $\mathcal{I}$ are compared for RAPTR. We find that training with first and last layers fixed helps faster training.

*where*

$$c_b = \begin{cases} -\sin(b), & \textit{if } |S| = 4t + 1 \textit{ for some } t \in \mathbb{N} \\ -\cos(b), & \textit{if } |S| = 4t + 2 \textit{ for some } t \in \mathbb{N} \\ \sin(b), & \textit{if } |S| = 4t + 3 \textit{ for some } t \in \mathbb{N} \\ \cos(b), & \textit{if } |S| = 4t \textit{ for some } t \in \mathbb{N}. \end{cases}$$

## J    SCALING

While working on subnetworks, it is important to appropriately rescale the network. In particular, bypassing layer(s) in RAPTR can shift the input distribution for layers compared to the full model. To mitigate this shift, we scale the output of the layers to maintain the norms of the input to each layer throughout training. The idea of scaling layer outputs has also been explored in prior work Huang et al. (2016); Fan et al. (2019).We use a different square-root scaling mechanism that is motivated by analysis on a randomly initialized Transformer architecture. We show that at initialization, the norm of the intermediate outputs $\boldsymbol{y}^{(\ell)}$ scale as $\sqrt{\ell}$.

**Theorem J.1** (Square-root scaling of input norms)**.** *At initialization, given any input sequence $\boldsymbol{x}_{1:N}$ and scalars $\alpha_{1:L}$, w.h.p. the intermediate sequences of $F$ (definition 2.1) satisfy*

$$\left\| \boldsymbol{y}_i^{(\ell)} \right\|_2^2 = \|\boldsymbol{x}_i\|_2^2 + \Theta(\sum_{j=1}^{\ell} \alpha_j^2 d), \quad \textit{for all } 1 \le i \le N, 1 \le \ell \le L.$$

Inspired by this, we define the function $h_{sqrt}$ that takes in a sequence of binary values $\zeta_{1:L} \in \{0,1\}^L$ and returns scalars $\overline{\zeta}_{1:L}$ where $\overline{\zeta}_i$ is the scaling for output of layer $i$. We find a scaling that satisfies the following two conditions hold (a) $\sum_{i=1}^{j} \overline{\zeta}_i^2 = j$ for all $j \le L$ (maintaining the norm), and (b) $\overline{\zeta}_i = 0$ if $\zeta_i = 0$ for all $i \le L$ (maintaining layers in random subnetwork). Formally, for each index $j$ with $\zeta_j = 1$, it finds the index minimum $\bar{j} > j$ with $\zeta_{\bar{j}} = 1$ and sets $\overline{\zeta}_j = \sqrt{\bar{j} - j}$. The RAPTR algorithm with square root scaling is presented in algorithm 1. In Figure 7 we compare square root scaling with some other scaling methods and find it to be slightly better.

### J.1    PROOFS

For simplicity, we present the results for transformers with single head in the self-attention layer. Furthermore, for simplicity, we use $\sigma_{relu}$ activation in the MLPs.

**Definition J.2.** [Layer Normalization] Define a normalization function $f : \mathbb{R}^d \to \mathbb{R}^d$ that performs $f(\boldsymbol{x}) = (\boldsymbol{x} - \mu)/\sigma$, where $\mu$ and $\sigma$ are the mean and standard deviation of $\boldsymbol{x}$, respectively. Then,

---

**Algorithm 3:** Transformer Layer

---

**Require:** 2 layer normalization layers $f_{attn}^{LN}, f_{mlp}^{LN}$ (J.2), an MLP layer $f^{mlp}$ (J.4), and a softmax self-attention layer $f^{attn}$ (J.3), input sequence $\{\boldsymbol{x}_n \in \mathbb{R}^d\}_{n=1}^N$.
1: Attention Layer normalization: returns $\{\boldsymbol{y}_n^{attnln}\}_{n=1}^N$ with $\boldsymbol{y}_n^{attnln} = f_{attn}^{LN}(\boldsymbol{x}_n)$ for all $n \leq N$.
2: Softmax self-attention: returns $\{\boldsymbol{y}_n^{attn}\}_{n=1}^N = f^{attn}(\{\boldsymbol{y}_n^{attnln}\}_{n=1}^N)$.
3: Residual connection: returns $\{\boldsymbol{y}_n^{attnblock}\}_{n=1}^N$, with $\boldsymbol{y}_n^{attnblock} = \boldsymbol{x}_n + \boldsymbol{y}_n^{attn}$ for all $n \leq N$.
4: MLP Layer normalization: returns $\boldsymbol{y}_n^{mlpln} = f_{mlp}^{LN}(\boldsymbol{y}_n^{attnblock})$ for all $n \leq N$.
5: MLP function: returns $\{\boldsymbol{y}_n^{mlp}\}_{n=1}^N$ with $\boldsymbol{y}_n^{mlp} = f^{mlp}(\boldsymbol{y}_n^{mlpln})$ for all $n \leq N$.
6: Compute $\boldsymbol{y}_n = \boldsymbol{y}_n^{mlp} + \boldsymbol{y}_n^{attnblock}$ for all $n \leq N$.
7: Return $\{\boldsymbol{y}_n\}_{n=1}^N$

---

layer normalization $f^{LN} : \mathbb{R}^d \to \mathbb{R}^d$ with parameters $\gamma, \boldsymbol{b} \in \mathbb{R}^d$ takes as input $\boldsymbol{x} \in \mathbb{R}^d$ and outputs $\boldsymbol{y} \in \mathbb{R}^d$, which is computed as $\boldsymbol{z} = f(\boldsymbol{x}), \boldsymbol{y} = \gamma \odot \boldsymbol{z} + \boldsymbol{b}$.

**Definition J.3** (Softmax self-attention). A self-attention layer $f^{attn} : \mathbb{R}^{N \times d} \to \mathbb{R}^{N \times d}$ with parameters $\{\boldsymbol{W}_Q, \boldsymbol{W}_K, \boldsymbol{W}_V, \boldsymbol{C}^{attn} \in \mathbb{R}^{d \times d}\}$ takes a sequence $\{\boldsymbol{x}_n\}_{n \leq N}$ and outputs a sequence $\{\boldsymbol{y}_n\}_{n \leq N}$, such that

$$\boldsymbol{y}_n = \boldsymbol{C}^{attn} \sum_{\overline{n}=1}^N a_{n,\overline{n}} \boldsymbol{v}_{\overline{n}},$$

$$\text{with } a_{n,\overline{n}} = \text{softmax}(\boldsymbol{K}\boldsymbol{q}_n)_{\overline{n}}, \quad \boldsymbol{q}_n = \boldsymbol{W}_Q \boldsymbol{x}_n, \quad \boldsymbol{k}_n = \boldsymbol{W}_K \boldsymbol{x}_n, \quad \boldsymbol{v}_n = \boldsymbol{W}_V \boldsymbol{x}_n,$$

for all $n \leq N$, and $\boldsymbol{K} \in \mathbb{R}^{N \times N}$ defined with rows $\{\boldsymbol{k}_n\}_{n=1}^N$.

**Definition J.4** (MLP). An MLP layer $f^{mlp} : \mathbb{R}^d \to \mathbb{R}^d$ with parameters $\{\boldsymbol{W} \in \mathbb{R}^{m \times d}, \boldsymbol{C}^{mlp} \in \mathbb{R}^{m \times d}\}$ and activation $\sigma_{relu}$, takes an input $\boldsymbol{x} \in \mathbb{R}^d$ and outputs $\boldsymbol{y} \in \mathbb{R}^d$, such that

$$\boldsymbol{y} = \boldsymbol{C}^{mlp} \sigma_{relu}(\boldsymbol{W}\boldsymbol{x}).$$

**Definition J.5** (Transformer layer). A pre-layernorm transformer layer with three sub-layers; 2 layer normalization layers $f_{attn}^{LN}, f_{mlp}^{LN}$ (J.2), an MLP layer $f^{mlp}$ (J.4), and a softmax self-attention layer $f^{attn}$ (J.3); takes a sequence $\{\boldsymbol{x}_{n \leq N}$ and outputs a sequence $\{\boldsymbol{y}_n\}_{n \leq N}$ in four steps.

1. First computes $\{\boldsymbol{y}_n^{attnln}\}_{n=1}^N$ using a layer normalization, i.e. $\boldsymbol{y}_n^{attnln} = f_{attn}^{LN}(\boldsymbol{x}_n)$ for all $n \leq N$.

2. Then it runs softmax self-attention to get $\{\boldsymbol{y}_n^{attn}\}_{n=1}^N = f^{attn}(\{\boldsymbol{y}_n^{attnln}\}_{n=1}^N)$.

3. The net output of the self-attention block is given by $\{\boldsymbol{y}_n^{attnblock}\}_{n=1}^N$, with $\boldsymbol{y}_n^{attnblock} = \boldsymbol{x}_n + \boldsymbol{y}_n^{attn}$ for all $n \leq N$.

4. Before passing to the MLP function, it is passed through another layer normalization, i.e. $\boldsymbol{y}_n^{mlpln} = f_{mlp}^{LN}(\boldsymbol{y}_n^{attnblock})$ for all $n \leq N$.

5. MLP function then returns $\{\boldsymbol{y}_n^{mlp}\}_{n=1}^N$ with $\boldsymbol{y}_n^{mlp} = f^{mlp}(\boldsymbol{y}_n^{mlpln})$ for all $n \leq N$.

6. $\boldsymbol{y}_n = \boldsymbol{y}_n^{mlp} + \boldsymbol{y}_n^{attnblock}$ for all $n \leq N$.

**Definition J.6** (Initialization of the weights in the transformer layer). The weights are initialized as follows:

$$\boldsymbol{C}^{mlp}, \boldsymbol{C}^{attn}, \boldsymbol{W}_Q, \boldsymbol{W}_K, \boldsymbol{W}_V \sim \mathcal{N}(\boldsymbol{0}, \frac{1}{\sqrt{d}}\boldsymbol{I}),$$

$$\boldsymbol{W} \sim \mathcal{N}(\boldsymbol{0}, \frac{\sqrt{2}}{\sqrt{m}}\boldsymbol{I}).$$

The parameters $\gamma, \boldsymbol{b}$ of the functions $f_{attn}^{LN}, f_{mlp}^{LN}$ have been initialized as $\boldsymbol{1}$ and $\boldsymbol{0}$ respectively.

**Lemma J.7** (Norm of the output of the MLP function, modification of lemma 7.1 in Allen-Zhu et al. (2019)). *For a given input sequence* $\{\boldsymbol{y}_n^{mlpln}\}_{n \leq N}$, *if* $\varepsilon \in (0, 1]$, *with probability at least* $1 - O(N) \cdot e^{-\Omega(m\varepsilon^2)}$ *over the randomness of* $\boldsymbol{C}^{mlp}, \boldsymbol{W}$, *we have*

$$\forall i \in [N] : \quad \left\|\boldsymbol{y}_n^{mlp}\right\| / \left\|\boldsymbol{y}_n^{mlpln}\right\| \in [1 - \varepsilon, 1 + \varepsilon].$$

**Lemma J.8** (Norm of the output of the layer normalization layers). *Given any input sequence* $\{\boldsymbol{x}_n\}_{n=1}^N$, *under the assumption for all* $n \leq N$, $\boldsymbol{x}_n - \sum_{i=1}^d x_{n,i}$ *isn't identically* $\boldsymbol{0}$, *we have*

$$\left\| \boldsymbol{y}_n^{attnln} \right\| = \sqrt{d}$$

*for all* $n \leq N$.

Similar result holds for $\{\boldsymbol{y}_n^{mlpln}\}_{n=1}^N$.

**Lemma J.9** (Norm of the output of the MLP block). *For a given input sequence* $\{\boldsymbol{y}_n^{attnblock}\}_{n \leq N}$, *with probability at least* $1 - \mathcal{O}(1)$ *over the randomness of* $\boldsymbol{C}^{mlp}, \boldsymbol{W}$, *we have*

$$\|\boldsymbol{y}_n\|^2 = \left\| \boldsymbol{y}_n^{mlp} + \boldsymbol{y}_n^{attnblock} \right\|^2 = \left\| \boldsymbol{y}_n^{attnblock} \right\|^2 + d$$
$$+ \mathcal{O}\left( \left\| \boldsymbol{y}_n^{attnblock} \right\| \log N + (d + \left\| \boldsymbol{y}_n^{attnblock} \right\|) \frac{\log^{3/2} N}{\sqrt{m}} \right),$$

*for all* $n \leq N$.

*Proof.* Combining lemmas J.7 and J.8, we have for the sequence $\{\boldsymbol{y}_n^{attnblock}\}_{n \leq N}$,

$$\left| \left\| \boldsymbol{y}_n^{mlp} \right\| / \sqrt{d} - 1 \right| \leq \mathcal{O}\left( \frac{\sqrt{\log N}}{\sqrt{m}} \right),$$

w.p. atleast $1 - \mathcal{O}(1)$.

Furthermore, due to the randomness of $\boldsymbol{C}^{mlp}$, we can further show that w.p. at least $1 - \mathcal{O}(1)$,

$$\left| \langle \boldsymbol{y}_n^{attnblock}, \boldsymbol{y}_n^{mlp} \rangle \right| \leq \mathcal{O}(\left\| \boldsymbol{y}_n^{attnblock} \right\| \log N),$$

for all $n \leq N$. Combining the two results, we have

$$\|\boldsymbol{y}_n\|^2 = \left\| \boldsymbol{y}_n^{mlp} + \boldsymbol{y}_n^{attnblock} \right\|^2 = \left\| \boldsymbol{y}_n^{attnblock} \right\|^2 + d$$
$$+ \mathcal{O}\left( \left\| \boldsymbol{y}_n^{attnblock} \right\| \log N + (d + \left\| \boldsymbol{y}_n^{attnblock} \right\|) \frac{\log^{3/2} N}{\sqrt{m}} \right).$$

$\square$

**Lemma J.10** (Norm of the output of the attention block). *For a given input sequence* $\{\boldsymbol{x}_n\}_{n \leq N}$, *if* $\varepsilon \in (0, 1]$, *with probability at least* $1 - O(1)$ *over the randomness of* $\boldsymbol{C}^{attn}, \boldsymbol{W}_V$, *we have*

$$\forall i \in [N]: \quad \|\boldsymbol{x}_n\|^2 \leq \left\| \boldsymbol{y}_n^{attnblock} \right\|^2 \leq \|\boldsymbol{x}_n\|^2 + \|\boldsymbol{x}_n\| + d + \mathcal{O}(\frac{\sqrt{\log N}}{\sqrt{d}}(d + \|\boldsymbol{x}_n\|)).$$

*Proof.* From the definitions of $\{\boldsymbol{y}_n^{attnln}\}$ and $\{\boldsymbol{y}_n^{attn}\}$, we have

$$\boldsymbol{y}_n^{attn} = \boldsymbol{C}^{attn} \boldsymbol{W}_V \sum_{j \leq N} a_{n,j} \boldsymbol{y}_j^{attnln}.$$

Thus, we can use a proof similar to the proof of lemma J.9 to argue that with the randomness of $\boldsymbol{C}^{attn}$ and $\boldsymbol{W}_V$,

$$\left| \frac{\|\boldsymbol{y}_n^{attn}\|}{\left\| \sum_{j \leq N} a_{n,j} \boldsymbol{y}_j^{attnln} \right\|} - 1 \right| \leq \mathcal{O}\left( \frac{\sqrt{\log N}}{\sqrt{d}} \right),$$

for all $n \leq N$ w.p. atleast $1 - \mathcal{O}(1)$.

Furthermore, due to the randomness of $\boldsymbol{C}^{attn}$, we can further show that w.p. atleast $1 - \mathcal{O}(1)$,

$$\left| \langle \boldsymbol{y}_n^{attn}, \boldsymbol{x}_n \rangle \right| \leq \mathcal{O}(\|\boldsymbol{x}_n\| \log N),$$

for all $n \leq N$.

Using Cauchy-Schwarz inequality, we must have

$$\left\| \sum_{j \leq N} a_{n,j} \boldsymbol{y}_j^{attnln} \right\| \leq \max_{j \leq N} \left\| \boldsymbol{y}_j^{attnln} \right\| = \sqrt{d}.$$

Thus, combining the results, we have

$$\left\| \boldsymbol{y}_n^{attnblock} \right\|^2 = \left\| \boldsymbol{y}_n^{attn} + \boldsymbol{x}_n \right\|^2 \leq \left\| \boldsymbol{x}_n \right\|^2 + d$$
$$+ \mathcal{O}\left( \left\| \boldsymbol{x}_n \right\| \log N + (d + \left\| \boldsymbol{x}_n \right\|) \frac{\log^{3/2} N}{\sqrt{d}} \right).$$

$\square$

**Lemma J.11** (Norm of the output of the transformer layer). *For a given input sequence $\{\boldsymbol{x}_n\}_{n \leq N}$, if $\varepsilon \in (0, 1]$, with probability at least $1 - \mathcal{O}(1)$ over the randomness of $\boldsymbol{C}^{attn}, \boldsymbol{W}_V, \boldsymbol{C}^{mlp}$, we have*

$$\forall i \in [N] : \quad \left\| \boldsymbol{x}_n \right\|^2 + d + \mathcal{O}(err) \leq \left\| \boldsymbol{y}_n \right\|^2 \leq \left\| \boldsymbol{x}_n \right\|^2 + 2d + \mathcal{O}(err),$$

*where $err = \mathcal{O}\left( \left\| \boldsymbol{x}_n \right\| \log N + (d + \left\| \boldsymbol{x}_n \right\|) \frac{\log^{3/2} N}{\sqrt{m}} \right)$.*

**Theorem J.1** (Square-root scaling of input norms). *At initialization, given any input sequence $\boldsymbol{x}_{1:N}$ and scalars $\alpha_{1:L}$, w.h.p. the intermediate sequences of $F$ (definition 2.1) satisfy*

$$\left\| \boldsymbol{y}_i^{(\ell)} \right\|_2^2 = \left\| \boldsymbol{x}_i \right\|_2^2 + \Theta(\sum_{j=1}^{\ell} \alpha_j^2 d), \quad \text{for all } 1 \leq i \leq N, 1 \leq \ell \leq L.$$

*Proof.* This follows from the fact that the transformer architecture is a stack of struturally identical $L$ transformer layers, and each transformer layer's output norm increases by $\Theta(d)$ compared to its input norm, as given by lemma J.11. $\square$

