# OpenReview forum: "Efficient stagewise pretraining via progressive subnetworks"
_ICLR.cc/2025/Conference — ICLR 2025 Poster_

### Official Review · Reviewer_tJ3v · 2024-10-27

**Soundness:** 2
**Presentation:** 2
**Contribution:** 2
**Rating:** 5
**Confidence:** 3

**Summary:**

This paper proposes a new approach to stage-wise pre-training of large models to improve the efficiency and computational cost. The main idea is to train random subnetworks within a base network for each stage of training and increase the complexity of the subnetworks as training progresses. Experimental results show some improvement in language models. Additionally, they include theoretical work to support their method, using a simple example of learning a polynomial function and demonstrate stability in stage-wise training.

**Strengths:**

1. The importance of improving pre-training efficiency is well justified, and it’s clear that addressing this issue is valuable.

2. Efforts have been made to offer theoretical justifications for the method.

**Weaknesses:**

1. For experiments on BERT, only a base model is used and the improvements are relatively marginal compared to other stacking or dropping methods and only 3 tasks in GLUE are reported. Some improvements for UL2 are also within the variance of the baseline.

2. Although the polynomial example illustrates how RAPTR learns lower and higher degree components, it is overly specific and lacks sufficient theoretical support to convince me that RAPTR is better than PLD.

3. Theorem 5.3 shows that the loss gap $|L_2(F) - L_1(F) |$ is upper bounded by the stability of the network and that is used to show that the losses between stages can be small for linear models in RAPTR. However, it is not clear if the same may hold for other stacking or dropping methods.

4. The empirical verification with BERT in section 5 does not seem to support the theory as in Fig 3, there is no linear decrease in loss gap as depth of model increases.

Formatting: Many typos, for example line 482: "layernorm is enabled or not respectively" should be the other way around, and figures with subplots are not labelled clearly.

**Questions:**

1. Is there a generalisation beyond the polynomial example for RAPTR capturing higher or lower degree components in each stage?

2. Can it be shown that the loss gap for other stacking or dropping methods are either unbounded or have a larger upper bound than RAPTR?

3. Can all GLUE tasks be reported together with results on BERT-large as a base model is quite small and easy to pre-train, while RAPTR is aimed at improving pre-training efficiency. More convincing experimental results using other models are also appreciated as the current results do not persuade me to use RAPTR.

4. Are we able to use RAPTR for fine-tuning on downstream tasks instead of pre-training?

---

> ### Author Response · Authors · 2024-11-21
>
> We thank the reviewer for their helpful comments and suggestions regarding our work. Please find our responses to your questions below.
>
> **Q1: Although the polynomial example illustrates how RAPTR learns lower and higher degree components, it is overly specific and lacks sufficient theoretical support to convince me that RAPTR is better than PLD.**
>
>
> **Response:**
> The theoretical setting on polynomials was meant to be an illustrative model on how RaPTr could perform better than PLD. The primary purpose is to provide useful insights for the success of RaPTr that can guide future work. While the difference in curriculum on the subnetwork size between RaPTr and PLD might seem minor at first, our theory highlights that such a change in curriculum could lead to substantial improvements in performance (which is clearly corroborated by our empirical results). The key insight is that **dropping more layers later in training reduces its capacity to capture complex correlations** in data. Our experiments on the polynomial setting clearly display this phenomenon. The experiments on BERT and UL2 also show that **RaPTr performs substantially better than PLD.** Modeling the real language distribution and providing theoretical results for it has proven very difficult for deep learning, and thus such intuitive theoretical results are the best we can hope for. We view the simplicity of the setting as a strength rather than weakness.
>
> **Q2: Theorem 5.3 shows that the loss gap $|L_2(F)−L_1(F)| $ is upper bounded by the stability of the network and that is used to show that the losses between stages can be small for linear models in RAPTR. However, it is not clear if the same may hold for other stacking or dropping methods.**
>
> **Response:** In section 5, we show that models trained by RaPTr are stable in training loss behavior across stage transitions. We create the theoretical setting to bound loss gap $|L_2(F) - L_1(F)|$ in order to explain the stability. Our framework shows that the change in loss can be bounded across the RaPTr stages. This, combined with the order in which subnetwork lengths are increased, our framework presents an effective order of minimizing loss over training. On the other hand, for PLD, one could potentially extend the stability framework to show small loss gaps across stages, however the reverse order of dropping of layers will limit the capacity of the model to minimize loss over training. **Moreover, for stacking, previous works have already observed instabilities in training loss behavior [1] . In general, it is not possible to show stability results for stacking.** This is because, new layers are introduced at each stage of training, which can arbitrarily impact optimization.
>
> **Q3: The empirical verification with BERT in section 5 does not seem to support the theory as in Fig 3, there is no linear decrease in loss gap as depth of model increases.**
>
> **Response:**
> The theoretical bound is an upper bound on the loss gap. The theoretical setting in section 5 was meant to give a worse case guarantee on how the loss could possibly be stable at stage transitions and how we could attribute the stability phenomenon to architecture components. In fact, empirically, we observe that the loss gap between a ($L-1$)-subnetwork and $L$-layer subnetwork is negative, which is an **even stronger and desirable finding than the theoretical result**, and understanding this is left for future work.

---

> > ### Comment · Reviewer_tJ3v · 2024-11-22
> >
> > Thank you to the authors for their efforts.
> >
> > Regarding Q1, while I do agree that theoretical results are quite difficult for deep learning, the example given seemed too specific and I am curious if it can be extended to a family of polynomials instead? For Q2, Is it possible to show that the loss gaps between stages for PLD has a higher upper bound than RaPTr? For Q3, I refer to line 468, "This implies that our upper bound on the loss gap $|L_2(F)−L_1(F)|$ will show a rough linear decrease with the depth of the model." which is the part that confused me about the linearity of the loss gap.
> >
> > I appreciate the results on GLUE and while not very convinced by them, I will refer to your results on UL2 in table 5 in the appendix which look good to me. Can I have an efficiency comparison in terms of flops and run time for the results in table 5? with regards to Baseline, Stacking and RAPTR.
> >
> > I also acknowledge that fine-tuning is not the main focus of the paper and should be deferred to future work. Though, it seems that LLMs will no longer be pre-trained from scratch but will instead rely on continual pre-training and fine-tuning.

---

> ### Author Response · Authors · 2024-11-21
>
> **Q4: Can all GLUE tasks be reported together with results on BERT-large as a base model is quite small and easy to pre-train?**
>
> Response: Based on the reviewer’s suggestion, we **ran evaluations on all GLUE tasks** and reported the results below. Overall, we continue to observe that RaPTr trained models are competitive or better than the best tuned stacking versions. We used BERT experiments mainly for ablations and extensive experiments on design choices. These were mainly meant to show that dropping based strategies can be made competitive to stacking (and even their best versions) which PLD had failed to show before.
>
> For a larger scale setting, we have the UL2 experiments using a decoder-only 1.5B parameter model, a setting that is more prominently used these days and is also much larger than BERT-large. Extensive pre-training and evaluations on a suite of 19 benchmarks for UL2 show that RaPTr can show a large improvement in few-shot evaluations (please see table 5 in the appendix for details results).
>
> Revised table 2:
> |            | Schedule | Loss | SST-2                | QNLI                 | MNLI                 | SNLI                 | RTE                 | MRPC                | CoLA                | QQP                 | Avg  |
> |------------|----------|------|----------------------|----------------------|----------------------|----------------------|---------------------|---------------------|---------------------|---------------------|------|
> | bert2bert  | 3-6-12   | 1.85 | 89.3 (0.1)           | 84.5 (0.1)           | 76.9 (0.0)           | **87.0** (0.0)       | 56.1 (0.7)          | 75.2 (1.3)          | 77.3 (0.4)          | 89.3 (0.0)          | 79.5 |
> | MSG        | 3-6-12   | 1.87 | 88.4 (0.2)           | 83.9 (0.2)           | 76.4 (0.3)           | **87.0** (0.2)       | 56.0 (1.1)          | 74.8 (1.0)          | 76.6 (1.1)          | 89.4 (0.1)          | 79.1 |
> | MSG        | 6-9-12   | 1.84 | **89.7** (0.3)       | 85.6 (0.2)           | **77.7** (0.1)       | 86.6 (0.2)           | **59.5** (1.1)      | 77.2 (1.1)          | **78.9** (0.2)      | **89.6** (0.1)      | 80.6 |
> | LEMON      | 6-9-12   | **1.83** | **89.8** (1.1)       | **87.2** (0.0)       | 77.3 (0.1)           | 86.4 (0.1)           | **58.4** (1.5)      | 79.8 (0.6)          | 78.0 (0.3)          | **89.7** (0.1)      | **81.1** |
> | MSG        | 6-9-12   | 1.84 | 89.0 (0.7)           | 85.1 (0.0)           | 77.5 (0.3)           | 86.9 (0.1)           | 57.7 (1.9)          | **79.9** (0.8)      | 77.5 (0.2)          | 89.5 (0.0)          | 80.4 |
> | Stacking   | 6-9-12   | 1.84 | 89.0 (0.4)           | 85.3 (0.1)           | 77.0 (0.1)           | 86.2 (0.1)           | 56.6 (1.3)          | 76.4 (1.1)          | 78.0 (0.4)          | 89.3 (0.0)          | 79.7 |
> | **RaPTr** | 6-9-12   | **1.83** | 89.3 (0.6)           | **85.8** (0.3)       | **78.6** (0.2)       | **87.0** (0.1)       | 57.7 (0.5)          | **80.8** (0.4)      | **79.1** (0.7)      | **89.6** (0.1)      | **81.0** |
>
>
> Revised table 4:
> |                     | Eval Loss | SST-2    | QNLI    | MNLI    | SNLI    | RTE     | MRPC    | CoLA    | QQP     | Avg   |
> |---------------------|-----------|----------|---------|---------|---------|---------|---------|---------|---------|-------|
> | **Head Operators**  |           |          |         |         |         |         |         |         |         |       |
> | MSG                 | 2.07      | 87.5 (0.1) | 83.4 (0.1) | 73.3 (0.3) | 85.5 (0.2) | 52.3 (0.4) | 72.5 (1.1) | 73.5 (0.2) | 88.6 (0.0) | 77.1  |
> | Attention-RaPTr     | **1.84**  | **89.8** (0.6) | **86.4** (0.2) | **77.5** (0.1) | **89.5** (0.4) | **60.5** (1.3) | **80.4** (1.7) | **78.7** (0.1) | **89.7** (0.0) | **81.6**  |
> | **Intermediate MLP size Operators** | | | | | | | | | |
> | MSG                 | 1.88      | **89.0** (0.3) | **85.5** (0.2) | 77.2 (0.2) | 87.2 (0.0) | **58.3** (0.6) | **80.0** (1.9) | 77.1 (1.0) | 89.6 (0.1) | **80.5**  |
> | MLP-RaPTr           | **1.84**  | 88.7 (0.3) | 85.3 (0.1) | **77.3** (0.3) | **88.6** (0.1) | 55.8 (1.9) | 79.8 (1.1) | **78.3** (0.2) | 89.6 (0.1) | 80.4  |
>
>
> **Q5: Are we able to use RAPTR for fine-tuning on downstream tasks instead of pre-training?**
>
> Response: Thank you for the interesting question. Indeed RaPTr can in principle be used for finetuning from a pre-trained checkpoint, by varying the layer dropping budget in stages. This is another potential advantage over stacking-based method because RaPTr maintains the model identity . However our primary focus in this paper was pre-training which is significantly more resource expensive than fine-tuning. We believe extending RaPTr to fine-tuning is an important future work, which requires understanding the structure of existing pre-trained models, and is beyond the scope of our current work.
>
> **References.**
>
> 1. Efficient Training of Language Models using Few-Shot Learning. Reddi et al.’2023

---

> ### Author Response · Authors · 2024-11-22
>
> We thank the reviewer for continuing the discussion in the rebuttal. Please find our responses to your questions below.
>
> **Comment 1: Regarding Q1, while I do agree that theoretical results are quite difficult for deep learning, the example given seemed too specific and I am curious if it can be extended to a family of polynomials instead?**
>
> **Response:** We can generalize the polynomial to a higher degree and more general polynomial of the form $\sum_i \prod_{j \in p(i)} x_j,$ where each $p(i) \subseteq [n]$ and $p(i) \subseteq p(k)$ for all $k>=i$. The important property that we need is the hierarchical nature of components in the polynomial. Our proof builds on existing theoretical works [1,2] that show gradient descent analysis for such hierarchical polynomials. However, our primary aim was to formalize the **difference between PLD and RaPTr through a short example, which can be generalized to higher degree polynomials.** Experimentally, we show the behavior of RaPTr and PLD in figure 2 on a degree 10 polynomial.
>
> The primary message of  our theoretical statement and figure 2 experiments is that **RaPTr ensures simple-to-complex learning**, which is useful when the simple features can help the model to learn difficult features in the later stages of training. On the other hand, due to decreasing capacity of the model, **PLD will restrict the model’s ability** to learn the difficult features towards the end of training.
>
> **References:**
>
> 1: Sgd learning on neural networks: leap complexity and saddle-to-saddle dynamics. Abbe et al’2023
>
> 2: The staircase property: How hierarchical structure can guide deep learning. Abbe et al’2021
>
>
> **Comment 2: For Q2, Is it possible to show that the loss gaps between stages for PLD has a higher upper bound than RaPTr?**
>
> **Response:** Our theoretical upper bounds in **section 5** won’t differentiate between PLD and RaPTr on loss gap across stages, and PLD will also be stable. The issue of PLD is not that of loss stability, but the increasing order of layer dropping probabilities and its restrictive capability to capture higher order features, which is studied in **section 3**.
>
> **Comment 3: For Q3, I refer to line 468, "This implies that our upper bound on the loss gap $|L_2(F)−L_1(F)|$  will show a rough linear decrease with the depth of the model." which is the part that confused me about the linearity of the loss gap.**
>
> **Response:** Thank you for catching this! We meant to say that our theoretical upper bound on the loss gap, which bounds the absolute difference between $L_2(F)$ and $L_1(F)$, will show a linear decrease with the depth of the model. Our theoretical upper bound is a worst case upper bound, through which we can only infer $L_2(F) < L_1(F) + O(1/L)$. On the other hand, in practice, we observe that $L_2(F)$ is less than $L_1(F)$, which shows that **loss can decrease across stages** in RaPTr, and is an **even stronger and desirable finding than the theoretical result**. We will clarify this clearly in the next version.
>
> **Comment 4: Can I have an efficiency comparison in terms of flops and run time for the results in table 5? with regards to Baseline, Stacking and RAPTR.**
>
> **Response:** Table 5 uses Baseline, Stacking and RaPTr runs from table 3. The theoretical FLOP benefits for Stacking and RaPTr compared to baseline are 1.2x, while the runtime benefits compared to baseline on our machine are roughly 1.19x. The run-times of training each subnetwork have been outlined in Appendix F and the efficient RaPTr implementation in distributed setting is outlined in Appendix G. We are happy to answer more questions about the UL2 experiments.
>
> **Comment 5: I also acknowledge that fine-tuning is not the main focus of the paper and should be deferred to future work. Though, it seems that LLMs will no longer be pre-trained from scratch but will instead rely on continual pre-training and fine-tuning.**
>
> **Response:** Thank you for proposing this! Understanding the structure of existing pre-trained models is important for using RaPTr during continued pre-training, which is something that we are planning to do as future work.

---

> > ### Comment · Reviewer_tJ3v · 2024-11-23
> >
> > Thank you to the authors for their response.
> >
> > For comment 2, I refer to Q2 in the question part of the review, then if the theoretical upper bounds in section 5 can differentiate RaPTr and other stage-wise pre-training methods? Otherwise, it is not convincing that RaPTr is better.
> >
> > Regarding comment 3, it would be good if the authors can revise the paper to clarify the comment on the Empirical verification with BERT. As we can submit revisions during this discussion period, I would encourage the authors to so with their clarification.
> >
> > Okay, I understand the results on UL2. However, if the experiments on BERT were primarily for empirical evidence, the experiments to show that RaPTr is a more effective pre-training recipe are not very extensive. Do you have other experiments on llms such as Llama-3.2-1B as well?

---

> ### Author Response · Authors · 2024-11-23
>
> We thank the reviewer for their continued discussion in the rebuttal phase. Please find our responses to your comments below.
>
> **Comment 1: For comment 2, I refer to Q2 in the question part of the review, then if the theoretical upper bounds in section 5 can differentiate RaPTr and other stage-wise pre-training methods? Otherwise, it is not convincing that RaPTr is better.**
>
> **Response:** The theoretical statement in section 5 was meant as a study on RaPTr and not comparison to other training methods. We start the section by asking the question **“What are the general conditions under which training loss under RAPTR stays stable at stage transitions?”** Furthermore as we remarked in our previous response, stacking does not demonstrate this loss stability empirically, whereas our results show that RaPTr can without requiring any additional tricks. This provides a contrast between stacking and dropping based methods. **Results in Section 3 already distinguish between RaPTr and PLD, and the experiments in section 4 show performance improvements with RaPTr on BERT and UL2.**
>
>
> **Comment 2: Regarding comment 3, it would be good if the authors can revise the paper to clarify the comment on the Empirical verification with BERT. As we can submit revisions during this discussion period, I would encourage the authors to do so with their clarification.**
>
> **Response:** We have uploaded a new version, with the aforementioned clarification. Changes have been highlighted in magenta for easy identification.
>
> **Comment 3: Okay, I understand the results on UL2. However, if the experiments on BERT were primarily for empirical evidence, the experiments to show that RaPTr is a more effective pre-training recipe are not very extensive. Do you have other experiments on llms such as Llama-3.2-1B as well?**
>
> **Response:**
> We respectfully disagree with the reviewer’s claims that experiments are not very extensive. We have covered two popular settings of **pre-training masked language models and causal language models**, both at **BERT** scale (with GLUE evaluations) and a much larger **1.6B parameter** scale (with evaluations on 19 benchmarks, 8 benchmarks marked as SuperGLUE). This is as, or arguably more extensive compared to previously published works on efficient training (please see a detailed note on this towards the end). Furthermore as an added contribution, we also provide theoretical results and intuitions supporting various design choices like schedule of dropping probability and loss stability. Such analyses are largely missing from the largely empirical work in this field.
>
> **Scale of experiments:** We note that each run on UL2-1.6B for 400k steps took us 3 days of training. We have reported numbers roughly for 30 UL2 runs (including all random seeds), amounting to 90 days worth of training. These experiments are at a much larger scale, something not considered in previous efficient training works.
>
> **Comparison of extensive-ness with prior works:** Relevant published works on stacking (as recent as ICLR 2024) primarily report results on Bert-base models and one additional setting at a scale that is much smaller than the results that we have reported on. Here are some references:
> - Masked Structural Growth for 2x Faster Language Model Pre-training. Yao et al. ICLR 2024. **(Bert-base, BERT-large, and GPT-2)**
> - LEMON: Lossless model expansion. Wang et al. ICLR 2024: **(Bert-base and Vision Transformer.)**
> - Efficient Training of Language Models using Few-Shot Learning. Reddi et al. ICML 2023.  **(Bert-base and Bert-large)**
> - bert2BERT: Towards Reusable Pretrained Language Models, ACL 2021. **(Bert-base, GPT-2)**
> - Efficient Training of BERT by Progressively Stacking. Gong et al. ICML 2019. **(Bert-base)**
>
> **Regarding LLama 3.2,** the pre-training data details are not public so those experiments are not possible to run. Even so, LLama 3.2 1B has been pretrained with 9T tokens for 370k GPU hours which is nearly impossible to run under any reasonable resource availability (e.g. even with 100 GPUs it would require 150 days!). Conceptually there is no reason to believe that the results will be very different from the 1.6B UL2 experiments presented in the paper since they are solving similar autoregressive problem.
>
> Link: https://huggingface.co/meta-llama/Llama-3.2-1B

---

> > ### Comment · Reviewer_tJ3v · 2024-11-27
> >
> > Thank you for your detailed responses and revision. I apologise for my delayed reply.
> >
> > I will raise my score accordingly. My main concerns remain in regards to the not quite convincing theoretical results of RaPTr over other stage-wise pre-training methods as well as the experimental results which are not sufficiently extensive.
> > While Llama-3.2-1B is not a good example for additional experiments, other models can be used to supplement your results. I do think UL2 is not as popularly used as models like ViT or GPT-2 and I would have preferred experiments on pre-training more widely used models to boost the credibility and practical value of the experiments.
> >
> > As such, I keep my score to 5.

---

### Official Review · Reviewer_zzqn · 2024-10-30

**Soundness:** 4
**Presentation:** 4
**Contribution:** 3
**Rating:** 8
**Confidence:** 3

**Summary:**

This paper argues that stagewise dropping strategies, such as layer dropping, can be effective for efficiently training large language models. The paper proposes a stagewise training framework of progressively growing a subnetwork and shows that it generalizes layer dropping. They theoretically illustrate the effectiveness of this strategy and moreover empirically demonstrate effectiveness in speeding up training on standard benchmarks.

**Strengths:**

The problem of efficient training is of increasing importance as we scale to larger models and data sets. The paper is well-written and comprehensive, covering theoretical justification, detailed numerical analysis, and implementation guidelines.

**Weaknesses:**

In the numerical evaluations, the selected settings of Rel. FLOPS seemed somewhat arbitrary (e.g., fixing to 1.33x for Table 1). It would be nice to get some intuition why these were chosen and the sensitivity to these experimental settings (e.g., how do the results hold as we sweep the Rel. FLOPS).

**Questions:**

Please see Weaknesses.

---

> ### Author Response · Authors · 2024-11-22
>
> We thank the reviewer for their positive and supportive comments about our work. Please find our responses to your questions below.
>
> **Q1:In the numerical evaluations, the selected settings of Rel. FLOPS seemed somewhat arbitrary (e.g., fixing to 1.33x for Table 1). It would be nice to get some intuition why these were chosen and the sensitivity to these experimental settings (e.g., how do the results hold as we sweep the Rel. FLOPS).**
>
>
> **Response:**
> Our experiments follow schedule selections from prior work [Reddi et al. 2023] -- Inverse, Equal and Proportional. For Equal, we split training into equal stages. For Proportional, we increase the length of a stage in proportion to index of the stage (more details in Appendix E). For Inverse, we decrease the length of a stage in proportion to the index of the stage. The FLOP computation was then given by the average number of layers used in subnetworks during training. Thus, for a 4-stage schedule with subnetwork lengths 6, 8, 10, 12 (used in table 1); Equal schedule will give an average layer use of 9 layers out of 12, effectively reducing the amount of FLOPs 1.33x. Similarly Proportional schedule will reduce FLOPs by 1.2x and Inverse schedule will reduce FLOPs by 1.5x. We primarily focus on Equal scheduling for BERT-base and Proportional scheduling for UL2.
>
> **Effect of Increase in Rel. FLOPs** Intuitively one would expect that a more aggressive schedule with higher FLOPs reduction will eventually lead to degradation of model quality. To verify this, we ran some experiments increasing the number of Rel. FLOPs by switching between Inverse, Equal, and Proportional. As expected, we see an improvement in the model’s performance with lesser speedup in both loss and downstream evaluations. We report experiments on BERT-base after 300k steps of training below for reference.
>
>  | Schedule | Stage length | Rel. FLOP reduction | Eval loss | SST-2 | QNLI |  MNLI | Avg |
> |-------------|---------------|--------------------|---------------------|--------------------|---------------------|---------------------|---------------------|
> |6-8-10-12 | Inverse | 1.5x | 1.86 | 88.4(0.2) | 86.2(0.4) | 77.8(0.1) | 84.1 |
> ||Equal | 1.33x | 1.84 | 88.7(0.2) | 86.1(0.3) | 78.2(0.3) | 84.3 |
> | | Proportional | 1.2x |  1.81 | 88.6(0.3) | 86.4(0.1) | 78.8(0.2) | 84.6 |
>
>
>
> **References**
> 1. Efficient Training of Language Models using Few-Shot Learning. Reddi et al.’2023

---

### Official Review · Reviewer_Edxi · 2024-11-03

**Soundness:** 3
**Presentation:** 3
**Contribution:** 3
**Rating:** 6
**Confidence:** 4

**Summary:**

This paper proposes a stagewise pretraining method, progressive subnetwork training. This method randomly selects subnetworks to train in each stage and gradually increases the size of the selected subnetworks in stages. They also discussed why layer dropping can hurt the models’ ability to capture complex correlations in the data. The authors verify the effectiveness of the proposed progressive subnetwork training method by pretraining BERT and UL2.

**Strengths:**

The proposed progressive subnetwork training method is straightforward and easy to understand.
The experiments show that the proposed method is simple but effective.
The paper is well-structured and written.

**Weaknesses:**

It is good that try to theoretically explain why layer dropping methods cannot achieve good performance. However, the illustration setting that using Polynomial learning and 2-layer residual network is too naïve, which makes the conclusion less convincing.

**Questions:**

Why can RaPTr achieve better performance than the baseline full-model? The baseline full-model is well-trained and is supposed to achieve the best performance.
How about the performance when more FLOPs are reduces? For example, reduce the FLOPs of the baseline model to half or 1/3?

---

> ### Author Response · Authors · 2024-11-21
>
> We thank the reviewer for their positive comments about the structure and motivation of the paper. Please find our responses to your questions below.
>
> **Q1: Why can RaPTr achieve better performance than the baseline full-model?**
>
> **Response:**
> Thank you for the question!
> We only have some preliminary hypotheses, like improved inductive bias of a model when the model is explicitly trained to build in a simple-to-complex fashion. Earlier stages train the model to behave as a mixture of random sub-networks, while later stages merge those random sub-networks to form one single model. On the other hand, baseline model training trains all the layers together, which could potentially result in the layers overfitting to noise. We believe that the study on inductive bias of RaPTr deserves extensive exploration on its own.
>
>
> **Q2: How does RaPTr compare to the performance of a baseline model, when the amount of FLOPs is reduced in the baseline model?**
>
> **Response:** We have compared RaPTr to a baseline model, whose FLOP counts have been reduced to match RaPTr’s, for BERT (table 7 in appendix) and UL2 (below). We reproduce those numbers here for ease of reference. With reduced FLOPs, baseline is worse than RaPTr that uses the same FLOPs.
>
> Performance on UL2:
>
> | Model   | Rel. FLOPs | Eval Loss | Trivia QA | Tydi QA | SQuADv2 | SGLUE | Avg.  |
> |--------------------------------------|------------|-----------|-----------|---------|---------|-------|-------|
> | Baseline         | 1.2        | 2.06      | 25.0      | 34.4    | 42.1    | 60.0  | 40.4  |
> | Equiflop Baseline | 1          | 2.07      | 24.3      | 35.7    | 42.4    | 60.6  | 40.7  |
> | RaPTr                                | 1          | 2.06      | 25.8      | 36.7    | 44.1    | 60.9  | 41.9  |
>
>
> Performance on BERT-Base:
>
>  | FLOPs   | Equiflop Baseline | Stacking | RaPTr |
> |---------|----------|----------|-------|
> | 75k     | 2.09     | 2.02     | 2.01  |
> | 170k    | 1.90     | 1.88     | 1.86  |
> | 510k    | 1.74     | 1.75     | 1.73  |
>
> Here, we report the performance of the model across 3 training horizons, where we match the FLOPs of baseline model, Stacking, and RaPTr.

---

### Official Review · Reviewer_Dgxw · 2024-11-04

**Soundness:** 3
**Presentation:** 3
**Contribution:** 3
**Rating:** 8
**Confidence:** 3

**Summary:**

This paper leverages the idea of stochastic depth for the training of language models. The paper proposed a particular training schedule (RAPTR) suitable for training with stochastic depth, where the depth of the network increases over the course of model training, ultimately training the full model. As a significant number of layers are skipped, this translates to direct gains in wall-clock time. The proposed RAPTR training scheme was evaluated on two encoder-decoder language models including BERT and UL2. Results on a small set of benchmarks indicate competitive performance w.r.t. the full model training and even outperforms it in some cases due to this acting as a regularization scheme.

**Strengths:**

- The paper is well written
- The approach is well motivated by the literature on stochastic depth
- The obtained results on a small set of benchmarks indicate either competitive or superior performance than the control (i.e., full model training) and directly translate to savings in wall-clock time

**Weaknesses:**

- The paper only focused on encoder-decoder models. I would like to see similar results on simple decoder-only language models, which are much more prevalent.
- Evaluation on only a limited number of tasks. I would expect evaluations on a very large number of tasks from eval-harness as evaluation costs are relatively modest in comparison to the training cost.
- Important implementation details are scattered throughout the paper. I would expect all of the details such as sequence length, and batch sizes to be specified at a single location to make it easier to understand.
- Simple implementation details buried in the appendix such as how to provide gradients to all the layers, which I expect to be a part of the main paper. It can be significantly shorter though by just stating that we need a way to compute gradients for all the layers even if that layer didn't participate in that round. Perhaps just include another trivial implementation that works even in distribution reduction i.e., to multiply block $i$ output with $\alpha_i$. This ensures that gradient is computed for all parameters, however, without any gain in wall-clock time.

**Questions:**

- Unclear how the authors decided to take specific mixtures of specific datasets. Was there no off-the-shelf dataset suitable for this analysis (such as FineWeb / FineWeb-edu)?

---

> ### Author Response · Authors · 2024-11-21
>
> We thank the reviewer for their positive comments about the structure of the paper and the motivation behind RaPTr. Please find our responses to your question below.
>
> **Q1: The paper only focused on encoder-decoder models. How would the results change with  simple decoder-only language models, which are much more prevalent?**
>
> **Response:** In fact UL2 experiments are with a decoder-only language model. UL2 training only changes the objective function (60% causal LM, 20% prefix LM and 20% span corruption), but still maintains the model architecture to be decoder-only.
>
> **Q2: “limited number of evaluations on UL2 in the main paper?”**
>
> **Response:** Our UL2 evaluations have reported the performance of trained models across **a suite of 19 benchmarks** (8 benchmarks averaged as SuperGLUE) in single-shot and multi-shot settings (please refer to Table 5). These benchmarks already evaluate **a wide range of capabilities** like closed-domain question answering, open-domain question answering, completion, and SuperGLUE. We restrict our evaluations to these benchmarks, as our experiments are at small scale with the model size being 1 billion parameters, and so we won’t observe non-trivial performance on math benchmarks like Gsm8k and MATH. However, the evaluation scores from these 19 datasets already shows that RaPTr can perform better than baseline (full-model) and stacking.
>
>
>
>
> **Q3: How did the authors decide the mixture composition of pre-training dataset? Why didn’t the authors use off-the-shelf dataset (such as FineWeb / FineWeb-edu)?**
>
> **Response:** We had conducted our experiments on UL2 prior to the release of FineWeb and DCLM. Our dataset mixtures were decided based upon the pre-training data mixtures used in LLaMA-1 [1]. Arxiv, C4 (Raffel et al., 2020), Github, and Wikipedia (Foundation) datasets were mixed with mixing ratios 9%, 57%, 17%, 17%, roughly 1:6:2:2 ratio as used in LLaMA-1.
>
> Suggestions:
>
> **1: “Simple implementation details buried in the appendix such as how to provide gradients to all the layers, which I expect to be a part of the main paper. “**
>
> **Response:**
> We thank the reviewer for pointing this out.  As our paper was dense with both empirical results and theoretical claims, we decided to move all our implementation details of RapTr to appendix G. We will include a summary of section G in the final version.
>
> **2: ”Important implementation details are scattered throughout the paper. All of the details such as sequence length, and batch sizes should be specified at a single location to make it easier to understand.”**
>
> **Response:** We thank the reviewer for pointing this out. We tried to give all details on our experiments in sections D and E. We will improve these sections in the next version.
>
>
> References:
>
> 1. LLaMA: Open and Efficient Foundation Language Models . Touvron et al.

---

> ### Comment · Reviewer_Dgxw · 2024-11-28
> **Thanks for responding to my comments**
>
> I want to thank the authors for responding to my comments. ICLR allows updates to the paper, so the comment from the authors that "they will update it in the next version" indicates they would like to retain their current version, which is of course up to the authors as it was just a suggestion from a reader's perspective. I think that this is a useful contribution, and I would increase my score from 6 to 8 to get it over the line.

---

### Meta-Review · Area_Chair_skVs · 2024-12-20

**Metareview:**

The paper introduces a stage-wise training framework for large language models, leveraging  progressive subnetwork growth. RaPTr skips layers during early training stages and gradually increases network depth, reducing computational FLOPs and wall-clock time. The authors provide both theoretical insights and empirical evidence across two model families—BERT (encoder-decoder) and UL2 (decoder-only). Results show that RaPTr can speed up training while achieving strong performance.

The proposed approach is straightforward, grounded in existing literature and well-motivated. It directly addresses efficiency in pretraining, an increasingly critical challenge in large-scale model. Despite its strengths, the theoretical analysis, while insightful, relies on oversimplified polynomial learning examples, limiting its generalizability to broader architectures. The experiments are limited in scope, with a focus on relatively small-scale models (BERT-Base and UL2-1B) rather than widely used larger models (e.g., GPT-series or ViT). While theoretical and experimental limitations exist, RaPTr demonstrates clear advantages in wall-clock time and computational savings without compromising performance, making it a significant step thus accept is recommended.

**Additional Comments On Reviewer Discussion:**

Reviewer Dgxw and Reviewer zzqn supported acceptance, citing the method's practical efficiency and theoretical insights. Reviewer tJ3v raised concerns about limited experiments and theoretical generalization, but the authors' rebuttal, additional results, and revisions addressed some of these, leading to a raised score.  While reviewers appreciated the experiments on BERT and UL2, they noted gaps in evaluating larger or more widely adopted models and limited theory. The authors argued resource constraints and the comparable significance of UL2 results.
The reliance on polynomial examples in theory was seen as overly specific. The authors acknowledged this limitation but clarified its purpose as illustrative rather than exhaustive.

---

### Decision · Program_Chairs · 2025-01-22

Accept (Poster)